# Investigating immune and non-immune cell interactions in head and neck tumors by single-cell RNA sequencing

Cornelius H. L. Kürten [1,2,3,4,13], Aditi Kulkarni[2,3,4,13], Anthony R. Cillo [2,3,5], Patricia M. Santos[2,3,4], Anna K. Roble [2,3,4], Sayali Onkar[2,3,5,6], Carly Reeder[2,3,4], Stephan Lang[1], Xueer Chen[7], Umamaheswar Duvvuri[4], Seungwon Kim[4], Angen Liu[4,8], Tracy Tabib[9], Robert Lafyatis[9], Jian Feng[2,10,11], Shou-Jiang Gao [2,10,11], Tullia C. Bruno [2,3,5], Dario A. A. Vignali[2,3,5], Xinghua Lu [7], Riyue Bao[2,12], Lazar Vujanovic [2,3,4,13] & Robert L. Ferris[2,3,4,5✉]

Head and neck squamous cell carcinoma (HNSCC) is characterized by complex relations between stromal, epithelial, and immune cells within the tumor microenvironment (TME). To enable the development of more efficacious therapies, we aim to study the heterogeneity, signatures of unique cell populations, and cell-cell interactions of non-immune and immune cell populations in 6 human papillomavirus (HPV)$^+$ and 12 HPV$^-$ HNSCC patient tumor and matched peripheral blood specimens using single-cell RNA sequencing. Using this dataset of 134,606 cells, we show cell type-specific signatures associated with inflammation and HPV status, describe the negative prognostic value of fibroblasts with elastic differentiation specifically in the HPV$^+$ TME, predict therapeutically targetable checkpoint receptor-ligand interactions, and show that tumor-associated macrophages are dominant contributors of PD-L1 and other immune checkpoint ligands in the TME. We present a comprehensive single-cell view of cell-intrinsic mechanisms and cell-cell communication shaping the HNSCC microenvironment.

[1] Department of Otorhinolaryngology, Head and Neck Surgery, University Hospital Essen, University Duisburg-Essen, Essen, Germany. [2] UPMC Hillman Cancer Center, University of Pittsburgh, Pittsburgh, PA, USA. [3] Tumor Microenvironment Center, UPMC Hillman Cancer Center, Pittsburgh, PA, USA. [4] Department of Otolaryngology, University of Pittsburgh, Pittsburgh, PA, USA. [5] Department of Immunology, University of Pittsburgh, Pittsburgh, PA, USA. [6] Graduate Program of Microbiology and Immunology, University of Pittsburgh School of Medicine, Pittsburgh, PA, USA. [7] Deparment of Biomedical Informatics, University of Pittsburgh, Pittsburgh, PA, USA. [8] Department of Pathology, University of Pittsburgh, Pittsburgh, PA, USA. [9] Division of Rheumatology and Clinical Immunology, University of Pittsburgh, Pittsburgh, PA, USA. [10] Department of Microbiology and Molecular Genetics, University of Pittsburgh, Pittsburgh, PA, USA. [11] Cancer Virology Program, UPMC Hillman Cancer Center, Pittsburgh, PA, USA. [12] Department of Medicine, University of Pittsburgh, Pittsburgh, PA, USA. [13] These authors contributed equally: Cornelius H. L. Kürten, Aditi Kulkarni, Lazar Vujanovic. ✉email: ferrisrl@upmc.edu

Traditionally, head and neck squamous cell carcinoma (HNSCC) treatments are based on approaches that target tumor cell oncogenic activities[1], however, the effectiveness of these therapies is limited by intra-tumor heterogeneity[2,3]. HNSCC treatment has seen a paradigm shift with the introduction of cancer immunotherapy, with the targeting of immune checkpoints such as the PD-1/PD-L1 interaction showing the greatest clinical activity[4–6]. The effectiveness of immunotherapies is affected by numerous resistance mechanisms, including immune exclusion, immune editing, decreased antigen presentation, nutrient deprivation, and immune suppression by soluble or cellular mechanisms[7]. These dynamics warrant a more holistic interrogation of the complex cancer–immune–stroma interaction in the tumor microenvironment (TME).

The study of heterogeneous cellular populations has been greatly facilitated by the development of single-cell RNA sequencing (scRNAseq), which allows for a comprehensive investigation into the transcriptomic profiles of individual cell populations[8]. This technique has been employed to study the TME in a variety of cancer types and reiterated concepts such as high intra- and inter-tumor heterogeneity, epithelial to mesenchymal transition (EMT), and the functional spectrum of tumor-infiltrating lymphocytes (TIL) and myeloid cells[9–11]. However, previous studies in HNSCC have been limited by one or a combination of the following aspects: a focus only on HPV⁻ disease, profiling TIL or malignant cells separately, a limited number of patients and/or cells profiled[9,10,12]. Using large-scale droplet-based technology[13], we aim to overcome these limitations and concomitantly interrogate the HNSCC TME of both HPV+ and carcinogen-induced (HPV⁻) etiologies.

We harness existing and novel technical and informatic methods to characterize major non-immune and immune cell types of the HNSCC TME and provide insight into their cancer–immune–stroma relationships. This study provides insights beyond previous scRNAseq studies reflecting dramatic complexity of HNSCC epithelial cells of differing etiologies, including directly mapping HPV encoded gene transcripts[10]. We leverage our comprehensive data set as a resource to explore differences between inflamed vs. non-inflamed as well as HPV+ vs. HPV⁻ micro-milieus. Using this data set, we observe an elastic sub-state of fibroblast differentiation and its negative prognostic impact that has not been previously described in HNSCC scRNAseq studies, and investigate cell-to-cell interactions of therapeutically relevant immune checkpoint receptor–ligand pairs, such as tumor-associated macrophages (TAM), which are major contributors of PD-L1 to CD8+ T cell interactions in HNSCC[14]. HPV encoded gene heterogeneity within tumor cells provides insights into differential pathways and microenvironmental impact.

## Results

**High-dimensional scRNAseq data analysis reveals cellular complexity of the HNSCC TME.** To capture a representative number of immune and non-immune cells per patient, we sorted CD45+ (immune) and CD45⁻ (epithelial and stromal) cells from freshly resected HNSCC tumors prior to scRNAseq (Fig. 1A). The cohort included 18 treatment-naive patients (6 HPV+ and 12 HPV⁻; Supplementary Table 1), out of which 15 had paired immune and non-immune cells, as well as matched peripheral blood leukocytes (PBL). We aimed to recover 2000 cells each of TIL, TME non-immune cells (i.e., epithelial and stromal cells), and PBL from each patient. After quality control, 134,606 cells with 1077 median genes per cell were retained and visualized as a comprehensive overview plot using Uniform Manifold Approximation and Projection (UMAP) (Fig. 1B)[15–18]. Highly diverse

cell populations of the TME were identified transcriptomically using canonical markers (Fig. 1C and Supplementary Fig. 1A). Diverging distributions of cell clustering were observed when depicting the UMAPs based on patient contribution, HPV status or tissue of origin, and gene count (Fig. 1B and Supplementary Fig. 1B). Additionally, a t-SNE plot depicting the number of genes per cell was also generated (Supplementary Fig. 1B). Absolute, as well as relative, contributions of each cell type to the cohort were analyzed (Supplementary Fig. 1C). Some clusters were enriched in cells from PBL (0: CD4+ T cells, 1 and 16: monocytes; 7: CD8+ T cells; 8: NK cells), while others were distinctly dominated by TIL [2 and 3: CD8+ T cells; 12, 18, 27: CD4+ T cells; 4: Treg; 23: NK cells; 6: macrophages; 21: myeloid dendritic cells (DC)]. Clusters 5 (B cells) and 25 (plasmacytoid dendritic cells; pDC) consisted of both tumor and blood-derived cells, indicating high transcriptomic similarity between these cells in the circulation and the TME (Fig. 1B). Transcriptomic differences between HPV+ vs. HPV⁻ leukocytes have been thoroughly investigated in our previous study[9].

**Classification of the HNSCC TME by inflammation status.** The tumor cohort was segregated into lesions with low, medium, or high lymphocyte infiltration based on the score generated from Hematoxylin and Eosin (H&E) staining of all lymphocytes (Supplementary Fig. 2B and Fig. 1D, E). There was no clear association between HPV status and inflammation score; one HPV+ lesion was in the low inflammation group, two were in the medium, and three were in the high group (Fig. 1D).

**Qualitative transcriptomic heterogeneity in CD8+ T from lesions with low and high inflammation scores.** Given the central role of CD8+ T cells in antitumor immunity[19–21], we focused on further characterization of this cell type. We extracted, subclustered, and visualized the transcriptomic profiles of 27,013 cells identified as CD8+ T lymphocytes (Fig. 2). Clinical and pathogenic information of tissues from which these cells were extracted are displayed in Fig. 2A[21,22]. Analogous to previous studies in other cancer types[20], we identified four major subtypes of T cells based on marker expression: naive-like T cells (Cluster 3, 5, 7, 8: *SELL*+, *IL7R*high, immune checkpoint receptor (ICR)negative/low), cytotoxic T cells [Cluster 1, 4, 5, 11, 12: *GNLY*high, *KLRG1*+, ICRlow/intermediate] and dysfunctional/exhausted cells (ICRhigh) that are further sub-divided into pre-dysfunctional (Cluster 1, 9, 10: *GZMK*high, *CXCL13*low, *LYAR*+) and terminally dysfunctional cells (Cluster 0, 2: *GZMK*low, *CXCL13*high, *ENTPD1*+) (Supplementary Fig. 3A, B). We also identified two clusters of cycling CD8+ T cells (clusters 6 and 13), expressing Ki67 (gene *MKI67*) and genes associated with the formation of the mitotic spindle apparatus (*TUBA1B*, *STMN1*) (Fig. 2A and Supplementary Fig. 3A). Interestingly, other effector molecules and cytokines were differentially expressed, with *GZMB* and *IFNG* having the highest expression in the most dysfunctional subsets.

After excluding circulating cells (PBL) from the analysis (Supplementary Fig. 3C), we explored the qualitative differences between CD8+ cells in TMEs with low and high immune infiltration scores. Gene set enrichment analysis on the differentially expressed genes (DEGs) was performed by inflammation score (low vs. high and vice versa). Significant enrichment of allograft rejection, IFN-γ and IFN-α response gene sets ($-\log(10)p$-value = 7.4, 7.4 and 6.7, respectively) was evident in CD8+ T cells from highly infiltrated tumors, while CD8+ T cells from non-inflamed tumors showed gene sets associated with TNFα signaling, hypoxia, inflammatory response, and apoptosis ($-\log(10)p$-value = 29.5, 10.0, 8.7 and 6.7, respectively, Fig. 2B). Examples of genes that were shared by the apoptosis or the hypoxia gene set were *GADD45B* (encodes Growth arrest and

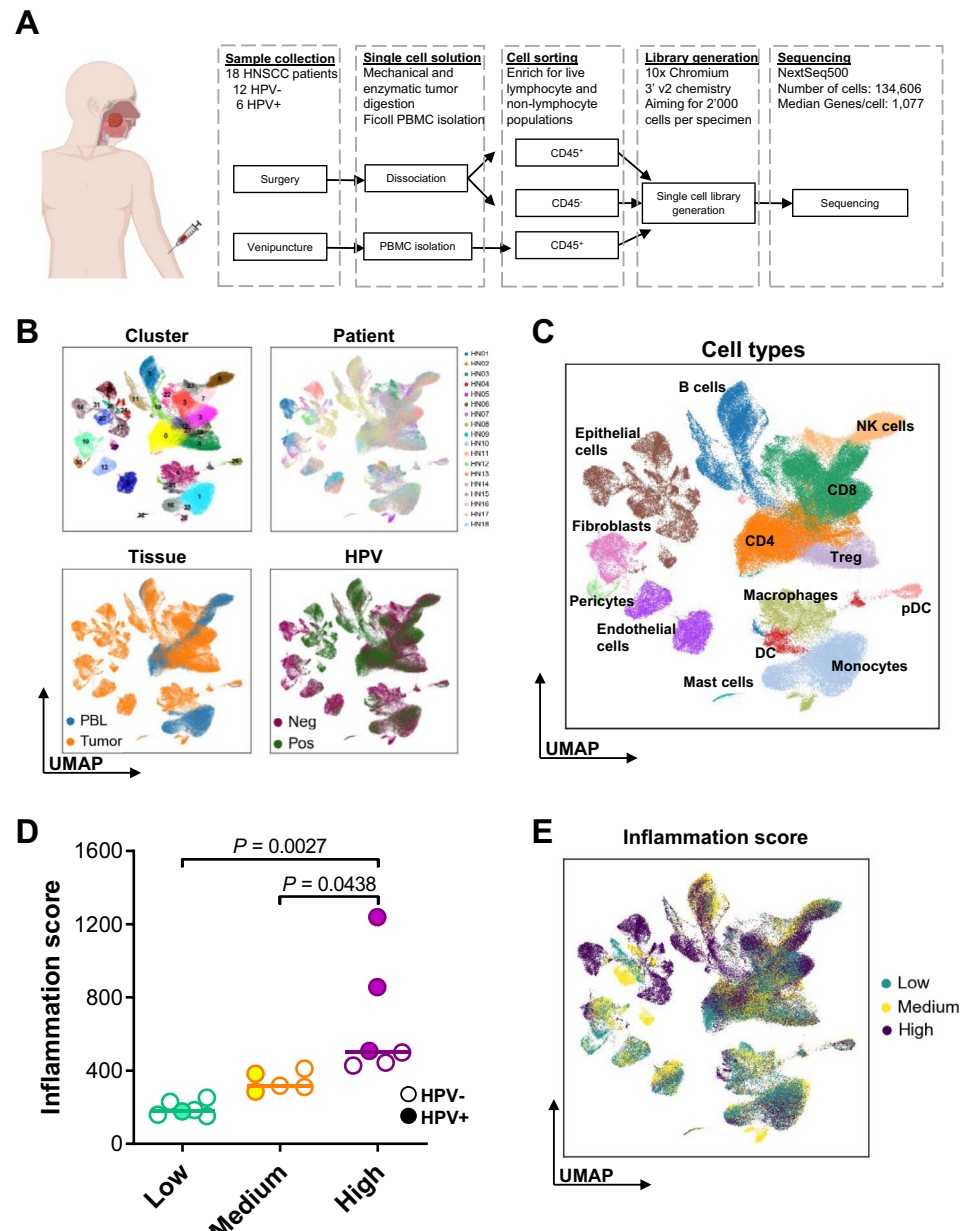

**Fig. 1 Workflow and cohort overview with cell-type identification, naming, and inflammation scoring. A** Fresh tumor and blood samples from HNSCC patients ($n = 18$ patients) were collected, dissociated, sorted, processed using a 10× Chromium controller, and then sequenced. Patient image was created using BioRender.com **B** UMAP dimensionality reduction of the total cohort of 134,606 cells was performed, based on visualization of relevant characteristics: patient contribution (HN01–HN18), cell clusters (0–33), tissue of origin (PBL vs. tumor), and viral-status (HPV+ vs. HPV−). **C** UMAP plot showing identified cell types [B cells, CD4+ T cells, CD8+ T cells, dendritic cells (DC), plasmacytoid DC, endothelial cells, epithelial cells, fibroblasts, pericytes, macrophages, monocytes, natural killer (NK) cells, T regulatory (Treg) cells, mast cells]. **D** Inflammation scores were determined by quantifying the total leukocyte infiltrate in each tumor based on H&E staining ($n = 17$ patients). Tertiles were used to segregate patients into three groups based on their inflammation score: low ($n = 6$ patients), medium ($n = 5$ patients), and high ($n = 6$ patients). Patients with HPV+ and HPV− etiologies are indicated. Center lines represent median values for each cohort. Inflammation scores were evaluated using a one-way ANOVA test. **E** UMAP plot showing cell contributions based on inflammation scores.

DNA-damage-inducible, beta), *BTG2* (encodes BTG anti-proliferation factor 2), *JUN* (encodes Jun proto-oncogene), and *FOS* (encodes Fos proto-oncogene). These genes were also enriched in the TNF-signaling pathway. This suggests a qualitative difference of CD8+ T cells depending on the inflammation status of the TME, with cells from tumors with high inflammation scores being more activated and effector-like, while cells from tumors with low inflammation score are stressed by the surrounding TME and may become (pre)-apoptotic.

**Identification and characterization of HPV+ and HPV− cancer cells.** Unlike previous scRNAseq analyses of HNSCC[9,10], this study extensively compares the immune as well as cancer/stromal cell landscapes of two contrasting, HPV− and HPV+ tumor etiologies, the latter of which have better clinical outcomes[23]. We identified and subclustered 14,920 cells of epithelial origin (Fig. 3A). The cancer cell phenotype was determined by copy number variation (CNV, Supplementary Fig. 4A) analysis and keratin expression (Supplementary Fig. 4B), as utilized

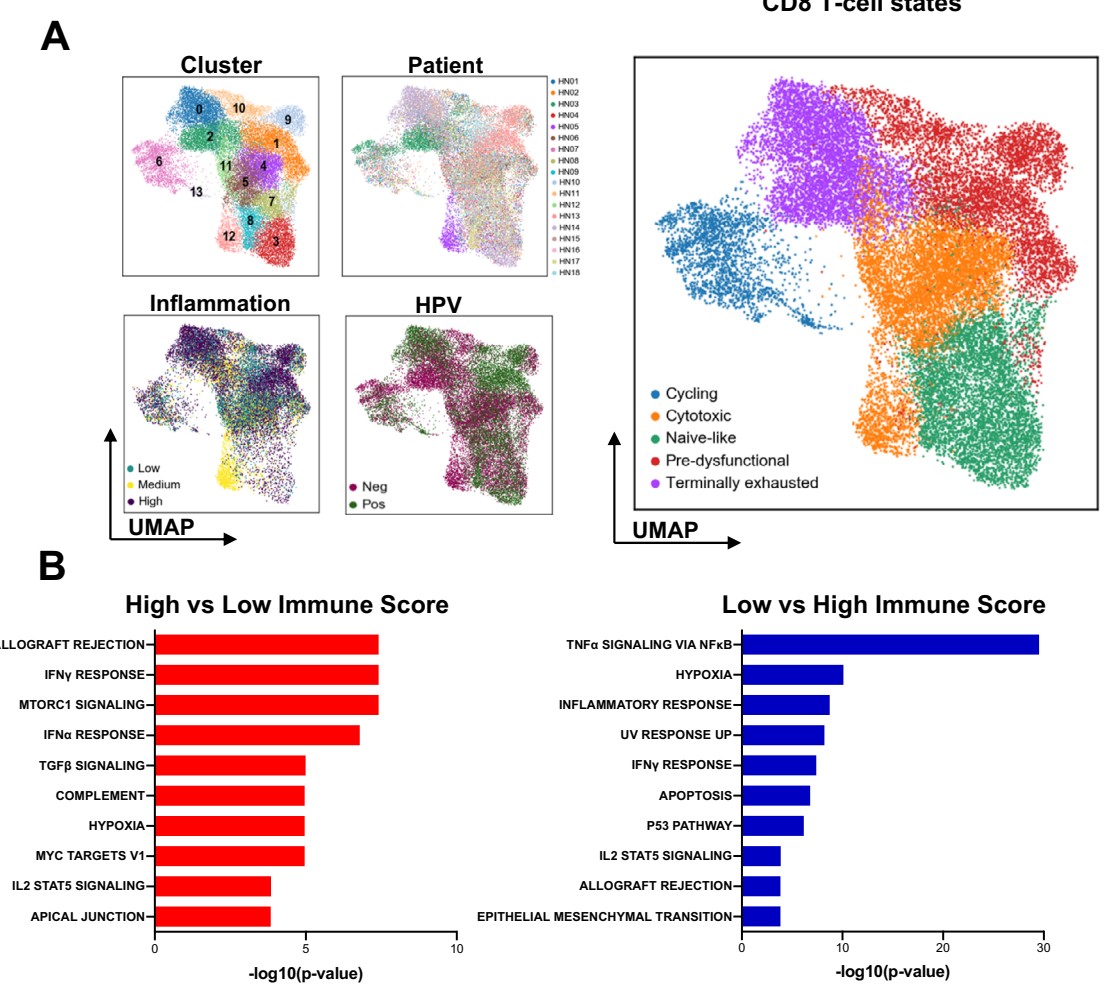

**Fig. 2 Detailed characterization of CD8⁺ T cells and intratumoral CD8⁺ cell differences in tumors with low and high inflammation scores. A** Sub-clustering of 27,013 CD8⁺ T cells ($n = 18$ patients; cluster 0–13) and visualization of relevant characteristics: patient contribution (HN01–HN18), viral-status (HPV⁺ vs. HPV⁻), inflammation score (low, medium, high; cells from the PBL, as well as all cells from HN03 were excluded) and CD8⁺ T cell sub-states (cycling, cytotoxic, naive-like, pre-dysfunctional, terminally exhausted). **B** Hallmark gene sets enriched (using the Computer Overlaps tool from MSigDB) in the top 100 DEGs between CD8⁺ T cells from high vs. low inflammation patients (and vice versa, all $p < 0.01$). $p$-values for enrichment were extrapolated from hypergeometric distribution.

previously[10]. Based on the CNV profile, clusters 13, 14, and 16 contained normal epithelial cells. Substantial patient-specific clustering was observed for malignant epithelial cells as previously described (Fig. 3A)[10]. As the UMAP visualization preserves spatial information[16], the adjacent clustering of HPV⁻ (HN01, HN07) and HPV⁺ (HN12, HN13, HN14, HN16, and HN17) might be suggestive of transcriptomic similarity. To further support this observation, we performed a hierarchical clustering analysis (Supplementary Fig. 4C) that showed that HPV⁻ and HPV⁺ clusters were allocated to separate branches of the hierarchical tree, thus supporting the notion of transcriptomic similarity across different tumors of the same etiology.

Differences in epithelial tumor cell gene expression levels based on inflammation score and etiology were detected by gene set enrichment analysis of the top 100 DEGs defining each group (Fig. 3B). Cancer cells from lesions with high infiltration showed significant enrichment of epithelial-to-mesenchymal transition (EMT), MYC targets, IFNγ response, mitotic spindle formation, and allograft rejection gene programs ($-\log(10)p$-value 4.8, 4.8, 3.7, and 3.7, respectively), while cells from a micro-milieu with low infiltration displayed significantly enriched P53 pathway, fatty acid metabolism, glycolysis and hypoxia gene sets ($-\log(10)$

$p$-value 8.5, 4,2, 3.7, and 3.7, respectively). When directly comparing HPV⁺ and HPV⁻ cancer cells, oxidative phosphorylation, TNFα signaling, as well as pathways for early and late estrogen response were significantly enriched in HPV⁺ tumors ($-\log(10)p$-value 11.3, 7.2, 7,2 and 6.0, respectively)[24], while EMT, IFNγ and IFNα responses were significantly enriched in HPV⁻ cancers ($-\log(10)p$-value 32.5, 23.7 and 18.3, respectively)[10].

To further leverage this unique data set containing HPV⁺ cancer cells, HPV16 encoded genes were added to the human genome reference file and their expression was quantified and visualized in epithelial cell transcriptomes. The viral genome was exclusively expressed by HPV⁺ HNSCC (Fig. 3C), defined by clinical p16 IHC testing[25]. Variable expression of HPV genes was detected in these p16⁺ oropharyngeal cancers, with *E1*, *E5*, and *E7* being the most and *L1* and *L2* being the least widely observed. To validate these findings, qPCR was implemented to quantify the relative amount of HPV16 *L1*, *L2*, *E6*, and *E7* transcripts in four HPV⁺ patients from our cohort. *L1/L2* were generally expressed at lower levels than *E6/E7* supporting our scRNAseq observations (Supplementary Fig. 5A). The substantial inter- and intra-patient variability of viral gene expression was further

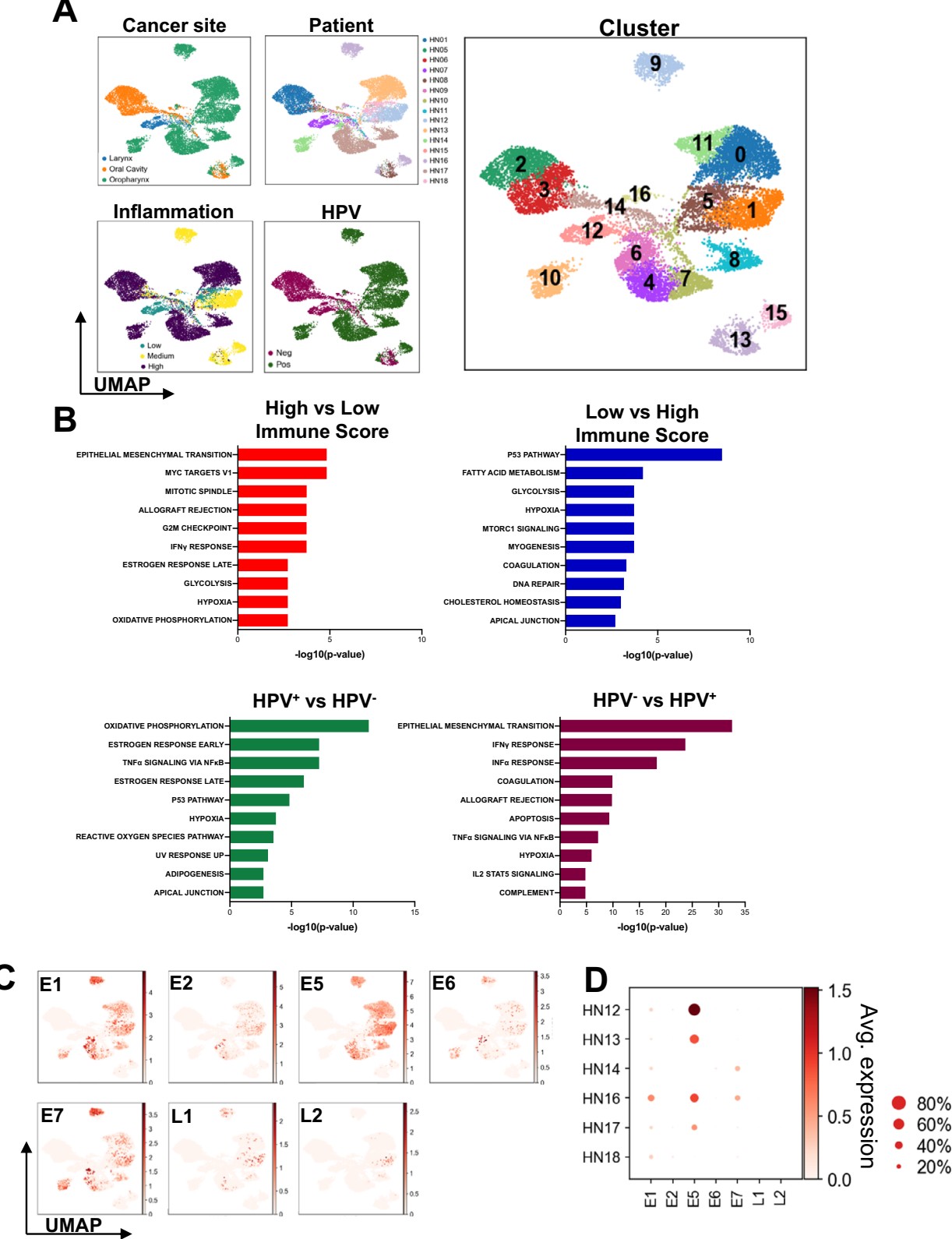

**Fig. 3 Squamous epithelial cancer cells and HPV gene expression. A** Sub-clustering of 14,920 epithelial cells ($n = 15$ patients; cluster 0–16) and visualization of relevant characteristics: patient contribution (HN01, HN05–HN18), viral-status (HPV+ vs. HPV−), cancer site (larynx, oral cavity, oropharynx) and inflammation score (low, medium, high). **B** Hallmark gene sets enriched (using the Computer Overlaps tool from MSigDB) in epithelial cells derived from tumors with low vs. high inflammation score, as well as HPV+ vs. HPV− lesions (and vice versa, all $p < 0.0001$). $p$-values for enrichment were extrapolated from hypergeometric distribution. **C** UMAP visualization of transcripts of HPV16 genes (*E1, E2, E5, E6, E7, L1,* and *L2*) in epithelial cells. **D** HPV+ patient-specific intensity and prevalence of HPV16 gene expression. Color bar indicates normalized gene expression.

visualized using a dot-plot (Fig. 3D). To validate the heterogeneity in HPV gene expression among malignant cells and to explore whether highly expressing viral oncoproteins localize to a particular area of the tumor, we performed in situ hybridization for high-risk HPV *E6/E7* mRNA (Supplementary Fig. 5B). We confirmed that only a subset of malignant cells expresses *E6/E7*. Spatially, we observed a stratified arrangement of *E6/E7* expression, with these oncogenes being most expressed by cells of the basal epithelial layer and decreasing towards the cystic lumen.

**The stromal compartment of the HNSCC TME consist of diverse cell types**. Among non-epithelial non-immune cells, we observed that endothelial cells, fibroblasts, and pericytes formed distinct clusters (Fig. 4A, B). No distinct patient- or HPV status-associated clustering was observed. Nearly all patients contributed to the pericyte cluster, with patients HN07, HN09, and HN11 contributing the most cells (Fig. 4C). Pericytes (cluster 7) clearly separated from other stromal cells. Hallmark genes of activated, pro-tumor type-2 pericytes, such as *ACTA2* (encodes α-smooth muscle actin) and *RGS5* (encodes regulator of G-protein signaling-5) were observed (Supplementary Fig. 5C)[26]. The origin, identification, and pathogenetic role of pericytes implicates them in hallmark processes of cancer development and progression such as neo-angiogenesis and leukocyte recruitment[27].

**Fibroblasts in HNSCC include normal/activated, cancer-associated, and elastic sub-states**. The tumor stroma consists of a heterogeneous group of mesenchymal cells manifesting a high degree of plasticity and multipotency. Fibroblasts ($n = 4034$) were segregated into 9 clusters (Fig. 5A), assigned to 3 different cell sub-states based on the highest expressing genes (Fig. 5B) and differentially expressed hallmark gene sets (Fig. 5C and Supplementary Figs. 6 and 7).

Clusters 0, 1, 2, 3, 4, 5, and 6 show expression of classical cancer-associated fibroblast (CAF) markers (*FAP*, *PDGFRA*, *LOX*, and metalloproteinases, Fig. 5B and Supplementary Fig. 6A). Normal/activated fibroblasts (NAF, clusters 4 and 6) showed a low expression of CAF markers (Supplementary Fig. 6A). One sub-state previously not described in HNSCC TME scRNAseq studies were fibroblasts expressing elastic fiber differentiation genes (cluster 7; Supplementary Fig. 6B and Supplementary Data 1), with increased expression of tropoelastin (*ELN*), fibrillin-1 (*FBLN1*), and Microfibril Associated Protein 4 (*MFAP4*)[28]. The presence of this fibroblast sub-state was confirmed in the bulk RNAseq data from the HNSCC TCGA cohort using CIBERSORTx[29] for deconvolution (Supplementary Fig. 6C), where 199/500 patient samples were predicted to have non-zero fractions of fibroblasts with elastic differentiation.

A comparative analysis of hallmark gene sets was performed (Fig. 5C), showing moderate heterogeneity between the clusters: NAF showed an intermediate enrichment of hallmark signaling pathways (e.g., protein secretion, DNA repair, and G2M checkpoint), while CAF were shown to be highly active. Fibroblasts with elastic differentiation on the contrary show no enrichment of most hallmark signaling pathways. Two of the CAF clusters were dominated by cells from one patient: cluster 2 by HN07 and cluster 0 by HN01, while CAF clusters 1 and 2 contained cells from multiple patients (Fig. 5A). These differences were underscored by divergent expression of metalloproteinases (Supplementary Fig. 6A) that are the main effector molecules of CAF and promote tumor invasion and metastasis: Cluster 2 mainly expressed *MMP1*, while cluster 1 expressed *MMP11* (Supplementary Fig. 6A)[30].

Functional differences were further explored using GO BP gene sets from the MSigDB (Supplementary Fig. 7A). Here, we show that in CAF, there is an enrichment of pathways of extracellular structure organization and collagen fibril organization, while elastic fibroblasts showed enrichment of pathways associated with secretion, adhesion, and cell proliferation. To corroborate these findings, we performed IHC staining on tumor sections from 6 patients from our scRNAseq cohort using MFAP4 as a marker for elastic type differentiation[31]. MFAP4 was expressed in spindle cells of the tumor stroma, but not in the tumor or normal epithelial cells (Supplementary Fig. 7B and Supplementary Table 2).

Given the impact of CAF on patient survival[30], we explored the potential prognostic value of fibroblasts with elastic differentiation in HNSCC using bulk RNA sequencing data from The Cancer Genome Atlas (TCGA) database. A negative prognostic impact was observed using the CAF signature scores in HPV⁺, but not in HPV⁻ samples (Supplementary Fig. 8A). Interestingly, the elastic fibroblast signature score also showed the same pattern in HPV⁺ samples (Supplementary Fig. 8B). We found that HPV⁺ patients with both low elastic fibroblast and CAF signature scores showed the best overall survival ($p = 0.0013$, Fig. 5D).

**Endothelial cells separate into two distinct cell types**. The tumor vasculature has been recognized as being organ- and disease-specific, and is a main determinant of the intratumoral immune landscape, due to its impact on cell extravasation and homing[32]. scRNAseq analysis of 7431 cells expressing endothelial markers showed a clear separation of cells into lymphatic and vascular endothelia (Fig. 6A) that could be further sub-divided based on the expression of specific genes (Fig. 6B) and gene set enrichment analysis (Fig. 6C). Except for cluster 1 (HN09), all clusters had contributions from different patients demonstrating heterogeneity, whereas no clear enrichment of endothelia based on HPV status was observed. The lymphatic endothelium was assorted into four clusters, with clusters 0 and 1 representing activated cells with increased inflammatory response and signaling gene sets (Fig. 6C). Interestingly, cells from pre- and post-capillary endothelial cells did not cluster together, but rather aggregated depending on activation status, with cluster 3 showing an enrichment of inflammatory response gene sets (Fig. 6D and Supplementary Fig. 7C)[33].

**Overview of putative ICR–immune checkpoint ligand (ICL) interactions between CD8⁺ T cell and non-immune or antigen presenting (APC) cells**. Our scRNAseq data set demonstrates the complexity of inflamed and non-inflamed HNSCC tumors and stresses the unique roles that non-immune cell-mediated interactions may play in the regulation of tumor inflammation status. It also allows for inference of putative receptor–ligand interactions between different cell types within the TME. Along those lines, we assessed the ICR/ICL landscape of HNSCC. Relative (Fig. 7A) and scaled (Supplementary Fig. 9A) expression levels of clinically targetable ICLs (binding PD-1, TIM-3, LAG3, and TIGIT) were quantified in all the major cell types detected. The main contributors of PD-L1 (Gene: *CD274*) in the TME were DC and macrophages. In contrast, highest levels of PD-L2 (Gene: *PDCD1LG2*) were detected in DC and fibroblasts. Galectin-9 (Gene: *LGALS9*) and *CEACAM1*, common TIM-3 ligands[34], showed unique distribution patterns. *LGALS9* was expressed at various levels by all the cell types, except pericytes. The highest transcript levels were observed in DC, macrophages, and endothelial cells. Low levels of *CEACAM1* were observed in epithelial and endothelial cells. Low expression levels of *FGL1* (encodes fibrinogen-like protein 1), a major inhibitory LAG3 ligand[35], were

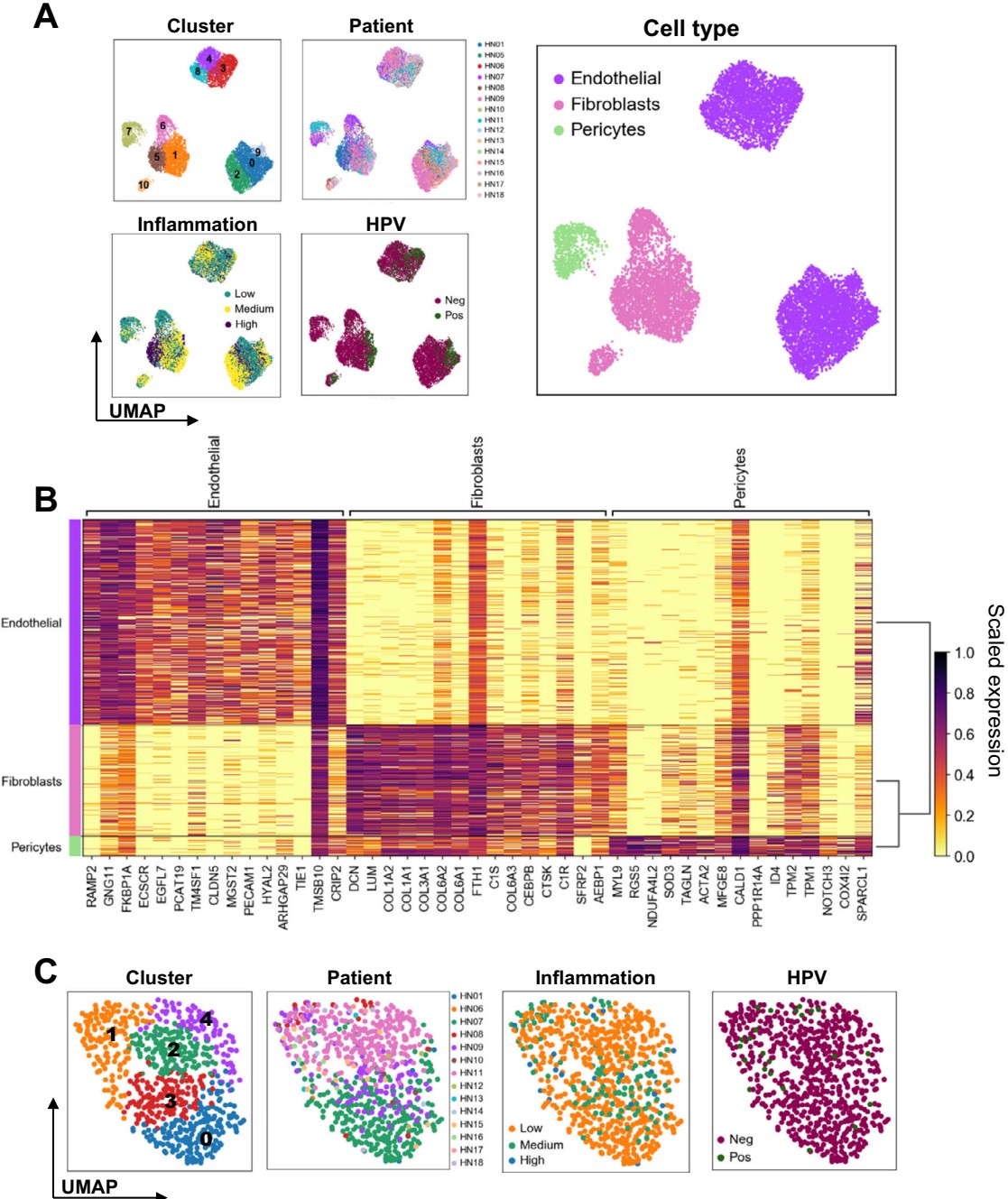

**Fig. 4 Overview of stromal cells with a focus on pericytes. A** Sub-clustering of 12,179 stromal cells (non-immune, non-epithelial cells; *n* = 15 patients; cluster 0–10) and visualization of relevant characteristics: patient contribution (HN01, HN05–HN18), viral-status (HPV+ vs. HPV−), cell type (endothelial, fibroblasts, pericytes) and inflammation score (low, medium, high). **B** Heatmap showing top 10 genes characterizing each cluster (color scale depicts scaled gene expression). **C** Sub-clustering of pericytes (cluster 0–4) and visualization of relevant characteristics: patient contribution (HN01–HN18), viral-status (HPV+ vs. HPV−), cell type (endothelial, fibroblasts, pericytes), and inflammation status (low, medium, high).

detected in fibroblasts. High expression levels of *HLA-DRA* and *HLA-DRB1*, prototypical LAG3 ligands[35], were detected in multiple cell types, with pericytes showing the lowest expression levels (Supplementary Fig. 9B). Low levels of CD155 (Gene: *PVR*), a high-affinity TIGIT ligand[36], were primarily observed in endothelial cells. CD112 (Gene: *NECTIN2*), the low-affinity TIGIT ligand[37], was expressed by all non-immune cells, as well as DC and macrophages. The highest *NECTIN2* expression levels were observed in endothelial cells. *PDCD1*, *HAVCR2* (encodes TIM-3), *LAG3*, and *TIGIT* were variably expressed on CD8+ and CD4+ T cells, NK cells, and Treg, with *LAG3* and *TIGIT*

expressed at the highest levels. *HAVCR2*, *LAG3*, and *TIGIT* were also expressed by macrophages, while *HAVCR2* and *LAG3* were also found in DC. While a recent publication has indicated that targeting of PD-1 on myeloid cells can induce antitumor immunity in mice[38], transcriptomically 1–2% of macrophages express low levels of *PDCD1* from our patient cohort. In tumors with high inflammation scores, *LAG3* expression was higher on DC and especially macrophages (Fig. 7A and Supplementary Fig. 9C). These data indicate that tumor inflammation status may associate with unique immune checkpoint signatures that could be used to tailor therapeutic strategies.

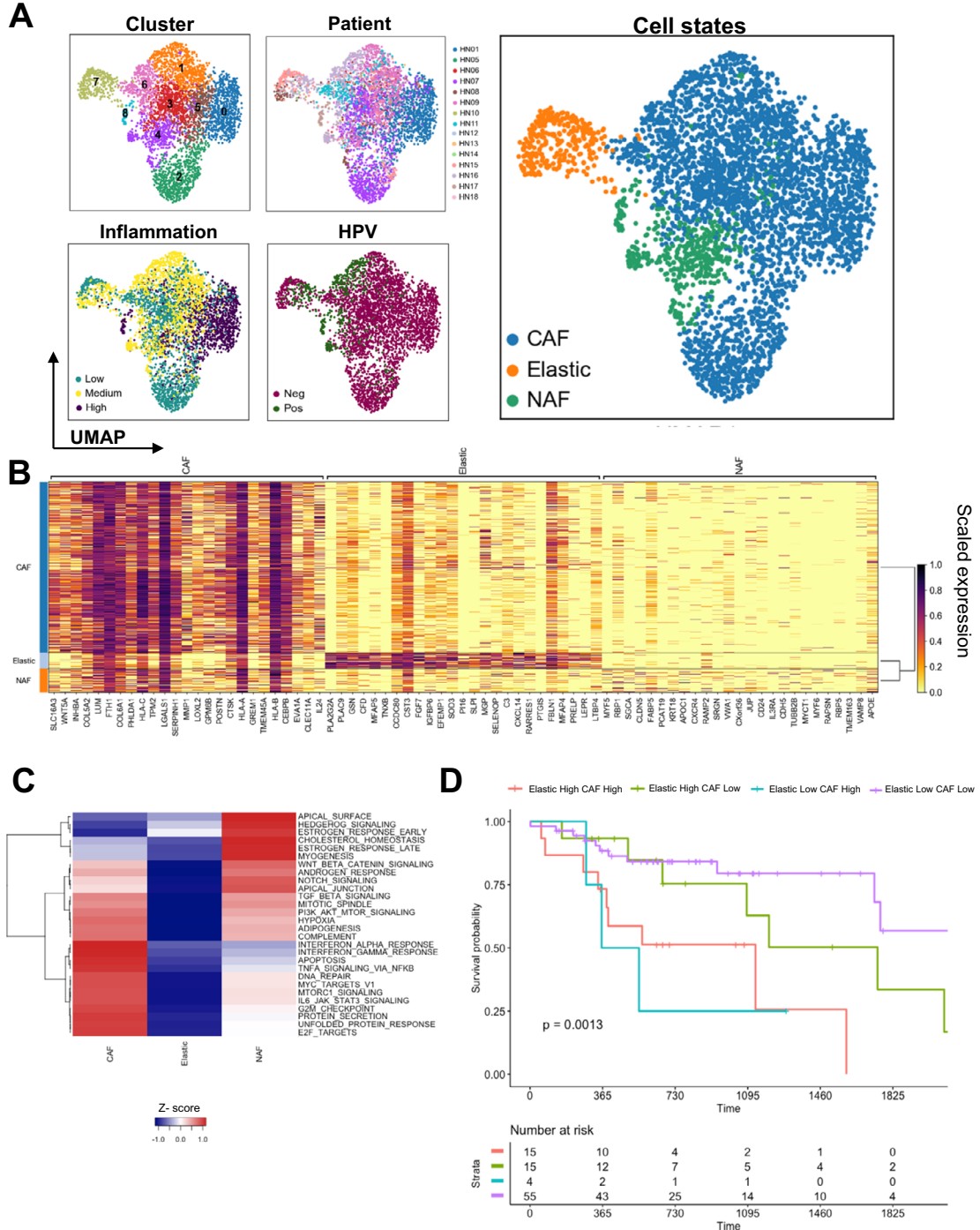

**Fig. 5 Fibroblast sub-states, DEG, and impact on survival. A** Sub-clustering of 4034 fibroblasts ($n = 15$ patients; cluster 0–7) and visualization of relevant sample characteristics: patient contribution (HN01, HN05–HN18), HPV status (HPV$^+$ vs. HPV$^-$), inflammation score (low, medium, high), and sub-states (cancer-associated fibroblasts, normal activated fibroblasts, and fibroblasts with elastic differentiation). **B** Heatmap showing top 25 genes characterizing each cluster (color scale depicts scaled gene expression). **C** Enriched hallmark gene sets between clusters based on results of gene set enrichment analysis ("singleseqgset" package). **D** Overall survival analysis of HPV$^+$ patients from the TCGA HNSCC bulk RNAseq cohort based on gene signature scores of fibroblasts with elastic differentiation and cancer-associated fibroblasts (log rank test $p = 0.0013$).

We next explored a curated list of ICL–ICR interactions between CD8$^+$ T cells and non-immune or myeloid APC in tumors with low and high inflammation scores (Fig. 7B, C; using the CellPhoneDB package)[39]. *CD274–PDCD1* interactions appeared to be primarily mediated by macrophages and were observed in 8/12 patients regardless of their inflammation status. In only 2/10 patients (HN01 and HN18), this interaction was predicted between CD8$^+$ T cells and epithelial cells. *HAVCR2–LGALS9* interactions were predicted to be primarily mediated by macrophages (10/12 patients) and endothelial cells (9/10 patients). Of the two TIGIT ligands, *TIGIT–NECTIN2* interactions were most common and were projected to be facilitated by non-immune cells, as well as DC and macrophages.

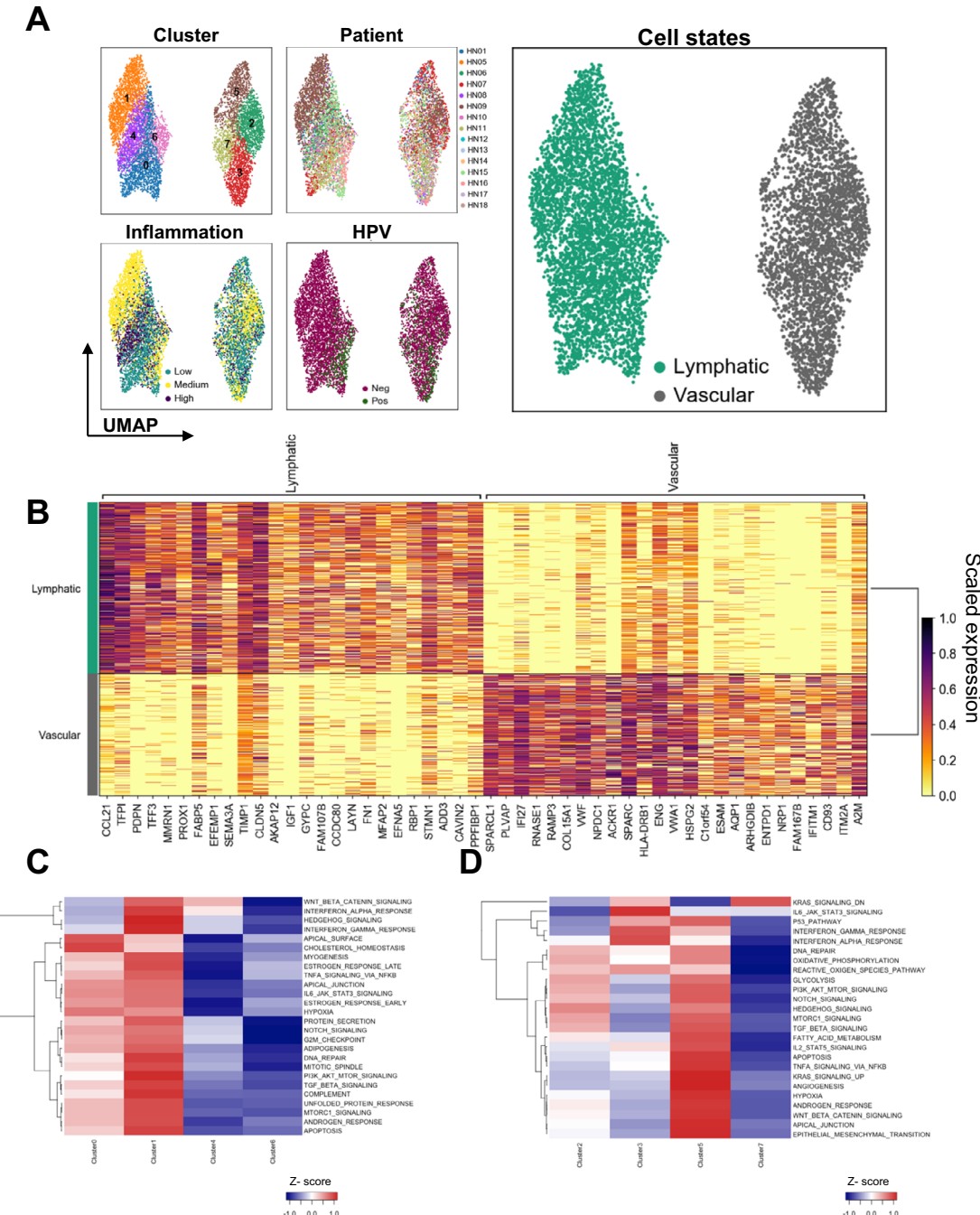

**Fig. 6 Endothelial cell subsets and DEG. A** Sub-clustering of 7431 endothelial cells ($n = 15$ patients; cluster 0–7) and visualization of relevant sample characteristics: patient contribution (HN01, HN05–HN18), HPV status (HPV$^+$ vs. HPV$^-$), inflammation score (low, medium, high) and types of endothelial cells (lymphatic and vascular). **B** Heatmap showing top 25 genes characterizing each cluster (color scale depicts scaled gene expression). Enriched hallmark gene sets between **C** lymphatic and **D** vascular endothelial cell clusters based on results of gene set enrichment analysis ("singleseqgset" package) are shown.

Due to the clinical relevance of PD-1[4], TIM-3[40], and LAG3-targeting[41] therapies currently being used or tested, we validated the PD-L1, galectin-9, and HLA-DR expression at the protein level and corroborated that myeloid population are the major contributors to the ICL expression in the HNSCC TME. Using multicolor flow cytometry (Supplementary Fig. 10) from 7 additional patient specimens (3 HPV$^-$, 4 HPV$^+$, Supplementary Table 1) macrophages (CD45$^+$Lin$^-$CD14$^+$HLA-DR$^{high}$) were identified as the primary PD-L1 contributors in the TME, with DC1 (CD45$^+$Lin$^-$CD14$^-$HLA-DR$^+$CD141$^+$), DC2 (CD45$^+$Lin$^-$CD14$^-$HLA-DR$^+$CD1c$^+$), fibroblasts (CD45$^-$CD90$^+$) and

endothelial cells (CD45$^-$CD90$^-$CD141$^+$) also contributing to the PD-L1 pool (Fig. 8A). Epithelial/tumor cells (CD45$^-$CD90$^-$CD141$^-$) expressed surprisingly low amounts of PD-L1. Galectin-9 was broadly expressed by a variety of myeloid subpopulations, and was especially prominent on non-immune cells, with fibroblasts expressing the highest galectin-9 levels (Supplementary Fig. 11A). Besides being expressed by macrophages, DC1, and DC2 (Supplementary Fig. 10), HLA-DR was commonly detected on non-immune cells, particularly fibroblasts and endothelial cells (Supplementary Fig. 11B).

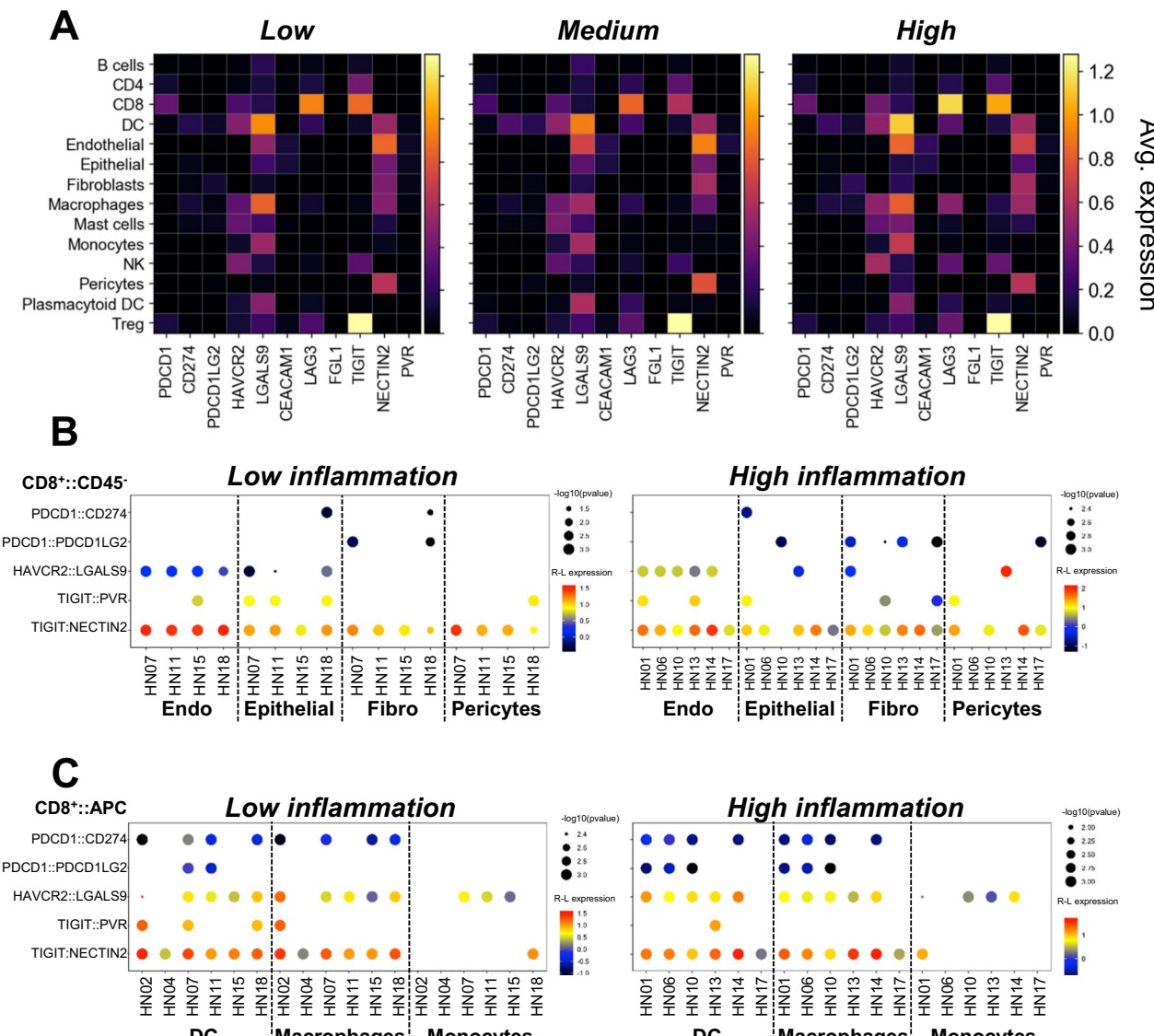

**Fig. 7 Crosstalk between various cellular constituents of the TME evaluated by potential ICR–ICL interactions. A** Average expression of immune checkpoint receptors and corresponding ligands on all cell types identified in tumors with low, medium, and high inflammation scores is summarized ($n = 17$ patients). CellPhoneDB package was used to predict patient-specific ICR–ICL interactions between CD8+ T cells and **B** CD45− endothelial cells (Endo), epithelial cells, fibroblasts (Fibro), and pericytes or **C** myeloid APC (DC, macrophages, and monocytes) in tumors with low and high inflammation scores. The colored scale represents the log2 mean expression of receptor–ligand pairs.

**PD-L1+ macrophages spatially associate with CD8+ T cells in the HNSCC TME.** As an example of immune checkpoint receptor:ligand pairs with known clinical implications, our scRNAseq and flow cytometry data strongly suggest that macrophages are a major source of PD-L1 within the HNSCC TME. A drawback of these methodologies is that they do not provide spatial associations of evaluated cell types within the TME once it is physically disaggregated. Therefore, it is impossible to establish physical proximity between PD-L1+ cells and CD8+ T cells, which is a prerequisite for cell-to-cell interactions. To study spatial association between the various cell types in the TME, sections of tumors were stained for DNA (DAPI), CD3, CD8, CD68, PD-L1, and pan-CK (Supplementary Table 3 and Fig. 8B) and evaluated by multispectral fluorescent microscopy. Each region of interest (ROI) evaluated was sub-divided into tumor and stromal regions to establish whether PD-L1 expression patterns and cell-to-cell associations are dictated by tumor geography. When comparing PD-L1+ cells, average PD-L1

expression intensity was higher on macrophages (CD68+) than on tumor cells (pan-CK+) in both the tumor bed and stromal regions for all patients evaluated (Fig. 8C, D). PD-L1 intensity was, on average, 19.8% and 18.7% higher on macrophages than on tumor cells in tumor bed and stroma, respectively (Fig. 8C), confirming our flow cytometric observations.

Next, we explored average distances of PD-L1+ macrophages or PD-L1+ tumor cells to CD8+ (Fig. 8E), CD8− and bulk CD3+ T cells (Supplementary Fig. 12) as surrogate biomarkers of cell-to-cell interactions. PD-L1+ macrophages were commonly found in tumor and, especially, stromal regions abundant in T cells. In 4/5 patients tested, PD-L1+ macrophages closely associated with CD8+ and CD8− T cells in both tumor (median distance across all patients of 19.6 and 21.5 μm, respectively) and stromal regions (median distance across all patients of 13.2 and 16.9 μm, respectively; Fig. 8E; Supplementary Fig. 12B). Both CD8+ and CD8− T cells were also detected in close proximity to tumor cells in both tumor (median distance across all patients of 10.5 and

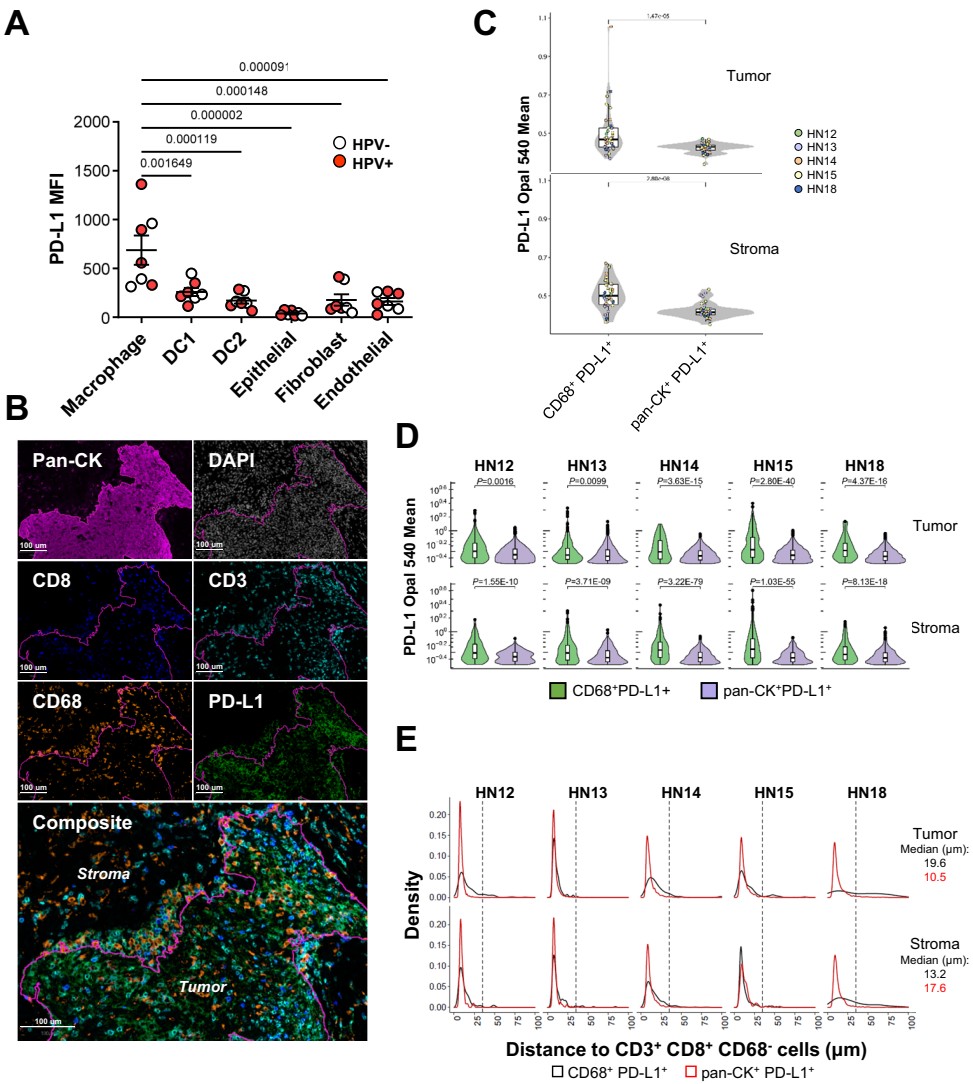

**Fig. 8 Macrophages are major contributors of PD-L1 to CD8+ T cells in HNSCC. A** Mean fluorescence intensity (MFI) of PD-L1 on immune (macrophages, DC1, and DC2) and non-immune (fibroblasts, epithelial, and endothelial) cell types within the HNSCC was evaluated by flow cytometry ($n = 7$ patients) as depicted in Supplementary Fig. 10. Datapoints represent individual patients. Center lines represent mean values and whiskers depict standard errors of means. *p*-values were calculated using the one-way ANOVA test. **B–E** Multispectral immunofluorescence (mIF) staining was performed on tumor sections obtained from patients HN12, HN13, HN14, HN15, and HN18 using the conditions described in Supplementary Table 4. Three or more high-resolution images of regions of interest (ROIs) that contained a balance of tumoral (tumor bed) and peritumoral/stromal regions were acquired from each tumor section. ROIs were selected based on H&E staining in addition to mIF whole slide scans. **B** A representative ROI selected from patient HN12 is shown. Individual channels for pan-CK, DAPI, CD8, CD3, CD68, and PD-L1 are presented, as well as the composite image. Tumor bed is depicted by the cyan-colored border line in the composite image. **C** Intensity of PD-L1 expression on CD68+PD-L1+ macrophages and pan-CK+ PD-L1+ tumor cells pooled from all patient-associated ROIs is shown ($n = 5$ patients). Datapoints represent 41 individual ROIs colored by patient id. **D** Patient-specific pooled PD-L1 expression levels are shown. 43,517 cells from the 41 ROIs in **C** were included in the analysis. **E** Measured distance to CD3+CD8+CD68− T cells from CD68+PD-L1+ macrophages or pan-CK+PD-L1+ tumor cells pooled from all patient-associated ROIs is shown. Dashed lines represent the 35 μm distance used as the cutoff to measure cell-to-cell interactions between evaluated cell types. This distance represents 1–2 cell diameter distance between macrophages (20–30 μm in diameter) and lymphocytes (5–7 μm in diameter). Calculated median distances across all patients and between evaluated cell types are shown. For boxplots, center lines represent median values, box limits represent upper and lower quartiles and whiskers represent 1.5× interquartile range. One-way ANOVA test was used in **A**. Linear mixed-effects models were used in **C** and **D**, with cell group as a fixed effect and individual patient as a random effect. BH-FDR method was used for multiple comparison adjustment. All tests are two-sided.

10.7 μm, respectively) and stromal (median distance across all patients of 17.6 and 20.0 μm, respectively) regions of all patients. The latter observation can be attributed to T cells aggregating around sporadic invasive tumor cells that were detected in the stroma (Fig. 8B). Close associations between PD-L1+ tumor cells and various T cell subsets within the tumor are due to the ubiquitous presence of tumor cells while macrophages are more heterogeneously distributed within the TME. Such studies

demonstrate how other potentially relevant receptor:ligand pairs can be identified to guide targeted therapeutic approaches using scRNAseq datasets.

## Discussion

The success of immuno-oncology therapies is dictated by numerous resistance mechanisms mediated by multifaceted

interactions between stromal, epithelial, and immune cells within the TME[7]. In order to develop more successful therapies, it is necessary to decipher these complex intercellular interactions. Novel technologies such as scRNAseq allow for a multi-dimensional analysis of tumor heterogeneity. Our goal was to generate the most comprehensive scRNAseq data set and profiling of immune and non-immune cells in the TME of HPV+ and HPV− HNSCC to date. Our study leverages the breadth and depth of a large data set for a detailed analysis that permits identification and sub-clustering of major cell subsets of the TME, exploration of cell type-, inflammation- and etiology-specific gene signatures and prediction of putative cell–cell interactions, for future translational applications and subsequent correlation with on-treatment specimen analyses.

CD8+ T cells are one of the main effector cell types that mediate anti-cancer immunity and the target of many currently emerging immunotherapy drugs, making them the focus of numerous single-cell studies[42–45]. Building on these studies, we were able to recapitulate the heterogeneity of CD8+ T cells (naive-like, cytotoxic, pre-dysfunctional, and dysfunctional cell states) as well as to show the differential expression of GZMK and CXCL13 transcripts in the continuum of dysfunctional CD8+ T cells. However, as previously discussed[20], even though scRNAseq gives a very detailed view of heterogeneous populations, dissecting exact cell states along a continuum is challenging for bioinformatical as well as biological reasons. It is especially interesting that terminally exhausted T cells seem to express the B-cell attracting chemokine CXCL13, suggesting that these cells may play a role in the development of mature tertiary lymphoid structures[46], which have been linked to better survival in head and neck cancer[47]. Previously we have reported that PD-1high CD8+ T cells contained the highest expression of granzyme B[19], which we were able to confirm transcriptionally here.

Clinical significance of the density and quality of immune cells in the (inflamed vs. non-inflamed) TME has been highlighted in many tumor types[48], with the caveat of widely differing descriptions of immune infiltration and methods that define it[21,22]. Thus far, the connection between quantitative and qualitative differences has not been evaluated at a single-cell level. We show that inflamed tumors not only have higher proportions of CD8+ T but also have a more effector/differentiated transcriptomic signature and a higher number of putative cell–cell interactions. This is in line with the current understanding that HNSCC tumors infiltrated by high amounts of CD8+ T cells have a better prognosis possibly due to improved effector function and tumor cell killing[49].

Since HPV+ HNSCC has overtaken cervical cancer as the most common HPV-associated cancer in the US[10,23,50], pathophysiological distinction and clarification become especially relevant[5]. We demonstrate that virally encoded genes can be mapped and quantified in the human scRNAseq data set, allowing for direct evaluation of HPV gene expression in individual HPV-transformed epithelial tumor cells, permitting transcriptomic comparisons and virus-specific targeting. Our findings support the current thinking on the pattern of HPV gene expression in cancer: the viral genome persists as shown in the expression in early genome maintenance genes (E1, E2, E5) and high-risk viral oncogenes (E6 and E7), but no virus is assembled and released (due to lack of L1 and L2 capsid protein expression)[51]. Further, the analysis of viral gene expression patterns highlights the heterogeneity of viral transcripts in HPV+ cancers at a cellular and an interpatient level, which may have potential therapeutic implications when targeting these transcripts[52,53].

Our cohort extends previous single-cell studies in HNSCC that also showed a partial-EMT signature for HPV− cancers[10], but which did not consider HPV+ disease. The estrogen response

signature we found in HPV+ lesions corroborates previous research in cervical cancer showing oncogenic synergism between HPV infection and estrogen receptor (ER) associated pathways[54]. In HPV+ HNSCC, this mechanism has not yet been as clearly shown, however, the expression and prognostic impact of estrogen receptors in the HNSCC TME has been recently reported[24].

The availability of scRNAseq data from non-immune cells enabled us to further delineate fibroblast, pericyte, and endothelial cell (vascular and lymphatic origin) clusters and to expand on previous studies that were limited by lower cell numbers[10]. These non-malignant cell populations exhibit significant interpatient heterogeneity. We also identify a sub-state of elastic fibroblast differentiation not previously reported in HNSCC scRNAseq studies. Fibroblasts with elastic differentiation are an emerging phenotypic subtype, and a grade-dependent change of morphology and orientation has been proposed[55]. Considering that cancer-associated fibroblasts are generally considered to be immunosuppressive, having a better resolution of their heterogeneity in the TME could open the way for new biomarkers, risk-stratification, and novel therapeutic targets[23]. Our observation that fibroblasts with elastic type differentiation have a negative prognostic value specifically for HPV+ patients stresses the importance of seeing HPV+ and HPV− disease as two different diseases that require tailored, specific biomarkers and treatment approaches.

This data set also allowed us to explore cell type-dependent differences in the interactome using algorithms that predict putative cell–cell interactions within the TME. CD8+ T cells in the HNSCC TME interactome, particularly with macrophages, are based on predicted CD274–PDCD1, HAVCR2–LGALS9, and TIGIT–NECTIN2 interactions that may dictate tumor rejection. This observation is supported by our current findings and corroborated by a previous study that showed that T cells readily co-localize with PD-L1+ macrophages in inflamed HNSCC lesions[56]. Combined scoring systems incorporating macrophages and tumor cells to determine patients' PD-L1 status appear most predictive of clinical efficacy[57]. Thus, with the approval of novel ICR-targeting therapies for the treatment of various malignancies including HNSCC[4], a better understanding of the primary cellular sources and expression patterns of associated ICL in the HNSCC TME is critical. Indeed PD-L1 and galectin-9 were previously reported to be expressed not only by tumor cells and macrophages, but also by Treg, NK cells, fibroblasts[58–61], and endothelial cells[62]. We validated by scRNAseq, flow cytometry, and multispectral microscopy that macrophages are the key contributors of PD-L1 in the HNSCC TME, paralleling clinical significance and the widespread use of the combined positive score (CPS) as a biomarker for PD-1-based immunotherapy[57]. In contrast, the galectin-9 protein was expressed primarily on fibroblasts and other non-immune cells, whereas transcriptomics suggested that macrophages were the dominant cell type. These data suggest that combined scoring of galectin-9 on non-immune cells and macrophages may also be necessary to identify patients that may benefit from ongoing TIM-3-targeting therapies. HLA-DR was expressed by both non-immune and immune cells within the TME both by scRNAseq and flow cytometry. Transcriptomic data indicate that LAG3 is differentially upregulated by DC and, particularly, macrophages from lesions with high inflammation score. Since LAG3 expression on TIL has been reported to correlate with increased CD8+ T cell infiltrate in HNSCC[63], LAG3-targeting therapies may be better suited for the treatment of patients with inflamed lesions.

In conclusion, we present a comprehensive scRNAseq investigation of HPV+ and HPV− HNSCC TME (cancer, stromal and immune cells). The breadth and depth of this data set permit for detailed sub-clustering and transcriptomic analysis of major cell

subsets and investigation of putative cell–cell interactions within the TME. We demonstrate the utility of this robust data set as a resource and that it can be reliably used to identify novel receptor or ligand expression patterns that can be validated at the protein level[14].

## Methods

**Ethical regulations**. The research presented here complies with all relevant local, national, and international regulations. For all human patient samples, informed written consent was obtained prior to donation. The University of Pittsburgh Cancer Institute Review Board (Protocol 99-069) approved the study.

**Patient cohort**. Patient characteristics are shown in Supplementary Table 1, and this cohort comprised transcriptomic profiles of $CD45^+$ PBL and TIL ($n = 18$ patients), as well as $CD45^-$ non-immune cells from patient tumors ($n = 15$ patients). Raw files and separate analyses from PBL and TIL isolated from HN01–HN15 were previously published[9]. All transcriptomic analyses of non-immune cell types and their interactions with immune cells, particularly $CD8^+$ T cells, are novel and unpublished. Data from all $CD45^-$ cells, as well as PBL and TIL from HN16-18 are also unpublished.

**Tissue dissociation and PBMC isolation**. After informed consent, fresh peripheral blood and tumor biopsies were obtained from treatment-naive HNSCC patients. After physical dissociation, tumors underwent a 30 min enzymatic digestion in a dissociation cocktail [1× HBSS supplemented with 50 IU/ml collagenase I, 25 IU/ml collagenase II, 50 IU/ml collagenase IV 0.025 mg/ml DNase I (STEMCELL Technologies; Vancouver, Canada) and 3 mM calcium chloride (Sigma-Aldrich; St. Louis, MO)] at 37 °C and cell extraction[64]. PBMCs were separated from blood using Ficoll Hypaque gradient centrifugation (Corning, Manassas, VA)[65]. A red blood lysis step was performed on both tumor single-cell suspensions and PBMCs using the 1× solution of RBC lysis buffer (ThermoFisher Scientific; Waltham, MA) for 2 min per manufacturer's protocol.

**Fluorescence-activated cell sorting (FACS)**. Prior to sorting cells, viability staining was performed using eBioscience Fixable Viability Dye eFluor 780 (ThermoFisher Scientific) per manufacturer's protocol. This was followed by a wash and staining with anti-CD45 PE (BioLegend; San Diego, CA) for 30 min in sorting buffer (0.1% BSA in PBS) at 4 °C. After washing, viable $CD45^+$ and $CD45^-$ cells were sorted using the Beckman Coulter MoFlo Astrios. Subsequently, the cells were washed twice and re-suspended in a sorting buffer. A cell number and viability count were performed on a Cellometer Auto 2000 using the ViaStain™ AOPI Staining Solution (Nexcelom Bioscience LLC, Lawrence, MA, USA) immediately prior to scRNAseq.

**Gel bead-in-emulsion (GEM) generation, reverse transcription, and PCR amplification**. Ready-to sequence Illumina single-cell cDNA libraries were generated using the Chromium Single Cell 3′ Reagent Kit (v2 Chemistry; 10× Genomics; Pleasanton, CA) per manufacturer's protocol. Single Cell Chip was loaded to retrieve 2000 PBMCs and 4000 cells for $CD45^+$ and $CD45^-$ tumor-derived cells.

**Sequencing**. Sequencing libraries (2 nM) were pooled as PBMC and TIL from the same patient and $CD45^-$ cells from two different patients. The resulting pooled libraries were diluted to 2 pM, denatured, and loaded on a NextSeq 500. For sequencing, NextSeq 500/550 High Output v2 kits (150 cycles) was used with the following parameters: Read 1: 26 cycles; i7 Index 8 cycles; Read 2: 98 cycles, as specified by the 10× Genomics guidelines.

**Generation of the aggregated gene-barcode matrix**. The CellRanger (v.3.0.0; 10× Genomics) pipeline provided was used to process the data. Initially, samples in each pool were demultiplexed using the sample index, and FASTQ files were created for each sample. From these FASTQ files, a counts matrix was generated for each sample by mapping the FASTQ files to a hybrid reference made of the human GrCh38 and the HPV16 genome (https://www.ncbi.nlm.nih.gov/nuccore/NC_001526.4). The counts matrix comprises gene expression by barcode for each cell present in the sample. In the final step, all these count matrices were aggregated into a single gene-barcode matrix.

**Quality control (QC) and filtering the data set**. QC metrics such as the estimated number of cells, mean reads per cell, sequencing saturation from the CellRanger Web Summary file were interrogated to ensure that the sequencing output was suitable for downstream analyses. Based on the QC metrics suggested in the Scanpy tutorial[17], cells with less than 200 genes expressed were filtered out. Cells expressing more than 5000 genes, and more than ten percent mitochondrial genes were also removed to ensure only the high quality of cells used in the downstream analyses. Genes expressed in less than 3 cells were also filtered out of the analysis.

**Normalization, dimensionality reduction, and data visualization**. After performing the filtering steps, the data set was normalized to correct for library size bias by scaling expression values to 10,000 counts per cell to control for differential sequencing depth per cell. The count-normalized expression matrix was then log normalized. Highly variable genes were identified based on dispersion (normalized dispersion greater than 0) and mean expression (between 0.0125 and 5) and carried forward to the subsequent analysis steps. The effects of mitochondrial genes and UMI (Unique Molecular Identifier) counts per cell were regressed out using simple linear regression as implemented in the Scanpy package (v1.4.5.post2) and the data were scaled to unit variance. All values exceeding standard deviation 10 were clipped. Principal Component Analysis (PCA) was used to reduce the dimensionality of the data. To denoise the data, the first 10 principal components were selected since they accounted for most of the variation in the data. A neighborhood graph was computed based on the PCA representation of the data and this graph was visualized using UMAP plots[15].

**Clustering and cell type assignment**. First, a neighborhood graph was constructed to identify related groups of cells. Next, Leiden clustering was performed on the neighborhood graph. The cluster assignments were then visualized on UMAP plots. Using a combination of top expressed genes in each cluster and a list of known marker genes (Supplementary Fig. 1A), cell types were assigned to each cluster.

**Calculating differentially expressed genes (DEGs) and gene set enrichment analysis**. DEGs were calculated using the "rank_genes_group" function in Scanpy using the Wilcoxon test. The DEGs are ranked by the $z$-score by default. The "Compute Overlaps" tool under the "Investigate Gene Sets" feature on the Molecular Signatures Database (MSigDB) was used to compute overlaps between the set of DEGs and gene sets in the MSigDB. We acknowledge our use of the gene set enrichment analysis, GSEA software, and Molecular Signature Database (MSigDB)[66]. For gene set enrichment analyses that did not start with a pre-calculated DEG list, the "singleseqgset" R package (v0.1.0.9000) was used to look for gene signatures enriched in the several clusters within each cell type[9].

**Using InferCNV to differentiate malignant and non-malignant cells**. InferCNV (v.1.2.1) was used to distinguish between the malignant and normal epithelial cells. Cells derived from the PBL were annotated as "Normal" to establish the baseline signal for gene expression. The epithelial cells were annotated as being "Tumor" and each cluster was annotated separately. Patterns between the baseline signal seen in the PBL cells were compared to epithelial cell clusters to determine if the cells in each cluster were malignant or normal epithelial cells.

**Deconvolution of bulk RNASeq data using CIBERSORTx**. Due to file size limitations on the CIBERSORTx web server, 5000 annotated cells were randomly sampled from the TME. Expression profiles of these cells were used to create the single-cell reference matrix. The "Impute Cell Fractions" module was used to infer proportions of all the cell types identified by our scRNA-Seq in HNSCC TCGA samples.

**Gene set score calculation and survival analysis**. A gene signature was created using the top 100 DEGs based on the score generated by the rank_genes_groups() function in Scanpy from fibroblasts with elastic differentiation as well as CAF. A GSVA score was computer per sample using TCGA HNSCC RNAseq data. All normal samples from the TCGA cohort were excluded from the analysis. Based on the median of the GSVA score, the samples were divided into 2 groups: High (above the median) or Low (below the median). Kaplan-Meier plots were generated to compare the overall survival in groups of interest using the survfit() function from the "survival" R package (v.3.2-11).

**All-component total immune cell scoring on H&E slides**. To segregate inflamed tumors, H&E staining was utilized as a simple, unbiased, and comprehensive measure of the lymphocyte content of each tumor (Supplementary Fig. 1C). Immune/inflammatory cells per 10 high-power fields were quantified by a trained pathologist unbiased to clinical-pathological information and values averaged. The sum of the average counts in the tumor edge, tumor and stroma was calculated as all-component total immune cell score. Samples above the median inflammation score were categorized as inflamed and samples below the inflammation score were non-inflamed. One patient's slides (HN03) had no viable tumor cells (only necrosis and stroma) after its use for fresh digestion and scRNAseq, so it was excluded from analysis of inflammation status. Furthermore, all UMAP plots showing inflammation scores do not include PBL cells and all cells from HN03.

**RNA in situ hybridization for high-risk HPV**. Three clinical cases of metastatic squamous cell carcinomas with unknown primary tested were reviewed with Advanced Cell Diagnostics (RNAscope® HPV-HR, Hayward, CA). All slides were handled based on the manufacturer's guidelines. Images were acquired using an Olympus BX45 (Olympus K.K., Tokyo, Japan).

**Quantitative PCR (qPCR) for HPV genes**. RNA was isolated from scraped FFPE tumor sections using the AllPrep DNA/RNA FFPE kit (QIAGEN; Germantown, MD). C3.43, an HPV-transformed cancer cell line, was used as the positive, while the HPV⁻ JHU029 cell line was used as the negative control for the analysis. Reverse transcription was performed with total RNA using the Maxima H Minus First Strand cDNA Synthesis Kit (ThermoFisher Scientific, K1652)[67]. qPCR analysis was performed using the SsoAd-vanced Universal SYBR Green Supermix Kit (Bio-Rad, 172-5272). The relative expression levels of target genes were normalized to the expression level of HPV16 E6 or E7, which yielded $2^{-\Delta\Delta Ct}$ cycle threshold (Ct) values. All reactions were run in triplicate. The primers used for gene expression were as follows: 5′-CAGGAGCGACCCAGAAAGTT-3′ (forward) and 5′-GCAGTAACTGTTGCTTGCAGT-3′ (reverse) for HPV16 E6; 5′-CCGGACA-GAGCCCATTACAA-3′ (forward) and 5′-GCTTTGTACGCACAACCGAA-3′ (reverse) for HPV16 E7; 5′-GGTGTTGAGGTAGGTCGTGG-3′ (forward) and 5′-CACACCTGCATTTGCTGCAT-3′ (reverse) for HPV16 L1; 5′-GAATTGGAA-CAGGGTCGGGT-3′ (forward) and 5′-AAGGGCCCACAGGATCTACT-3′ (reverse) for HPV16 L2.

**IHC staining and quantification**. 5uM thick formalin-fixed paraffin-embedded (FFPE) tissue sections for selected patient tumor samples were mounted on slides. Sections were deparaffinized at 60 °C for 30 min. and rehydrated using a standard histology protocol. Antigen retrieval was performed using an EDTA buffer (Cell Signaling, Danvers, MA) in Decloaking chamber at 120 °C for 2 min. The slides were stained using an Autostainer Plus (Agilent Dako) platform with TBST rinse buffer (Cell Signaling). The IHC slides were treated with 3% hydrogen peroxide for 5 min. The primary antibody, MFAP4 (Rabbit Polyclonal IgG; cat.# NBP2-30439, Novus Biological, Centennial, CO) was applied using a dilution of 1:200, at room temperature for 30 min. The detection applied, consisted of Mach 2 Rabbit, HRP (Biocare Medical, Pacheco, CA) for 20 min. at room temperature. The substrate, 3,3, Diaminobenzidine+ (Agilent Dako), was applied for 5 min. The slides were then incubated in Denature solution (Biocare Medical). The slides were then counterstained with Hematoxylin (Agilent Dako).

**Flow cytometry analysis**. Single-cell tumor suspensions were stained with Zombie NIR (Biolegend) labeling per the manufacturer's protocol. Samples were washed with FACS buffer (0.2% BSA, 0.02% NaN₃, PBS) and labeled for 20 min at 4 °C using the anti-human antibodies detailed in Supplementary Table 3. Data were acquired using LSR Fortessa cytometer (BD Biosciences) and analyzed using FlowJo version 10.6.1 software.

**Vectra staining and imaging**. Multispectral immunohistochemistry was performed on 5uM thick FFPE tissue sections using Akoya Manual 7 color IHC kit (Cat# NEL811001KT). Briefly, tissues were deparaffinized followed by 7 cycles of antigen retrieval, blocking, primary antibody followed by secondary- HRP and Opal staining[9]. Panel markers and dilutions used are listed in Supplementary Table 4. Imaging was performed at ×20 on the Vectra Polaris 3.0.

**Analysis of multispectral immunofluorescence images**. Images were inspected using Akoya InForm® (v2.4.6) and Phenochart™ (v1.0)(Akoya Biosciences, Inc.) to select ROIs consisting of both tumor and peritumoral/stromal regions. Channels were spectrally unmixed and exported as multi-channel composite TIFF files for data analysis in QuPath (v0.2.3), including cell segmentation by watershed algorithm and classification of marker positive/negative cells using a machine learning approach[68]. Briefly, a small number of positive/negative cells (30–50 per class per ROI) were manually selected from multiple ROIs as the training set to build a Random Forests model, which was then used to predict all remaining cells. Tumor/peritumoral/stromal compartment annotation was performed using pixel classification by pan-CK intensity signals. Measurement matrices consisting of centroid position (x,y), per-channel intensity, and class label of phenotyped cells were exported and further processed in R (v4.0.3). PD-L1⁺ cell density was calculated as the number of PD-L1⁺ cells divided by CD68⁺ cells, PanCK⁺ cells, or all cells from each compartment. Distance between CD68⁺PD-L1⁺ or pan-CK⁺PD-L1⁺ cell subsets to CD3⁺CD8⁺CD68⁻ cell subsets was computed using the nearest neighbor algorithm with Euclidean distance, defined as the smallest distance between each CD68⁺PD-L1⁺ or pan-CK⁺PD-L1⁺ cell to CD3⁺CD8⁺CD68⁻ cell within the same compartment. Differences in mean PD-L1 intensity per ROI between groups of interest were tested using a linear mixed-effects model based on restricted maximum likelihood in R (lme4[69] (v1.1.26)), with cell type (CD68⁺PD-L1⁺, pan-CK⁺PD-L1⁺) as the fixed effect and individual patient as the random effect. Contrasts of cell subsets were performed using t-tests with the Benjamini–Hochberg FDR adjustment for multiple comparisons. Marker intensities were log-transformed before statistical testing. All tests are two-sided unless otherwise noted.

**Reporting summary**. Further information on research design is available in the Nature Research Reporting Summary linked to this article.

## Data availability

Raw data are available on NCBI Sequence Read Archive: accession ID SRP301444. Processed gene barcodes are available on the Gene Expression Omnibus database: accession ID GSE164690. The bulk RNAseq and clinical HNSCC data utilized for survival analysis and deconvolution using CIBERSORTx from TCGA is available through the Broad Genome Data Analysis Center Firehouse (https://gdac.broadinstitute.org/). Gene signatures from the MSigDB can be found on the database website (http://www.gsea-msigdb.org/gsea/msigdb). The remaining data are available within the Article, Supplementary Information or Source Data file. Source data are provided with this paper.

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

## Acknowledgements

We thank Christine Platania for organizational and administrative support in this project; Merida Serrano, Amy Cuda, and Denise Kroll for assistance with patient sample procurement; Bratislav Janjic and Ernest Meyer (Hillman Cancer Center Flow Cytometry Core) for cell sorting and support with flow cytometry; Robert Lafyatis, Tracy Tabib and Zengbiao Qi (the University of Pittsburgh Single Cell Core) for sc-RNAseq library preparation; William Horne (Health Science Core Research Facilities Genomics Research Core) for next-generation sequencing, Hillman Cancer Center Tissue and Research Pathology Services (TARPS) staff for tissue staining, Simion I. Chiosea for performing in situ hybridization for high-risk HPV E6/E7 mRNA, the University of Colorado Denver Human Immune Monitoring Shared Resource (Kimberly Jordan and Angela Minic) for multicolor fluorescence scanning; the Immunologic Monitoring and all members of the Ferris lab for helpful discussions. The results shown here are in part based upon data generated by the TCGA Research Network (https://www.cancer.gov/tcga). This research was supported in part by the University of Pittsburgh Center for Research Computing through the resources provided. This research utilized the Hillman Cancer Center Flow Cytometry Core Facility, supported in part by award P30 CA047904 (RLF). Research funding: P50 CA097190, R01 CA206517, Hillman Foundation, Mosites Initiative for Personalized Head and Neck Cancer Therapy. A.R.C. was supported by the CITP T32 CA082084 and a Hillman Postdoctoral Fellowship for Innovative Cancer Research. C.H.L.K. was supported by the Programm zur internen Forschungsförderung Essen (IFORES) and the UMEA Junior Clinician Scientist Stipendium.

## Author contributions

C.H.L.K. designed and performed experiments, interpreted data, and wrote the manuscript. A.K. analyzed and interpreted data and wrote the manuscript. L.V. designed and performed experiments, interpreted data, and wrote the manuscript. R.L.F. conceived, designed, and funded the project, interpreted the data, and wrote the manuscript. T.A.C., S.L., T.B., and D.A.A.V. provided expertise in experimental design and data analysis. P.M.S. and A.K.R. performed flow cytometry experiments, while S.O. performed multicolor immunofluorescence staining. A.L. performed the H&E and IHC slide review and inflammation scoring. R.B. analyzed the multispectral immunofluorescence images. C.R., J.F., and J.-S.G. performed RNA isolation and RT-PCR for HPV genes; X.C., R.B., and X.L. provided input on data analysis and computational approaches. R.L.F., U.D., and S.K. identified patients and collected specimens. T.T. performed the library preparation for scRNA-seq, R.L. provided input for scRNA-seq library preparation and data interpretation. All authors reviewed and contributed to the manuscript.

## Competing interests

R.L.F: Aduro Biotech, Inc; EMD Serono; MacroGenics, Inc.; Numab Therapeutics AG; Pfizer; Sanofi Tesaro; Zymeworks, Inc (honoraria); Astra-Zeneca/MedImmune (clinical trial, research funding); Bristol-Myers Squibb (honoraria, clinical trial, research funding); Merck (honoraria, clinical trial); Novasenta (honoraria, stock, research funding); (research funding). D.A.A.V.: Stock: Novasenta, Tizona, Trishula, Oncorus, Werewolf, Apeximmune; Consultancy: Tizona, Werewolf, F-Star, Astellas, BMS, MPM, Incyte, Bicara, Apeximmune, G1 Therapeutics, Innovent Bio, Kronos Bio; Grants: BMS, Astellas, Novasenta; Patents licensed and Royalties: Astellas, BMS, Novasenta. The remaining authors declare no competing interests.
