## [Peer Review File · Nature Communications]

Reviewers' Comments:

Reviewer #1:

Remarks to the Author:

Kurten et al present scRNAseq analysis of 136,947 cells from HNSCC tumors and peripheral blood. While scRNAseq analysis has been performed for HNSCCs previously, this is the largest such dataset and the first to include HPV+ tumors. Novel observations include a subset of fibroblasts expressing an elastic fiber differentiation program. The authors also perform a more extensive analysis of predicted immune cell interactions than has previously been described in HNSCC, providing potentially translational insights given the use of checkpoint inhibitors in the clinic. Overall, the present study builds logically upon previous scRNAseq analyses, and the size and diversity of this dataset make a significant contribution to the field (and should be deposited in SRA before this paper is accepted).

1. The authors often label the HPV- tumors in this study as "carcinogen-induced." Do these tumors display transcriptomic signatures associated with specific carcinogen exposure, if not why not just label them as HPV-?
2. Regarding line 104: Are the proportions of inflamed vs. non inflamed tumors in either HPV+ or HPV- tumors consistent with previous reports/larger cohorts? Further, the samples are grouped into inflamed and non-inflamed subtypes, based on an inflammation score (Figure 2A). If samples have an inflammation score less than the median, they are classified in the non-inflamed group. Why did the authors choose this heuristic cut point, and why did they choose to specifically subset these samples into only two classifications?
3. The inter- and intra-tumoral heterogeneity in viral gene expression is of interest, especially given recent work towards targeting viral oncoproteins (for example CUE101). The authors should elaborate on potential implications for therapeutic resistance in the discussion. Among malignant cells from HPV+ tumors, are differences in HPV gene expression associated with any other differentially expressed gene expression programs? Do cells highly expressing viral oncoproteins localize to a particular area of the tumor? IHC or RNAscope or other imaging should be performed in tissues to validate this observation and confirm that it is not an artifact of the sequencing platform.
4. In Figure 4D (described on page 9) the authors show that the top 100 DEGs from their elastic fibroblasts are associated with poor survival in HPV+ TCGA patients. Is the survival association truly specific to these elastic fibroblasts? If the analysis were performed using DEGs characterizing fibroblasts in general, would the result be the same? A heatmap describing DEGs for each fibroblast subtype and associations with survival would be helpful in contextualizing this finding and understanding whether the poor survival is unique to elastic fibroblasts.
5. Are there any clues in the gene expression program as to the functional role of this subtype that could be expanded on in the discussion? The gene list should also be included in the supplement.
6. Is there any supporting evidence for elastic fibroblasts in additional tissue samples that can be expanded upon? Staining of tissue samples by IHC and/or RNAscope should be performed to support the distribution of these states.
7. The authors should comment on inflamed/non-inflamed status when describing CD274-PD1 expression and interaction. Spatially, do either CD274+ macrophages or epithelial cells co-localize with T cells in inflamed and non-inflamed tumor specimens?
8. The color scales in all figures (heatmaps, dot plots) should be labeled with units. This applies to several panels in Figures 3-6.

Minor points/typographical errors:

Figure 4C, 5D- color scale labeling is incomplete.

Line 144: Possible typo (HPV-) intended to read HPV+, as this section seems to contrast the two

etiologies

Line 200: First word possibly a typo

Line 200: This statement requires a citation

Line 286: typo (HLD DR)

Line 315: "TME" should not be in parentheses

Line 322: Typo (HNSSC)

Line 334: More accurately, Puram et al (reference 10) describe a signature they term "partial-EMT" (vs. "pre-EMT")

Line 350: "observation that elastic fibroblast gene signature has negative prognostic value" might be more accurate?

Reviewer #2:

Remarks to the Author:

Modeling of Immune-Epithelial-Stromal Cell Interactions in the Head and Neck Tumor Microenvironment by Single-Cell RNA Sequencing

The major claims of the paper are:

1. First comprehensive scRNAseq data of HNSCC patients, including: inflamed vs. non-inflamed tumors, HPV+ vs HPV-, immune cells vs. non-immune cells from the TME.
2. Description of a new TME stroma sub-population: elastic fibroblasts that have a negative prognostic value for HPV+ patients.
3. Prediction of therapeutically targetable checkpoint receptor-ligand interactions.
4. Tumor-associated macrophages are the primary contributors of PD-L1 and other immune checkpoint ligands in the TME.

Part of the findings of this manuscript are novel and some corroborate previous finding made by the authors and others. The findings and the comprehensive data on HNSCC patients dataset would be of interest to the community and to other solid tumors researchers.

This work contains satisfactory cohort and scRNAseq data. However, few comments needs to be addressed prior to publication:

Moderate:

1. Line 229: "The overall number of shared interactions between CD8+ T and non-immune cells was 586. In contrast to inflamed tumors that had 63 predicted unique R-L interactions between CD8+ T and non-immune cell types, non-inflamed tumors had 42." – it is not clear what is the background here. In addition, Fig 6A is not clear, it looks like CD8+ cells has more interaction through ligands compared to receptors.
2. Line 232: The authors show that "The interactome between CD8+ T and APC had 653 shared interactions. 13 unique interactions were predicted in inflamed vs. 26 in non inflamed tumors." and then they suggest that "APC-mediated interactions may play in the regulation of tumor inflammation status" – This reviewer finds this finding somewhat surprising, usually during inflammation interactions between immune cells increase. In any case do the authors think that regulation/suppression require less interactions than activation?
3. Line 159: "clustered adjacent to each other, suggesting transcriptomic similarity." Additional analysis should be performed to support this claim of similarities between different clusters.

Minor:

1. As supplemental figure please provide also a tSNE plot with # of genes per cell.
2. Some of the marker genes in Fig S1A. are very low for example CD4+ cells hardly express CD4.
3. Fig S1B. patient #HN05 is singular maybe due to age (15) and therefore, express different relative proportion (only macrophages) – The authors should comment on this and might consider removing this sample in certain analysis.
4. Inflammation score should be added to the patient summary table
5. Fig S2E is not exactly supplemental figure but rather additional analysis and should go together with Fig 2D.

6. Fig S3A is not clear (small font in X axis, maybe they should add the cluster number). In addition, what was the purpose? comparing PBL to epithelial cells in order to identify malignancy.
7. Line 109: "and CD4+ T cells (median: 22.9% vs. 35.3%, p-value = 0.185)." - need to be consistent with the p-value thresholds
8. Figure 2D panel inflammation why there are less cells?
9. Line 127 could the authors provide also a heatmap of top DE genes per cluster
10. Line 133 missing figure ref probably Figure 2D panel inflammation, where it is shown that clusters 1,4,8,9 are enriched?
11. Line 133 Cluster 4 is from one patient.
12. Line 142 "from non-inflamed tumors are stressed by the surrounding TME and may become (pre)- apoptotic." - Please mentioned some relevant genes
13. Line 161: "etiology were detected by gene set enrichment analysis of the top 100 genes defining each group" - a better threshold would be based on statistical significance
14. Line 169: if this is trivial why show it? Can they speculate on the virus state? Does the abundance correlates with anything else?
15. Line 180: "allocated to 4 different cell types based on the highest expressed genes (Fig. 4B) and differentially expressed hallmark gene sets (Fig. 4C)." - please refer to the relevant supp figure.
16. The authors did not mentioned any results regarding differences between smoking and alcohol.

Reviewer #3:

Remarks to the Author:

I read "Modeling of immune-epithelial-stromal cell interactions in the head and neck tumor microenvironment by single cell RNA-sequencing" by Kurten et al with great interest. The authors interrogate a group of HPV+ and HPV- head and neck tumors, focusing their studies on the tumor microenvironment and dovetailing on their recently published study in Immunity. Overall, I was quite excited to read through the manuscript but left disappointed by the lack of novelty and insights provided, gaps in data interpretation, and lack of biological validation. I think this manuscript would be better suited in a more clinically oriented journal, where the target audience might find the reported associations of greater interest.

MAJOR COMMENTS:

1. My primary concerns regarding novelty relate to the authors most exciting findings comparing HPV+ and HPV- tumors already being shared in their recent article. Here the emphasis is on elastic fibroblasts as well as expression of PDL1 by TAMs, the latter of which has been well documented across oncology, including in H&N. Similarly, the idea of a more fibrous stroma as supported in some of the authors refs is fairly well developed and so I do not think the description of an "elastic fibroblast" really advances the field in the way expected for this journal. In particular, the idea of activated fibroblasts expressing elastin related proteins/fibers has been previously suggested in both oncology and other pathologic contexts. In addition, the findings related to elastic fibroblasts raise concerns as outlined below.
2. The authors appear to have "re-used" several tumors from their recent publications in Immunity. By my count there are at least 9 of their 20 tumors for scRNA-seq that are the same and this to me represents a major issue in terms of being scientifically transparent about their dataset (i.e. republishing published data) and secondly in terms of their being substantial batch effects. I defer to the Editor regarding this major concern. Along these lines, are the data robust to their original cohort of origin (this paper vs the old dataset).
3. The median number of detected genes seems to be quite low. Typically this should be in the 2,000+ range for 3'v2 technology. The threshold for cell exclusion of cells with <200 genes detected also seems quite liberal. Typically this cutoff would be at least 500 genes. In addition, the QC as outlined in the methods should be better shown and documented. Again the cutoff of mitochondrial genes of 10% seems quite high, as typically this number would be lower. Combined with the low number of cells per sample on average, this is an issue. Typically one 10X channel will yield high quality data from at least 5,000 cells, at least based on what other single cell studies have suggested.
4. Can the authors re-demonstrate their clustering with an alternative method to confirm it is robust? kNN or tSNE would both be informative.

5. The variability in epithelial cells captured (S1B) raises questions regarding the technical processing of such tumors – some samples have almost no epithelial cells captured, which is in contrast to other studies with scRNA-seq and leads to questions about whether these data are representative and reflect poor tissue processing that may be biasing the results.
6. The authors state that their inflammatory tumors have greater lymphocyte infiltrate, yet only CD8 cells seems to be statistically different between the two (S2B) and in some sense this appears a bit self-fulfilling: Tumors that are categorized as inflamed are then shown to have more immune cells – seems tautological. The tumors appear to be arbitrarily separated into these groups (indeed some tumors in the inflamed group have less CD8 cells than those in the non-inflamed group).
7. “Differences in TIL content between patients were visualized with UMAP and quantified” – this does not make sense to me – one cannot quantify differences in TIL content based on a UMAP. Along these lines, I have major concerns with making any quantitative conclusions about cell type proportions based on droplet-based approaches. Processing of tumors can be widely variable which has been well documented by Mario Suva and others and this could absolutely represent random variation or technical artifact. Thus, comments about relative enrichment of fibroblasts or any cell type for that matters based on scRNA-seq simply are not reliable.
8. Pseudotime analyses across multiple tumors is fraught with error in my opinion and this analysis is not convincing or helpful.
9. The authors do not ever show differences in inflamed and non-inflamed tumors based on HPV but they subtly hint and mention this throughout. I think this aspect of their findings needs to be much more clear.
10. Could the inflammatory signature reflect post-processing or ischemia time? In this case, the phenomenon mentioned might be purely artefactual.
11. The finding of elastic fibroblasts seems to be limited to very few tumors based on the UMAP with patient annotations, raising concern about whether this is a bona fide subtype of cells. Furthermore, in Fig 4B, the same elastin markers do not come up in unbiased analyses raising further question. Moreover, the term elastic fibroblasts is somewhat interesting but I do not think the authors have really met the standard of thoroughly demonstrating this to be a real cell type. These cells are not shown by IHC, they are found in only a few tumors, and they are not that plentiful. This represents a serious concern of the generalizability of the reported findings, which are one of the paper’s main conclusions.
12. The use of the top 100 DEGs to define elastic fibroblasts seems quite arbitrary, and a more rationally driven cutoff based on the degree of differential expression would make more sense. The finding of worse outcome in elastic fibroblasts is somewhat predictable in the sense that CAFs have broadly been associated with worse survival across oncology and HNSCC (e.g. mesenchymal subtype). It would behoove the authors to show that their effect is somehow specific to elastic fibroblasts. Did the authors use deconvolution to define the elastic fibroblast score? If not, then there may be other cell types contributing to the score even though it is based on DEG that does not mean the absence of other cell type contributors. The proportion of elastic fibroblasts inferred from bulk data would be more meaningful. Also, why are the data in the KM curve limited to just 90 patients – TCGA has 500+ tumors and it would be important to show this in all patients. In addition, TCGA provides a rich source of clinical data and it is possible that elastic fibroblasts merely reflect inflammation from smoking, for example, and therefore an MVA showing the presence of elastic fibroblasts has an independent effect on survival will be important. Finally, how are the elastic high and low strata defined – given the low number of patients included in this analysis I suspect only patients at the extreme were analyzed which is worrisome regarding the generalizability of these findings. I’d like to see the KM curve with all tertiles/quartiles/bins.
13. Elastic fibroblasts are proposed to have a specifically worse impact on HPV+ patients but 1) this is not clearly shown as far as I can tell anywhere in the paper and 2) the elastic fibroblasts are predominantly found in HPV- patients based on the UMAP.
14. I find Fig 6 problematic – the differences between inflamed and non-inflamed tumors are not substantial or really convincing in my opinion. There is no statistical quantification and “by eye” the changes are quite modest. The interactions as described would really need to be confirmed by IHC to have any real external validation.
15. Having a higher % of PDL1 positive cells is not the same as showing a given cell type is the major contributor of PDL1. There is a hole here between the data and the conclusions. The authors do not draw this careful distinction and the FACS analyses aren’t able to justify the conclusions as stated.

MINOR CONCERNS:

1. The authors should really cite other important single cell studies of H&N including Sharma et al, Nature Communications, 2018. This represents a very important study in the field.
2. Are there other clinical factors that vary between patients such as prior radiation or chemo? Are all tumors primaries? I was surprised to see a 15 yo patient included in this study as most IRBs/tumor banks exclude children. Please confirm this age is accurate and covered by the IRB. Why do some tumors have no T-stage and no N-stage?
3. The marker based UMAPs are not really meaningful (S1C) as presented -- a UMAP of all clusters is really where this should be shown to demonstrate specificity for the various subpopulations. Why does the UMAP have a different topology depending on the marker if these are all DCs?
4. Why do the authors not see a separate cluster of regulatory T-cells as has been shown in other solid tumors among their T-cell clusters? They do show this in the original clustering.
5. S3A – Cluster 6 and cluster 4 (second from bottom) seem to be quite similar. It appears that cluster 4 might also be non-malignant cells. I am not sure what “intermediate status” that the authors refer to really means as this term is not used in the literature.
6. There is mention of etiology dependent “spatial adjoinment” in the UMAP in 3A but I remind the authors that distance/closeness in a UMAP is not representative of similar expression and should not be interpreted or stated as such.
7. What is the implication of variability in viral gene expression – is the conclusion merely that viral proteins have heterogeneous expression in HPV? This needs to be expanded.
8. Where are the cancer associated pericytes on the original UMAP in Fig 1? I would typically think of pericytes as endothelial associated cells so I find their including in Fig 4 a bit strange.
9. The authors mention that inflamed tumors have greater T cell infiltrate and likely more favorable outcome as a result, yet inflamed tumors are also suggested to have more elastic fibroblasts and thus a worse outcome. This contradiction is bothersome.

Robert L. Ferris, M.D., Ph.D.
Director, UPMC Hillman Cancer Center
Hillman Professor of Oncology
Associate Vice-Chancellor for Cancer Research
Co-Director, Tumor Microenvironment Center
Professor of Otolaryngology, of Immunology, and of Radiation Oncology

UPMC Hillman Cancer Center
Cancer Pavilion, Suite 500
5150 Centre Avenue
Pittsburgh, PA 15232
Tel: 412-623-3205
Fax: 412-623-3210
Email: ferrisrl@upmc.edu

We would like to thank the reviewers for the positive and thoughtful feedback, as well as their patience during these trying times. Please see below for point-by-point responses to their comments and concerns. Major edits are highlighted within the paper.

General comment: The revised manuscript draft is newly titled, “Investigating Immune and Non-Immune Cell Interactions in Head and Neck Tumors by Single-Cell RNA Sequencing.” For this revision, the data were reanalyzed to be updated to a newer version of the scanpy pipeline (see Traag et al., reference #18 in updated manuscript), in which the clustering algorithm was changed from Louvain to Leiden. Also, based on the comment from Reviewer #3, patient number HN05 was removed due to its (clinical and biological) singularity. A separate figure (updated **Fig. 4**) was included to introduce the stromal compartment.

Response to reviewer comments

Reviewer #1

1. *“Kurten et al present scRNAseq analysis of 136,947 cells from HNSCC tumors and peripheral blood. While scRNAseq analysis has been performed for HNSCCs previously, this is the largest such dataset and the first to include HPV+ tumors. Novel observations include a subset of fibroblasts expressing an elastic fiber differentiation program. The authors also perform a more extensive analysis of predicted immune cell interactions than has previously been described in HNSCC, providing potentially translational insights given the use of checkpoint inhibitors in the clinic. Overall, the present study builds logically upon previous scRNAseq analyses, and the size and diversity of this dataset make a significant contribution to the field (and should be deposited in SRA before this paper is accepted).”*

Response: We thank Reviewer #1 for the comments and appreciate that our work is considered a significant advancement to the field. We feel the reviewer understands that this comprehensive dataset of the head and neck tumor microenvironment can be used as a resource for future exploratory and confirmatory studies. We retrospectively confirmed that our dataset has an adequate number of PBL and CD45⁺ cells sequenced using the SCOPIT algorithm (Davis, A., et al. *BMC Bioinformatics*. 2019; 20(1): 566.). Mast cells were the rarest cell type identified in both the PBL and tumor cells in our cohort. Using the frequency of mast cells, the number of cells to be sequenced for a 0.95 probability of success was predicted to be 4575 for PBL and 69981 for CD45⁺ cells and our dataset meets both these thresholds (36,390 cells from PBL and 71,102 CD45⁺ cells). To facilitate the usage of our comprehensive dataset, we have uploaded the raw data to SRA and the processed files to GEO (GSE164690). The more specific comments are addressed in detail below.

2. *“The authors often label the HPV- tumors in this study as “carcinogen-induced.” Do these tumors display transcriptomic signatures associated with specific carcinogen exposure, if not why not just label them as HPV-?”*

Response: This interesting point is beyond the scope of our paper, so we agree to now adopt the naming suggestion of the reviewer for all non-HPV+ patients as “HPV-”.

3. *“Regarding line 104: Are the proportions of inflamed vs. non inflamed tumors in either HPV+ or HPV- tumors consistent with previous reports/larger cohorts?...”*
Response: Yes. HNSCC immune cell infiltrates can vary according to anatomical subsite and etiology. Given that there is not a single universal method to classify the HNSCC TME into inflamed vs. non-inflamed tumors, we used a median to divide the cohort into two groups, as well as a tertile approach which separated them into 3 groups (as used by Mandal et al *JCI Insight* 2016; 1(17): e89829, see point #4 below), observing similar results. We now reference other publications (refs. 19 and 20) in the manuscript to support the consistency of our findings.
4. *“...Further, the samples are grouped into inflamed and non-inflamed subtypes, based on an inflammation score (Figure 2A). If samples have an inflammation score less than the median, they are classified in the non-inflamed group. Why did the authors choose this heuristic cut point, and why did they choose to specifically subset these samples into only two classifications?”*
Response: Due to the aforementioned limitations in classifying the HNSCC TME, the methodology to choose a cutoff point is not standardized. To better represent the continuum of lymphocyte infiltration we have now divided the samples into three groups (low, medium and high; **Fig. 1D**). A similar strategy was previously implemented by Mandal, R. et al. *JCI Insight*. 2016; 1(17): e89829. The enriched gene sets are largely consistent with the previous comparison of two groups (**Fig. 2D**).
5. *“The inter- and intra-tumoral heterogeneity in viral gene expression is of interest, especially given recent work towards targeting viral oncoproteins (for example CUE101). The authors should elaborate on potential implications for therapeutic resistance in the discussion. ...”*
Response: We thank the reviewer for this comment. We indicate this therapeutic relevance in our discussion and cite the above-mentioned study on CUE101 by Quayle et al. alongside the study on TCR gene therapy by Doran et al. (references #54 and #55 in the updated submission; lines 403-406). We also present additional pathways expressed in cells with specific viral genes being transcribed.
6. *“... Among malignant cells from HPV+ tumors, are differences in HPV gene expression associated with any other differentially expressed gene expression programs? ...”*
Response: We have performed a gene set enrichment analysis showing that primarily gene sets concerning DNA replication and viral infection and response are upregulated in cells expressing viral genes (**Suppl. Fig. 5A**, see lines 197-199 of the updated manuscript).
7. *“... Do cells highly expressing viral oncoproteins localize to a particular area of the tumor? IHC or RNAscope or other imaging should be performed in tissues to validate this observation and confirm that it is not an artifact of the sequencing platform.”*
Response: To address this question, we have used *in situ* hybridization for high risk HPV E6/7 to validate the heterogeneous expression in tissue (**Suppl. Fig 5B**). Indeed, an enrichment of E6/7 expression in the basal epithelial layer (lines 199-206 of the updated manuscript) was observed, which explains the cellular heterogeneity seen in digested, scRNAseq profiles.
8. *“In Figure 4D (described on page 9) the authors show that the top 100 DEGs from their elastic fibroblasts are associated with poor survival in HPV+ TCGA patients. Is the survival association truly specific to these elastic fibroblasts? If the analysis were performed using DEGs characterizing fibroblasts in general, would the result be the same? A heatmap describing DEGs for each fibroblast subtype and associations with survival would be helpful in contextualizing this finding and understanding whether the poor survival is unique to elastic fibroblasts.”*
Response: We thank the reviewer for this comment. In addition to looking at CAF and elastic fibroblast gene signatures individually, we also performed an integrated analysis incorporating the elastic fibroblast and the CAF signature. Pericytes were removed due to comments from Reviewer #3. Our updated analysis demonstrates that, while CAFs have a negative prognostic impact (as previously shown), there is an added negative prognostic impact in patients with the elastic fibroblast transcriptional program (**Fig. 5D, Suppl. Fig. S7A-C**).

9. *“Are there any clues in the gene expression program as to the functional role of this subtype that could be expanded on in the discussion? The gene list should also be included in the supplement.”*
Response: We have performed gene set enrichment analysis using the GO BP gene sets from the MSigDB of elastic fibroblasts and CAFs which is now included in **Suppl. Fig. 6C**. We can show that CAFs and fibroblasts with an elastic differentiation both upregulate pathways associated with extracellular matrix formation and turnover. The gene list is included as **Suppl. Table 2**.
10. *“Is there any supporting evidence for elastic fibroblasts in additional tissue samples that can be expanded upon? Staining of tissue samples by IHC and/or RNAscope should be performed to support the distribution of these states.”*
Response: We performed IHC staining in 6 patients from this cohort using MFAP4 (Microfibril Associated Protein 4) as a marker for elastic differentiation. Our data show that MFAP4 is expressed only by cells of fibroblast morphology within the tumor stroma (**Suppl. Fig. 6D**; lines 245-248 of the manuscript), as scored by our pathologist collaborator, and provides their spatial distribution.
11. *“The authors should comment on inflamed/non-inflamed status when describing CD274-PD1 expression and interaction. Spatially, do either CD274+ macrophages or epithelial cells co-localize with T cells in inflamed and non-inflamed tumor specimens?”*
Response: The reviewer brings up an excellent question. A drawback of flow cytometry and scRNAseq is that they do not provide spatial associations of evaluated cell types within the TME once it is physically disaggregated. Therefore, it is impossible to establish a physical proximity between evaluated PD-L1⁺ cells and CD8⁺ T cells that is a prerequisite for cell-to-cell interactions. To address it, we have performed spatial analysis of five patient sections using the Vectra multispectral imaging system. Our data presented in **Figs. 8B-E** and **Suppl. Fig. 11** are described in the new section, “PD-L1+ macrophages spatially associate with CD8⁺ T cells in the HNSCC TME,” on pp. 14-15. In 4/5 patients tested, PD-L1⁺ macrophages closely associated with CD8⁺ T cells in both the tumor bed and stromal regions, regardless of their inflammation status. PD-L1+ tumor epithelial cells also associated spatially and co-localized with CD8+ T cells.
12. *“The color scales in all figures (heatmaps, dot plots) should be labeled with units. This applies to several panels in Figures 3-6.”*
Response: We thank the reviewer for this comment. The color scales in the heatmaps and dot plots are unitless depictions of relative gene expression. We have made this clearer in the figure legends.
13. *“Figure 4C, 5D- color scale labeling is incomplete.”*
Response: We have adjusted the color scale labeling in the updated Figures.
14. *“Line 144: Possible typo (HPV-) intended to read HPV+, as this section seems to contrast the two etiologies”*
Response: We have changed the subheading to “HPV+ and HPV- cancer cells” (see above).
15. *“Line 200: First word possibly a typo.”*
Response: We have changed “that” to “the”.
16. *“Line 200: This statement requires a citation”*
Response: We have added a citation (Kallkuri, reference number 27).
17. *“Line 286: typo (HLD DR)”*
Response: We have changed “HLD DR” to “HLA-DR”.
18. *“Line 315: “TME” should not be in parentheses”*

Response: We have removed “TME” from the parentheses.

19. *“Line 322: Typo (HNSSC)”*

Response: We have changed “HNSSC” to “HNSCC”.

20. *“Line 334: More accurately, Puram et al (reference 10) describe a signature they term “partial-EMT” (vs. “pre-EMT)”*

Response: We have changed “pre-EMT” to “partial-EMT”.

21. *“Line 350: “observation that elastic fibroblast gene signature has negative prognostic value” might be more accurate?”*

Response: In line with the reviewer’s comment, we have adjusted our terminology. Since reviewers pointed out the difficulty of differentiating a novel subset of cells from a substrate of other cells (in this case CAFs) we have changed “elastic fibroblasts” to “elastic fibroblast gene signature”, “elastic sub-state of fibroblast differentiation” or “fibroblasts with elastic differentiation.”

Reviewer #2

1. *“The major claims of the paper are: 1. First comprehensive scRNAseq data of HNSCC patients, including: inflamed vs. non-inflamed tumors, HPV+ vs HPV-, immune cells vs. non-immune cells from the TME. 2. Description of a new TME stroma sub-population: elastic fibroblasts that have a negative prognostic value for HPV+ patients. 3. Prediction of therapeutically targetable checkpoint receptor-ligand interactions. 4. Tumor-associated macrophages are the primary contributors of PD-L1 and other immune checkpoint ligands in the TME. Part of the findings of this manuscript are novel and some corroborate previous finding made by the authors and others. The findings and the comprehensive data on HNSCC patient’s dataset would be of interest to the community and to other solid tumors researchers. This work contains satisfactory cohort and scRNAseq data. However, few comments needs to be addressed prior to publication.”*

Response: We thank Reviewer #2 for the commentary and agree that this study is of interest to the researchers in HNSCC and beyond. Indeed, this dataset can be used to generate new hypotheses and corroborate previous findings. Specific comments are addressed in detail below.

2. *“Line 229: “The overall number of shared interactions between CD8+ T and non-immune cells was 586. In contrast to inflamed tumors that had 63 predicted unique R-L interactions between CD8+ T and non-immune cell types, non-inflamed tumors had 42.” – it is not clear what is the background here. In addition, Fig 6A is not clear, it looks like CD8+ cells has more interaction through ligands compared to receptors.”*

Response: We agree with the reviewer. While this analysis was meant to be an unsupervised method of observing R-L interactions between CD8⁺ T cells and other cell types, it raised more questions than answers and did not contribute significantly to our main stories. Consequently, due to space limitations and clear need to delve more deeply into experimental and bioinformatics investigation to validate and clarify these findings, we removed the respective paragraph and associated figure.

3. *“Line 232: The authors show that “The interactome between CD8+ T and APC had 653 shared interactions. 13 unique interactions were predicted in inflamed vs. 26 in non inflamed tumors.” and then they suggest that “APC-mediated interactions may play in the regulation of tumor inflammation status” – This reviewer finds this finding somewhat surprising, usually during inflammation interactions between immune cells increase. In any case do the authors think that regulation/suppression require less interactions than activation?”*

Response: Like the previous comment, we agree with the reviewer that data in this figure require additional experimental and bioinformatic validation that are beyond the scope of this manuscript. Thus, we have removed the figure and associated text.

4. *“Line 159: “clustered adjacent to each other, suggesting transcriptomic similarity.” Additional analysis should be performed to support this claim of similarities between different clusters.”*

Response: We thank the reviewer for this comment. It has been shown that UMAP preserves the global and local structure of data better than visualization methods such as t-SNE (see Becht et al., reference #15). To avoid confusion, we have rephrased that comment and performed additional analysis to show that HPV+ and HPV- clusters are more similar among each other (see lines 173-176 of the revised manuscript). The result of hierarchical clustering demonstrating this is shown in **Suppl. Fig. 4C**.

5. *“As supplemental figure please provide also a tSNE plot with # of genes per cell.”*

Response: We have generated tSNE and UMAP (to maintain consistency with other plots in our paper) plots of our data which displaying the number of genes per cell (**Suppl. Fig. 1A** and **Suppl. Fig. 2C**).

6. *“Some of the marker genes in Fig S1A. are very low for example CD4+ cells hardly express CD4.”*

Response: Cluster naming and cell type allocation is a subject of discussion and controversy in the single cell genomics field. We have employed a widely used approach where the co-expression of several canonical markers is used for cluster naming. We also looked at the fraction of cells expressing the marker genes to distinguish between cell types. Also, one must consider the differences between RNA and protein expression levels, which leads to different markers being used for cell type identification by scRNAseq than, for example, flow cytometry. We have used the single-color plots, where only one gene is displayed at a time. For the above-mentioned example of CD4+ T cells we have used IL7-Receptor (Gene: IL7R), a well-described marker for T helper cells in scRNAseq data. The UMAP plots below clearly indicate that both CD4 and IL7R are expressed on CD4+ T cells from our analysis.

7. *“Fig S1B. patient #HN05 is singular maybe due to age (15) and therefore, express different relative proportion (only macrophages) – The authors should comment on this and might consider removing this sample in certain analysis.”*

Response: We previously thought that this unique case might offer special insight into age-associated differences in HNSCC biology, but now agree that it complicates the main message of the study. As suggested, we have removed HN05 from the cohort and re-analyzed the data and have readjusted patient number assignments in **Suppl. Table 1**.

8. *“Inflammation score should be added to the patient summary table”*

Response: We agree with the reviewer. The inflammation status column (now in tertiles of ‘low’, ‘medium’, ‘high’) was added to **Suppl. Table 1**.

9. *“Fig S2E is not exactly supplemental figure but rather additional analysis and should go together with Fig 2D.”*

Response: We thank the reviewer for this comment. The pseudotime analysis was moved to the main figure (**Fig. 2B**).

10. *“Fig S3A is not clear (small font in X axis, maybe they should add the cluster number). In addition, what was the purpose? comparing PBL to epithelial cells in order to identify malignancy.”*

Response: Thank you for pointing this out. We manually added larger labels on the x-axis and the Cluster numbers on the y-axis, for the figure to be more legible than the automated software output. The purpose of the figure is to differentiate malignant epithelial cells from non-malignant ones using copy-number-variation (CNV). For this, the software needs a normal control that is

relatively quiescent and has an expected low CNV, which in our study is the patient's peripheral blood lymphocytes.

11. *“Line 109: “and CD4+ T cells (median: 22.9% vs. 35.3%, p-value = 0.185).” – need to be consistent with the p-value thresholds.”*

Response: The respective passage was removed due to comment #8 and #9 from Reviewer #3.

12. *“Figure 2D panel inflammation why there are less cells?”*

Response: The number of cells is lower in the inflammation analysis because the peripheral blood cells were excluded from the analysis (lines #129-130 of the updated manuscript). Including them would be erroneous, because the inflammation score is a measurement of the TME that does not necessarily reflect the cells in the circulation. Also, patient HN03's slides had no viable tumor cells (only necrosis and stroma) after its use for fresh digestion and scRNAseq, so it was excluded from analysis of inflammation status (lines #564-567 of the updated manuscript).

13. *“Line 127 could the authors provide also a heatmap of top DE genes per cluster”*

Response: We thank the reviewer for this comment. We have provided a heatmap of DEG in **Suppl. Fig. 3B**.

14. *“Line 133 missing figure ref probably Figure 2D panel inflammation, where it is shown that clusters 1,4,8,9 are enriched?”*

Response: We thank the reviewer for this comment. We have updated our figures to include the low, medium and high classification.

15. *“Line 133 Cluster 4 is from one patient.”*

Response: Please see our response to the previous comment. The updated analysis generally shows good patient mix in the respective clusters.

16. *“Line 142 “from non-inflamed tumors are stressed by the surrounding TME and may become (pre)-apoptotic.” - Please mentioned some relevant genes.”*

Response: This observation is referring to the fact that the hallmark gene sets of apoptosis and hypoxia were top hits in CD8+ T cells from non-inflamed TMEs. Examples of genes from our gene list that were found in the apoptosis and the hypoxia gene set were GADD45B (Growth arrest and DNA-damage-inducible, beta), BTG2 (BTG anti-proliferation factor 2), JUN (Jun proto-oncogene) and FOS (Fos proto-oncogene) (see lines 153-155 of the updated manuscript). These genes and transcription factors promote invasiveness and pro-survival pathways consistent with tumor progression.

17. *“Line 161: “etiology were detected by gene set enrichment analysis of the top 100 genes defining each group” – a better threshold would be based on statistical significance”*

Response: Currently, there is no agreement in the field on what constitutes a correct threshold for picking the number DEGs for downstream analysis. The DEGs are ranked by a score that includes both the fold change and p-value. We picked 5% of the median genes expressed by cells in our dataset. Any other cut-off for the top DEGs would also be arbitrary in some way and picking all the genes passing the statistical significance threshold might make our gene signature non-specific. Concerning the statistical significance, all selected genes were significantly and differentially expressed.

18. *“Line 169: if this is trivial why show it? Can they speculate on the virus state? Does the abundance correlates with anything else?”*

Response: We do not view this analysis as trivial. As far as we are aware, this is the first time HPV gene expression has been analyzed in a single cell data set in HNSCC or other cancers. To validate our observations and expand potential utility, we have now analyzed enrichment of other gene sets depending on viral gene expression in response to Reviewer 1's comment #6 (**Suppl.**

Fig. 5A). We have added discussion on the biological meaning of these findings in lines #394-406 of the revised manuscript.

19. *“Line 180: “allocated to 4 different cell types based on the highest expressed genes (Fig. 4B) and differentially expressed hallmark gene sets (Fig. 4C).” - please refer to the relevant supp figure.”*

Response: We now also refer to **Suppl. Fig. 6A**, as requested.

20. *“The authors did not mention any results regarding differences between smoking and alcohol.”*

Response: The patient smoking and alcohol status is listed in the **Suppl. Table 1**. Indeed, these carcinogens are an interesting patient characteristic due to their pathogenic relevance to etiology. However, the causal link between smoking/alcohol and an individual patient’s cancer is not always clearly ascribed, which is why we now refer to patients as either HPV+ or HPV-. We feel that a larger patient cohort specifically designed for this purpose (e.g., containing never-smoker/drinker vs. heavy smoker/drinker) would be necessary to thoroughly investigate this important question.

Reviewer #3

“My primary concerns regarding novelty relate to the authors most exciting findings comparing HPV+ and HPV- tumors already being shared in their recent article. Here the emphasis is on elastic fibroblasts as well as expression of PDL1 by TAMs, the latter of which has been well documented across oncology, including in H&N. Similarly, the idea of a more fibrous stroma as supported in some of the authors refs is fairly well developed and so I do not think the description of an “elastic fibroblast” really advances the field in the way expected for this journal. In particular, the idea of activated fibroblasts expressing elastin related proteins/fibers has been previously suggested in both oncology and other pathologic contexts. In addition, the findings related to elastic fibroblasts raise concerns as outlined below.”

Response: We address all comments below.

1. *“The authors appear to have “re-used” several tumors from their recent publications in Immunity. By my count there are at least 9 of their 20 tumors for scRNA-seq that are the same and this to me represents a major issue in terms of being scientifically transparent about their dataset (i.e. republishing published data) and secondly in terms of their being substantial batch effects. I defer to the Editor regarding this major concern.”*

Response: While we believe that we have been transparent about data usage, both in the cover letter and in the *Methods* section, we understand Reviewer’s concern. To make these comments more visible to the reader, we created a separate “Patient cohort” section within the *Methods* (lines 461-468) that explains our inclusion of unique patient specimens and that all data and analyses are novel and previously unpublished. Also, of the 15 lymphocyte samples that were used in the Cillo et al. study, the corresponding CD45- cells from 12 patients are unpublished. This is especially relevant as the majority of the analysis in this study is performed on CD45- cells. Furthermore, unpublished data from the additional 3 patients are added to our cohort (CD45+ PBL and TIL, as well as CD45- cells). This distinction from the previous study was further enhanced by the updated tissue analysis (IHC, ISH, Vectra) for the revised draft. Finally, we would note that others are also entitled to re-analyze data from prior publications, utilizing open source, publicly available data sets for exploratory or aggregated analyses. These new analyses would be considered appropriate and publishable.

2. *“Along these lines, are the data robust to their original cohort of origin (this paper vs the old dataset).”*

Response: All samples were processed in an identical fashion as described in the *Methods* (lines #468-490 in the updated manuscript), and 15 paired PBL and TIL specimens in Cillo et al were processed by Kurten (first author of the present manuscript) in the Ferris lab, supporting robustness and technical reproducibility. Samples were prepared fresh with tumor and peripheral blood in parallel to account for technical variation. The additional specimens included in this new manuscript demonstrate robustness with the original patients and dataset. We have inserted a Figure of all PBL by patient below to show that clustering is not patient or cohort specific.

3. *“The median number of detected genes seems to be quite low. Typically this should be in the 2,000+ range for 3’v2 technology. (...)”*

Response: Since our dataset is a mixture of various cell types originating from the peripheral blood and tumor, the reported number is the median across all samples. Indeed, the number of genes detected is not only determined by the technology used, but also by the activation state of the cells under investigation. We have added plots showing the number of genes captured per cell (**Suppl. Fig. 1A** and **Suppl. Fig. 2C**). Here one can appreciate that the number of genes detected is far higher than 3000 genes in activated lymphocytes (i.e. from the TME) and goes up to 5000 in stromal/epithelial cells.

4. *“...The threshold for cell exclusion of cells with <200 genes detected also seems quite liberal. Typically this cutoff would be at least 500 genes. In addition, the QC as outlined in the methods should be better shown and documented. Again the cutoff of mitochondrial genes of 10% seems quite high, as typically this number would be lower. Combined with the low number of cells per sample on average, this is an issue. Typically one 10X channel will yield high quality data from at least 5,000 cells, at least based on what other single cell studies have suggested.”*

Response: We followed the recommendation in the Scanpy package to pick a cut-off of 200 genes. Considering that PBL samples have around 1000 median genes, using a cut-off of 500 genes would be very stringent and would lead to potential loss of information. Although a higher % of mitochondrial genes may indicate stressed or dying cells, this is not the case for all cell types. To keep the analysis uniform, we retained the 10% cut-off since other studies have shown this to be a reasonable cut-off (Osorio et. al. Scientific Reports 2019, Al Janahi et al. Mol Ther Methods Clin Dev 2018).

5. *“Can the authors re-demonstrate their clustering with an alternative method to confirm it is robust? kNN or tSNE would both be informative.”*

Response: Clustering is a complex multi-step process which is dependent on many parameters chosen during the analyses. The analysis was now further improved by updating to the Leiden clustering algorithm, which was shown to outperform the previously used Louvian algorithm (see Traag et al., reference #18 in updated submission). The first step in the clustering process is creating a neighborhood graph (n=15 was used) which relies on the kNN method. Thus, the first clustering algorithm suggested by reviewer #3 already underlies the current analysis. The second step is embedding the neighborhood graph in 2D for which we use the UMAP projection. The third step is the final visualization which uses the neighborhood graph created earlier to cluster the cells. tSNE (the second “clustering” tool suggested by Reviewer #3) is a visualization method and not a clustering algorithm. We have added a tSNE plot that visualizes the clustering below.

To further demonstrate clustering robustness, we have subsampled our data to 90%, 80% and 70% of the dataset and repeated the clustering keeping all the parameters identical. We observe a high similarity in the number of clusters indicating that the clustering is robust even after subsampling the dataset (see below).

100% (full dataset, see Fig. 1B)

90% subsample

80% subsample

70% subsample

Finally, UMAP visualization using the scanpy pipeline is a popular single cell analysis tool, which is widely used. Changing parameters can lead to different clustering outcomes and evaluating each of these would need a thorough Bioinformatics methods comparison and is out of the scope of this study. Benchmark analysis of the algorithm has been previously performed (see Becht et al., reference #15).

6. *“The variability in epithelial cells captured (S1B) raises questions regarding the technical processing of such tumors – some samples have almost no epithelial cells captured, which is in contrast to other studies with scRNA-seq and leads to questions about whether these data are representative and reflect poor tissue processing that may be biasing the results.”*

Response: As with any laboratory technique, scRNAseq does not fully represent the original state of the tumor microenvironment and brings with it some experimental and technical variation. Our microscopic tissue analyses (H&E, IHC, Vectra) aim to overcome that issue. However, we would assert that these limitations are well known in the field of scRNAseq and the data presented here are of equal quality compared to previous studies. For example, compared to our **Fig. 1D**, the landmark paper of Lambrechts et. al. (reference #11 in updated submission) also showed a marked variation of the number of cancer cells. Other single cell studies, such as Puram et al. (reference #10 in updated submission) used different scRNAseq wet lab techniques (i.e., sorting into plates) and therefore acquired more consistent cell numbers. This advantage is, however, offset by a lower yield in overall cell number.

7. *“The authors state that their inflammatory tumors have greater lymphocyte infiltrate, yet only CD8 cells seems to be statistically different between the two (S2B) and in some sense this appears a bit self-fulfilling: Tumors that are categorized as inflamed are then shown to have more immune cells – seems tautological. The tumors appear to be arbitrarily separated into these groups (indeed some tumors in the inflamed group have less CD8 cells than those in the non-inflamed group).”*

Response: We thank the reviewer for this comment. As discussed above (see Reviewer 1’s comment #3) there is not a universally accepted way to classify the HNSCC TME into inflamed vs. non-inflamed tumors. We have used a composite lymphocyte score, that counts all infiltrating cells. This was done to overcome one limitation of scRNAseq, which is that absolute cell counts, and spatial organization are lost. The relative count of recovered immune cells now is descriptive only, with all significance testing removed, but corroborates that CD8 infiltration is a dominant feature. We now also mention different methods that have been proposed to classify the HNSCC TME based on inflammation. Concerning the cut-off for the inflamed vs. non-inflamed group we have now divided the samples into three groups (high, medium, low) to better represent the continuum of inflammatory cell tumor infiltration. As discussed above, the enriched gene sets are largely consistent with the former comparison of two groups.

8. *“Differences in TIL content between patients were visualized with UMAP and quantified” – this does not make sense to me – one cannot quantify differences in TIL content based on a UMAP. Along these lines, I have major concerns with making any quantitative conclusions about cell type proportions based on droplet-based approaches. Processing of tumors can be widely variable which has been well documented by Mario Suva and others and this could absolutely represent random variation or technical artifact. Thus, comments about relative enrichment of fibroblasts or any cell type for that matters based on scRNA-seq simply are not reliable.”*

Response: Based on the reviewer’s comment, we now rephrase this statement to be clearer: “The inflammation status was visualized depending on the sample of origin”. The significance testing was removed from this figure and it was moved to supplemental, as the relative content of cells recovered in each group are an important information to understand the composition of the data, but no conclusions are drawn on the absolute cell composition of the TME, which is why we used the microscopic methods.

9. *“Pseudotime analyses across multiple tumors is fraught with error in my opinion and this analysis is not convincing or helpful.”*

Response: We have now removed the pseudotime plots of cancer cells.

10. *“The authors do not ever show differences in inflamed and non-inflamed tumors based on HPV but they subtly hint and mention this throughout. I think this aspect of their findings needs to be much more clear.”*

Response: We thank the reviewer for this comment. **Fig. 2A** displays the inflammation and the HPV status of the patient cohort (now updated to contain low, medium, and high inflammation). This is commented on in lines #108-113 in the updated manuscript. There is no clear link between inflammation and HPV-status (In one HPV+ sample inflammation is low, 2 are medium, 3 are high). A division into 6 groups (inflammation low, medium, and high vs. HPV+/-) would be too granular.

11. *“Could the inflammatory signature reflect post-processing or ischemia time? In this case, the phenomenon mentioned might be purely artefactual.”*

Response: We thank the reviewer for this comment. Please see response to comment #6 from Reviewer #3 concerning the question of sample quality. On a given day, the samples (tumor and peripheral blood) were handled in parallel. Given the clustering across patients in PBL, there seems to be no batch effect due longer processing or ischemia time. Also, technical artifacts described above would affect all samples similarly and thus unlikely emerge from a comparison of two otherwise comparable groups of samples (inflamed vs. non-inflamed).

12. *“The finding of elastic fibroblasts seems to be limited to very few tumors based on the UMAP with patient annotations, raising concern about whether this is a bona fide subtype of cells. Furthermore, in Fig 4B, the same elastin markers do not come up in unbiased analyses raising further question. Moreover, the term elastic fibroblasts is somewhat interesting but I do not think the authors have really met the standard of thoroughly demonstrating this to be a real cell type. These cells are not shown by IHC, they are found in only a few tumors, and they are not that plentiful. This represents a series concern in terms of the generalizability of the reported findings, which are one of the paper’s main conclusions.”*

Response: We agree and have adjusted the terminology, now referring to this cell-state as “fibroblast with elastic differentiation” (see response to comment #20 of Reviewer #1), which we now show provides additional, statistically significant, negative prognostic value over that of CAF alone. We now show in updated **Fig. 5B** that important genes showing elastic differentiation (LTBP4, FBLN1 and MFAP4) are differentially expressed in an unbiased analysis. Additionally, we have stained for MFAP4 using IHC, as requested, and confirm that fibroblast morphology cells in the HNSCC stroma express this marker. These IHC results provide the requested validation of our scRNAseq data regarding biological and clinical importance of this cellular substate.

13. *“The use of the top 100 DEGs to define elastic fibroblasts seems quite arbitrary, and a more rationally drive cutoff based on the degree of differential expression would make more sense. The finding of worse outcome in elastic fibroblasts is somewhat predictable in the sense that CAFs have broadly been associated with worse survival across oncology and HNSCC (e.g. mesenchymal subtype). It would behoove the authors to show that their effect is somehow specific to elastic fibroblasts. Did the authors use deconvolution to define the elastic fibroblast score? If not, then there may be other cell types contributing to the score even though it is based on DEG that does not mean the absence of other cell type contributors. The proportion of elastic fibroblasts inferred from bulk data would be more meaningful. Also, why are the data in the KM curve limited to just 90 patients – TCGA has 500+ tumors and it would be important to show this in all patients. In addition, TCGA provides a rich source of clinical data and its possible that elastic fibroblasts merely reflect inflammation from smoking, for example, and therefore an MVA showing the presence of elastic fibroblasts has an independent effect on survival will be important. Finally, how are the elastic high and low strata defined – given the low number of patients included in this analysis I suspect only patients at the extreme were analyzed which is worrisome regarding the generalizability of these findings. I’d like to see the KM curve with all tertiles/quartiles/bins.”*

Response: We thank the reviewer for this comment. We have shown an independent association with survival in the HPV+ cohort of 89 patients and not the HPV- patients, now more clearly specified in the text. We now show both the effect of the top 100 DEGs from elastic fibroblasts as well as classical CAFs. Please see response to comment #17 of Reviewer #2 concerning the DEG cutoff.

We used Cibersortx for deconvoluting the TCGA samples. Gene signatures for each cell type were derived from our single cell dataset. In general, all deconvolution algorithms are more established for canonical and highly abundant cell types but not for substates of cells (in this case fibroblasts with elastic differentiation). In our analysis, 97/500 all HNSCC (10/89 for HPV+) patients have predicted non-zero fractions of elastic fibroblasts, while 458 and 496 have endothelial or epithelial cells respectively. The high and low groups are defined based on the median of the GSVA score calculated for the signature. All samples above the median are in the high group and below the median are in the low group. We have included all the patients are not just patients on extreme ends which was a concern mentioned above.

In addition, using MFAP4 IHC we were able to show fibroblasts with elastic differentiation in 6 HNSCC samples. By comparison, even though Treg are an established component of the HNSCC TME, they only get detected in 37/500 samples, while the “main cell category” of CD4 cells gets detected in 479/500 samples. In conclusion, we believe that while it is a helpful tool in general, deconvolution might not be sensitive enough for certain cell subcategories in this cohort. Consequently, we did not include deconvolution in the definition our elastic score.

14. *“Elastic fibroblasts are proposed to have a specifically worse impact on HPV+ patients but 1) this is not clearly shown as far as I can tell anywhere in the paper and 2) the elastic fibroblasts are predominantly found in HPV- patients based on the UMAP.”*

Response: The survival analysis (now **Fig. 5D**) only includes HPV+ patients (see our response to the previous comment).

15. *“I find Fig 6 problematic – the differences between inflamed and non-inflamed tumors are not substantial or really convincing in my opinion. There is no statistical quantification and “by eye” the changes are quite modest. The interactions as described would really need to be confirmed by IHC to have any real external validation.”*

Response: We have updated the content of **Fig. 7A** (former **Fig. 6B**) to represent our new inflammation scoring system based on tertiles, similar to a prior publication (Mandal et al, *JCI Insight*, 2017). After comparing patient-specific frequencies and average expression levels of individual ICR and ICL transcripts in specific cell types using the Kruskal-Wallis and Dunn’s multiple comparisons tests, we agree that most of the genes evaluated are not affected by inflammation scores. Thusly, we have toned down our figure description and conclusions to mirror the data. We did, however, observe that LAG3 expression levels were elevated in macrophages from highly inflamed lesions (new **Suppl. Fig. 8**) and this is briefly described on p.13 lines #298-300 and p.19 lines #448-452. Regarding confirming cell-to-cell interactions by IHC, please see our response to the next comment.

16. *“Having a higher % of PDL1 positive cells is not the same as showing a given cell type is the major contributor of PDL1. There is a hole hear between the data and the conclusions. The authors do not draw this careful distinction and the FACS analyses aren’t able to justify the conclusions as stated.”*

Response: This is an important point and similar to the one raised by Reviewer 1, comment #11. Flow cytometry provides us with a snapshot of cell phenotypes found within the HNSCC TME but not their spatial arrangement. While we initially provided the flow cytometric assessment of non-immune and myeloid cells that expressed PD-L1 (new **Fig. 8A**), we now confirm this observation by multicolor immunofluorescence (**Fig. 8C-D**). As an additional corroboration, we describe close spatial associations between CD8⁺ T cells and PD-L1⁺ macrophages, as well as PD-L1⁺ tumor cells within the HNSCC TME using multispectral imaging (**Fig. 8C-E** and **Suppl. Fig. 11A-B**; text

on pp.14-15). Cumulatively, these data support that macrophages are prominent contributors of PD-L1 to T cells within the TME.

17. *“The authors should really cite other important single cell studies of H&N including Sharma et al, Nature Communications, 2018. This represents a very important study in the field.”*

Response: We thank the reviewer for this comment. We have included the reference as requested (see Sharma et al., reference #12 in updated manuscript).

18. *“Are there other clinical factors that vary between patients such as prior radiation or chemo? Are all tumors primaries? I was surprised to see a 15 yo patient included in this study as most IRBs/tumor banks exclude children. Please confirm this age is accurate and covered by the IRB. Why do some tumors have no T-stage and no N-stage?”*

Response: We thank the reviewer for this comment. All biopsies were from primary tumors of treatment-naive patients. The 15-year-old patient had an IRB exemption and was thus correctly included. However, we removed this patient from the cohort due to the valid concern about biological variation. The T- and N stage of tumors was updated in **Suppl. Table 1**. Regarding incomplete TNM staging for patients HN05 and HN08, the reasons are as follows. Patient HN05 had no documentation of clinical depth of invasion of this primary tumor. Since depth of invasion is required to stage the primary tumor according to the 8th edition AJCC staging guidelines, this tumor could not be fully clinically T staged. Patient HN08 had a primary tumor that was incidentally found during surgery for a simultaneous gum primary and, consequently, had no clinical work up and could not be staged according to the 8th edition AJCC guidelines. As we did not have clinical staging for these two patients, we now provide their pathologic staging in **Suppl. Table 1**.

19. *“The marker based UMAPs are not really meaningful (S1C) as presented -- a UMAP of all clusters is really where this should be shown to demonstrate specificity for the various subpopulations. Why does the UMAP have a different topology depending on the marker if these are all DCs?”*

Response: After reanalyzing our dataset with a newer version of the clustering algorithm, we were able to identify a specific DC cluster as shown in **Fig. 1C**. Consequently, the figure of concern has been removed from this manuscript.

20. *“Why do the authors not see a separate cluster of regulatory T-cells as has been shown in other solid tumors among their T-cell clusters? They do show this in the original clustering.”*

Response: We thank the reviewer for this comment. UMAP visualizes the Leiden clustering generated by the scanpy pipeline. Regulatory T cells were identified by showing transcripts for CD3, CD4, ILR7 and strong FOXP3 (see also response to comment #6 from Reviewer #2). Treg are only mentioned in **Fig. 1C** and **Suppl. Fig. 1A-B**. If the reviewer refers to the Cillo et al study as “original clustering”, then differences in visualization, such as cell types forming separate clusters, are due to diverging clustering and visualization algorithms that underlie the figures in the two different studies.

21. *“S3A – Cluster 6 and cluster 4 (second from bottom) seem to be quite similar. It appears that cluster 4 might also be non-malignant cells. I am not sure what “intermediate status” that the authors refer to really means as this term is not used in the literature.”*

Response: We thank the reviewer for this comment. We have updated the figures (see general comments). We have identified the new clusters 13, 14 and 16 to be non-malignant with no cells of intermediate status being present.

22. *“There is mention of etiology dependent “spatial adjoinment” in the UMAP in 3A but I remind the authors that distance/closeness in a UMAP is not representative of similar expression and should not be interpreted or stated as such.”*

Response: Please see our response to Reviewer 2’s comment #4 on this issue.

23. *“What is the implication of variability in viral gene expression – is the conclusion merely that viral proteins have heterogeneous expression in HPV? This needs to be expanded.”*

Response: We thank the reviewer for this comment. We have expanded our analysis of this aspect of the dataset in response to Reviewer 1's comment #6 and Reviewer #2's comment #20. We agree that significant additional research is needed beyond the scope of this manuscript.

24. *"Where are the cancer associated pericytes on the original UMAP in Fig 1? I would typically think of pericytes as endothelial associated cells so I find their including in Fig 4 a bit strange."*

Response: We thank the reviewer for this comment. We have changed the overall outline of the paper by introducing an introductory figure describing the stroma, moving the pericytes to a separate figure (**Fig. 4C**).

25. *"The authors mention that inflamed tumors have greater T cell infiltrate and likely more favorable outcome as a result, yet inflamed tumors are also suggested to have more elastic fibroblasts and thus a worse outcome. This contradiction is bothersome."*

Response: We thank the reviewer for this comment. We cannot see where we made the claim that inflamed tumors have more elastic fibroblasts. We would note that, in the complex cancer-immune-stroma interaction, there are likely several mechanisms that influence survival or predict response to certain therapies. We have not used scRNAseq data itself for prognostic association, but rather as a discovery platform, then pursued validation in a sufficiently larger dataset (TCGA) with well-curated survival outcomes.

Reviewers' Comments:

Reviewer #1:

Remarks to the Author:

The authors have addressed all of my concerns and have significantly improved the manuscript. Limitations on the samples as previously noted do persist; however, the relative depth of sequencing in the specific populations is adequate. The multi-color IF data also adds convincing support for the differences reported, which importantly have been toned down in the conclusions/discussion portion of the paper. I believe that the manuscript will make a substantial positive impact in opening additional exploratory analysis surrounding the presence and role of these cells in HNSCC.

Reviewer #3:

Remarks to the Author:

I reviewed "Investigating immune and non-immune cell interactions in head and neck tumors by single cell RNA sequencing" again with interest, hoping the authors would come back with a substantial set of new analyses/experiments to address my feedback. While the authors have written an extensive reply, I continue to maintain scientific concerns and remain fairly underwhelmed by what the novel scientific insights are here. "Elastic fibroblasts" are not really matured as a concept with any sort of biological validation and the idea of activated fibroblasts expressing elastin is not novel, which was a comment provided in my initial review. Furthermore, the expression of PDL1 by TAMs has been well documented and the new IHC analyses do not do much of anything to go beyond this observation. The idea of investigating interactions in HNSCC as suggested by the title is really not done and its hard to extract any specific conclusion related to this title from the data presented. In addition, the authors do not really address most of my major concerns and did not add real, substantive changes to the manuscript; instead, they have focused on explaining away my comments for the most part. I therefore remain concerned about the manuscript in its current form, in spite of the authors' edits and modifications.

I will re-iterate my outstanding concerns as well as comment on some of the points made by the authors:

- 1) Use of additional tumors – the authors have made this point more clear in the revised manuscript, but to argue that these points were transparent in the original manuscript is not true. In the original manuscript, the authors did not articulate this in any reasonable form. The authors' point that others are entitled to re-analyze data is entirely valid, however, when this is done, it is clearly acknowledged and noted. This was not the case in the original manuscript.
- 2) Demonstrating PBL clustering by patients is a fairly low bar to show robustness of data. While I am pleased that the first author is the same person who processed samples for all studies, this point does not yet directly address the question. Robustness of the dataset should really be shown for the tumor cells (malignant and non-malignant) which are where greater variability is likely to be found compared to TILs.
- 3) Supplemental Figure 2C does not show the genes captured per cell.
- 4) The use of a uniform gene cutoff for PBL and tumor represents a continued flaw. The authors suggest a cutoff of 500 would be too stringent, but this missed the point that a more appropriate cutoff should have been used for tumor cells. The citations mentioned are not prominent single cell studies in the field, and it seems a bit surprising that the authors choose two references from low impact factor (<4) journals to justify their approach considering the 50+ major high impact studies using scRNA-seq.
- 5) The authors sorted tumors by CD45+ and CD45- thus they already have been able to "pull on the levers" of the cell type proportions, yet some tumors remain very low in epithelial fractions. I continue to have this concern and do not believe that magnetically sorted samples used for scRNA-seq intrinsically have this limitation, so remain a bit surprised and concerned about this.
- 6) The splitting of tumor into low, medium, and high addresses a portion of my comment in that regards, but now the medium inflamed tumors seems to have the lowest CD8+ T-cells. This raises concern again over the classification of tumors. I agree that there is not a good system to define such states in HNSCC, so why even use this metric. The authors own data does not support it. Note, there are ways to actually look at true cell type proportions such as bulk RNA-seq with

deconvolution, but the authors did not do this.

7) The authors indicate they removed the pseudotime plots in response to my comment that such analyses are not informative across tumors (regardless of the cell type being analyzed), yet pseudotime analyses are still present in the manuscript (Fig 2B).

8) Demonstrating consistency in PBLs is a fairly unconvincing method to suggest there are no batch effects due to longer processing or ischemia time. The tumors cells are what need to be demonstrated. I don't agree that technical artifacts would affect all samples similarly, as ischemia time is well documented to affect samples at a transcriptomic and proteomic level.

9) Fibroblasts with elastic differentiation in single cell analyses primarily came from HPV- tumors based on Fig 5, with only one HPV+ sample contributing to this cluster. Yet, the survival analysis is limited to HPV+ tumors only. I ask again: What does the same analysis show for HPV-? I described my interest in these analyses in my first critique and the authors have not addressed it. One would expect similar findings if this concept is biologically relevant and I find it curious that the TCGA analysis only utilized HPV+ tumors for which numbers are low (as reflected by the number at risk in 5D). In addition the authors do not address my comments about not simply dividing their survival analysis cohort in high and low and a request for seeing tertiles/quartiles to confirm a step-wise effect.

10) While deconvolution analyses are generally established for highly abundant cell types, it is not necessarily the case that unique populations (especially if defined by a high degree of DEGs) cannot be identified. In fact, this is one of the beautiful aspects of CIBERSORTx, which the authors used. CIBERSORTx should be able to independently pull out the fibroblasts with elastic differentiation, and its unclear why the authors did not use this with this methodology. I appreciate the clarification that high and low cohorts were those above and below the mean, respectively, but the authors do not comment on survival in any of the HPV- tumors which is strange.

11) The MFAP4 validation is shown for two tumors, but the authors mention six tumors had this pattern yet do not mention this in the text or show this in the figures, only in the response. Why not include this and show it if it is as strong as suggested?

12) The authors acknowledge that the high and low inflammation gene sets are not significantly different – this is concerning. While I appreciate the toned down description, the authors still state that “tumor inflammation status may associate with unique immune checkpoint signatures that could be used to tailor therapeutic strategies.” I don't think showing higher LAG3 expression in macrophages is really enough to support this statement, and without that, there really is not much that is coming out of these analyses. I don't see any statistical tests performed with correction for multiple testing. Additionally, its not clear what the plots in Fig 7B and C contribute.

13) The new MFI analyses are underwhelming – A) I don't see much co-staining of macrophages with PDL1, B) the correct analysis for 8E should be looking at PD1+ macrophages in relationship to exhausted T-cells (not any T-cell), C) the comparison to panCK+ cells seems strange as this is the lowest possible bar one could set (showing that macrophages have PD1 > malignant cells is already established). D) cutoff of 35 um is very arbitrary.

As requested, I am providing my feedback on the comments of Reviewer #2.

1) The authors address a number of the original criticisms by simply removing the data on the R-L interactions. This not only undercuts the main title of the paper and findings, but also fails to address the deeper underlying issue of the quality of the analysis and data. As noted inflamed tumors would be expected to have more interactions, yet the authors did not see this. Simply removing the data does not address this point. A similar example is the removal of HN05 (15 yo patient).

2) The additional analyses related to CD4 are clear and adequate,

3) Although the authors have added the inflammation heat map in Fig 2D (bottom left) as requested, the density of cells here does not match the other plots (e.g. cluster 3). Something is not done correctly here.

4) Picking DEGs is certainly an art – but a volcano plot is often helpful to provide a visualization of relevant fold change and p-value to help identify the most relevant DEGs. Just because the analysis is subjective, does not mean it cannot be addressed rigorously.

5) The new analyses of HPV expression raise more questions than answers – is the difference in viral expression simply related to drop outs rather than meaningful? L1 and L2 are notoriously hard to detect so I think comments on the absence of signal in these transcripts is very risky.

What HPV type were the tumors analyzed and did the authors use the corresponding annotations for their analysis for that type? The comments in the current form of the results on targeting viral transcripts is highly speculative and not particularly helpful.

RESPONSE TO REVIEWER COMMENTS

General comments

Since there were common concerns and some critiques were raised several times throughout the reviewer comments, we would like to address them here in a general and unified approach, so that we can refer back to them, where appropriate.

Batch effects

Batch effects are a general concern in experimental sciences as differences between groups might result from technical rather than biological variation. In the context of single cell RNA sequencing (scRNAseq), the concern for batch effects arises in the context of combined datasets across studies or large multi-institutional efforts to assemble datasets for example the human cell atlas (Haghverdi et al, 2018, *Nature Biotechnology*; Tran et al., 2020, *Genome Biology*). The critique of a batch effect in this study is now addressed in several respects

1. Since batch effect is a repeated concern by reviewer #3, we tested for it using the CellMixS package (Lütge et al. *Life Science Alliance*, 2021) to calculate a Cellspecific Mixing Score (cms) for our dataset. We split the cells into 3 groups – PBL, immune cells and non-immune cells. As expected, the cms is very high in the PBL and TIL (and thus, there is a low suspicion of batch effect). In the non-immune cells, malignant epithelial cells show low cms (as expected), but even the non-malignant cell of the tumor stroma (endothelial cells, fibroblasts) also show a high cms. This supports our visual observation of good mixing across patients (except for tumor cells, as expected) using a statistical tool.

this even further, we compared the cms of our PBL data before and after batch correction using the harmony algorithm (Korsunsky et al. *Nature Methods*, 2019) (see below). The cms did not improve suggesting that there is no evidence of batch effect.

2. This is a single-center study with one laboratory performing the logistics uniformly across the study (same cell dissociation protocols and “ischemia time,” same library-preparation technologies, same sequencing method, see **Methods** section on pages 20-21). Thus, batch

effect by design is less of a concern in this study than in multi-center studies or integrated datasets.

3. The peripheral blood lymphocytes (PBL) from each patient were handled in parallel technically and temporally, along with the tumor dissociation and FACS-sorting, so that they are exposed to the same “stress” as the TIL and non-lymphocytes. This was done even though the peripheral blood is already highly pure for viable lymphocytes. In PBL clustering, there is no indication of a batch effect.
4. It was suggested that the PBL are not a suitable control, but rather that tumor cells should be used to check for batch effects. This is a problematic suggestion. Several high-impact papers, also in HNSCC, have shown that malignant cells cluster by sample of origin/patient (Tirosh et al., *Science*, 2016 in Melanoma Fig. 1C; Puram et al., *Cell*, 2017, in HNSCC Fig. 2C; Couturier et al. *Nature Communications*, 2020, Glioblastoma Fig. 1A) and as such are highly unsuitable to check for technical variation due to their inherent patient-specific variation. Just to reiterate this point: the ideal cell type for a technical control are cells that are biologically very similar between patients (such as healthy mucosa, skin, or blood) and not tumor cells which are perturbed by individual somatic mutations.
5. It was suggested that processing induces a batch effect via a stress reaction in the cells such that ischemia reactions are induced. Even though this is theoretically possible, processing and dissociation techniques similar to those employed here were used in previously published high-impact reports (e.g., Lambrechts et al. 2018, *Nature Medicine*, reference #11, Cillo et al. 2020, *Immunity*, reference #9, Couturier et al. *Nature Communications*, 2020). Further, it has been shown that even harsher methods such as freeze-thaw protocols do not induce significant technical variation (Guillaumet-Adkins et al., *Genome Biology*, 2017). We agree that standardized and validated methods of cell extraction and sample processing are desirable. We maximized this technically and confirmed it bioinformatically (see above).

Gene cutoffs

The gene cutoff is used uniformly across the whole dataset of a given study during data aggregation to exclude low quality cells. The gene cut-off is dependent on the mix of cell types included in the study. One challenge is that apoptotic cells with a high natural gene count (e.g., tumor cells) might be similar to viable quiescent cells with a low natural gene count (e.g., PBL). As such deciding on a gene cutoff is a balance of cell yield and cell viability/quality. Of note, the cell quality can be further controlled using separate metrics such as mitochondrial cut-offs to exclude dead or dying cells, as done in this study.

To put the QC-parameters chosen by us into perspective, we have reviewed the literature to find comparable studies (using 10x Genomics 3' v2 kit, in solid tumors, mixed sample of immune and non-immune cells, published in high-impact journals). This shows that among the 6 studies compared below, our study has one of the most stringent quality control parameters, using the highest lower gene cut-off (200), the lowest higher gene cut off (> 5000) and the lowest mitochondrial gene cut-off (10%).

Author	Year	Journal	Cancer type	Cell types	Platform	Lower gene cut-off	Upper gene cut-off	Mitoch. cut-off
Bi et al	2021	Cancer Cell	Renal	Immune + non-immune	10x 3' (v2)	< 200	N/A	> 25 %
Wang et al	2021	Nature Med	Gastric	Immune + non-immune	10x 3' (v2)	< 200	N/A	> 15 %
Ireland et al	2020	Cancer Cell	SCLC	Immune + non-immune	10x 3' (v2)	< 100	> 10000	> 25 %
Durante et al	2020	Nature Com	Uv. mela.	Immune + non-immune	10x 3' (v2)	< 100	> 8000	> 10 %
Lambrecht et al	2018	Nature Med	NSCLC	Immune + non-immune	10x 3' (v1+2)	< 101	> 6000	> 10 %
Kürten et al		Nature Com	HNSCC	Immune + non-immune	10x 3' (v2)	< 200	> 5000	> 10%

We agree that common standards for setting quality parameters in data aggregation are desirable. However, establishing those methods is beyond the scope of this study. By putting our methodology in the context of existing literature, we show that our selected cutoffs are well within the range of what is supported by others' publications.

CIBERSORTx

We agree that imputing proportions of cell states from bulk RNAseq data using tools such as CIBERSORTx (Newmann et al., *Nature Biotechnology*, 2019) provide a useful application of scRNAseq data. As an extension and further application of our dataset, we have now included the results from deconvoluting bulk RNAseq (please see below). Due to file size limitations on the Cibersortx server, we randomly sampled 5000 tumor cells from our cohort and used these to create the single cell reference matrix. We then used the "Impute Cell Fractions" module to deconvolute the TCGA samples. 199/500 samples were predicted to have non-zero fractions of elastic fibroblasts.

While we were hesitant to add such an analysis to our study, the left panel of the figure above was added to the manuscript (**Suppl. Fig. 6C**) and briefly described in the fibroblasts section:

“The presence of these fibroblast sub-states was confirmed in the bulk RNASeq data from the HNSCC TCGA cohort using CIBERSORTx for deconvolution (Suppl. Fig. 6C), where 199/500 patient samples were predicted to have non-zero fractions of fibroblasts with elastic differentiation.”

Reasons why we were hesitant to add these data previously were:

1. In-depth analyses of the head and neck TME using such a tool have already been performed (Mandal et al, *JCI Insight*, 2016 reference #20 and Qi et al, *Cancers (Basel)*, 2021)
2. Recently, the limitations of such a tool became apparent – especially, when the single cell and bulk RNA sequencing was performed in different labs (Gustaffson, *PLOS One*, 2020)

Inflammation score

As acknowledged by the reviewers, there is no commonly agreed upon way to classify the head and neck TME with respect to TIL infiltration or inflammation. The classification into low, medium and high inflammation also seems to better represent the spectrum of immune infiltration in the eyes of the reviewers than a binary classification. There are two outstanding critiques with regards to the inflammation score.

1. The CD8+ T cells from scRNAseq do not linearly correlate with the low, medium and high infiltration.

In our view, this is not a conclusion that can be drawn since scRNAseq data is not a direct representation of the tumor microenvironment. For this, we would like to point the reviewer #3 to their own criticism from the last round of review not to infer quantitative information about the relative or absolute cell content of the sample from single cell data (comment #9 reviewer#3: “*I have major concerns with making any quantitative conclusions about cell type proportions based on droplet-based approaches*”). **Suppl. Fig. 2 B/C** is only a descriptive panel to show the composition of the dataset, not an actual representation of the HNSCC TME (which is why we performed H&E, IHC and Vectra multiplex staining), so the conclusion that “*now the medium inflamed tumors seems to have the lowest CD8+ T-cells*” cannot be drawn from the figure.

Further, in the context of the inflammation score and its CD8+ T cell component we would like to point out the following:

- As previously discussed, we used an H&E-based scoring system that is not relying on just CD8 frequencies but rather is a composite lymphocyte score of the total immune infiltrate. While we acknowledge the limitations in this regard, all similar studies like ours are subject to the issues raised above using scRNAseq for CD8 infiltration. Interpatient heterogeneity that exists when it comes to tumor microenvironments and their immune infiltrates requires several orthogonal approaches given current technical limitations.
- We do not think that only ‘clinical grade’ methods should be used in exploratory analysis. Inflammation status is only one aspect by which the head and neck TME is stratified in this study.
- As there is no established definition of an inflamed HNSCC TME, the CD8 content of the TME may not correlate with the total cellular immune infiltrate as measured in this study. In addition, absolute CD8 count might not be valuable without knowing the CD8 cell state (activated, naïve, memory), and location in the TME (Galon et al, *Science*, 2005).

2. The T cell inflammation panel (**Fig. 2A**) has a lower cell density than other panels.

This is due to the removal of PBL as explained in the text, methods and figure legend (answer to comment #3 of reviewer #3’s input on reviewer #2).

HPV expression

>95% of oropharyngeal HNSCC is caused by HPV16 (Ang et al., *NEJM*, 2010), which is why type specific testing is not routinely performed. However, we have now performed qPCR to quantitatively profile the relative abundance of HPV RNA for L1, L2, E6 and E7 transcripts. For a detailed description of the experiment, please see the newly added “Quantitative PCR (qPCR) for HPV genes” section under *Materials and Methods* on pp. 24-25. For 4/6 of our HPV+ samples, FFPE tumor sections were available and total RNA was isolated. The kit utilized for this experiment (Maxima H Minus First Strand cDNA Synthesis Kit, Thermo Fisher Scientific) uses random priming (not oligo dT for 3’UTR), which means it can detect the expression of all viral genes regardless of the presence or absence of 3’UTR in the transcripts. All 4 HPV+ samples tested were of the HPV16 subtype.

The relative expression levels of L1 and L2 genes were normalized to those of HPV16 E6 or E7. As shown below, L1/L2 expression levels were indeed substantially lower than those of E6/E7 (apart from L2 in HN18). These findings confirm our observations from scRNAseq data, and though they cannot

fully account for 5' or 3' UTR contributing to some alterations explaining level of scRNAseq detection, they do appear to show that low L1/L2 expression in scRNAseq data is not simply due to “drop out” using this novel technique, as low levels are detected.

We added the following description on lines 197-200:

“To validate these findings, qPCR was implemented to quantify the relative amount of HPV16 L1, L2, E6 and E7 transcripts in four HPV+ patients from our cohort. L1/L2 were generally expressed at lower levels than E6/E7 supporting our scRNAseq observations (data not shown).”

Reviewer #1

“The authors have addressed all of my concerns and have significantly improved the manuscript. Limitations on the samples as previously noted do persist; however, the relative depth of sequencing in the specific populations is adequate. The multi-color IF data also adds convincing support for the differences reported, which importantly have been toned down in the conclusions/discussion portion of the paper. I believe that the manuscript will make a substantial positive impact in opening additional exploratory analysis surrounding the presence and role of these cells in HNSCC.”

Response: We thank Reviewer #1 for the enthusiastic review.

Reviewer #3

I) “While the authors have written an extensive reply, I continue to maintain scientific concerns and remain fairly underwhelmed by what the novel scientific insights are here. “Elastic fibroblasts” are not really matured as a concept with any sort of biological validation and the idea of activated fibroblasts expressing elastin is not novel, which was a comment provided in my initial review.”

Response: We agree with the reviewer, which is why we have previously toned down our conclusions in comparison to our first draft. We do not claim that we have identified a new subset of fibroblasts. Rather, we claim that this cell subset/phenotype has not been well characterized in, nor has it been used as a prognostic biomarker for HNSCC patients.

II) “Furthermore, the expression of PDL1 by TAMs has been well documented and the new IHC analyses do not do much of anything to go beyond this observation.”

Response: We agree that it is a well-known fact that TAMs express PD-L1 within a TME. What is poorly understood is the relative contribution of PD-L1 by different cell types within the TME. Many investigators believe that tumor cells are the main contributors, which dictates how patients are selected for anti-PD-1 therapy in some tumor types. Our study indicates that, at least in the context of HNSCC, macrophages appear to be a primary source of PD-L1 based on scRNAseq, flow cytometry and IF data. Using multiplex *in situ* analysis, we show that not only is expression of PD-L1 higher on macrophages, but also that they closely associate with CD8+ T cells in the TME. This is of significant clinical importance as it gives support to the combined (PD-L1) positive score (CPS) of tumor cells,

lymphocytes and macrophages as a biomarker to select anti-PD-1-eligible patients (Cohen, EEW et al. *The Lancet*. 2019).

III) *“The idea of investigating interactions in HNSCC as suggested by the title is really not done and its hard to extract any specific conclusion related to this title from the data presented.”*

Response: We disagree with this statement for the following reasons:

1. Our manuscript is an overall view of the TME cell types with some focus on interactions of these cell types, and serves as a significant data resource that can be utilized by others to explore additional potential immune-non-immune cell interactions within the HNSCC TME.
2. In addition, this is the most comprehensive single cell study of the HNSCC TME in terms of the number of cells and cell types (immune and non-immune cells from HPV+ and HPV- disease). As such, in-depth description of non-immune cells (**Figs. 3-5**) must precede an investigation of immune and non-immune interactions.
3. We are specifically investigating immune-non-immune cell interactions in main **Figs. 7** and **8**, as well as **Suppl Fig. 12**. Consequently, we believe that our title is suitable for the analyses and findings we are presenting.

IV) *“In addition, the authors do not really address most of my major concerns and did not add real, substantive changes to the manuscript; instead, they have focused on explaining away my comments for the most part. I therefore remain concerned about the manuscript in its current form, in spite of the authors’ edits and modifications. I will re-iterate my outstanding concerns as well as comment on some of the points made by the authors (...).”*

Response: We are surprised that the Reviewer feels that we have not addressed most of their concerns. We spent a significant amount of time addressing several issues raised by all three reviewers and merging them into a unified story. Here is a list of substantive experimental and analytic changes to the manuscript:

- 1) We have reanalyzed all the scRNAseq data and updated the figures to address all the concerns raised by the reviewers.
- 2) We have generated new survival curves using TCGA dataset in response to Reviewers’ critiques.
- 3) In situ hybridization validation of E6/E7 genes (**Suppl. Fig. 5**) was added
- 4) Identification of fibroblasts with elastic differentiation status by MFAP4 IHC (**Suppl Fig. 7B; Suppl. Table 3**)
- 5) Expanding on our scRNAseq and FACS observations that macrophages are one of the main contributors of PD-L1 in the HNSCC TME by performing a 6-color multiplex immunofluorescence (**Suppl. Table 4, Fig. 8, Suppl. Fig 12**). Adding this advanced technique is challenging both in terms of wet-lab method as well as bioinformatic analyses.
- 6) We had performed CIBERSORTx analysis and described the results in response text in the previous round of revision. We are now adding it to the manuscript.

“1) Use of additional tumors – the authors have made this point more clear in the revised manuscript, but to argue that these points were transparent in the original manuscript is not true. In the original manuscript, the authors did not articulate this in any reasonable form. The authors’ point that others are entitled to re-analyze data is entirely valid, however, when this is done, it is clearly acknowledged and noted. This was not the case in the original manuscript.”

Response: We believed that we were transparent in our initial submission (discussing our dataset in the cover letter and in the *Methods* section of the original manuscript). As the reviewer acknowledges, we have now made those comments more visible to the reader (i.e., “Patient cohort” section within the *Methods*). We state that, “Patient characteristics are shown in **Suppl. Table 1**, and this cohort comprised transcriptomic profiles of CD45⁺ PBL and TIL (n=18), as well as CD45⁻ non-immune cells from patient tumors (n=15). Raw files and separate analyses from PBL and TIL isolated from HN01-

HN15 were previously published.⁹ All transcriptomic analyses of non-immune cell types and their interactions with immune cells, particularly CD8⁺ T-cells, are novel and unpublished. Data from all CD45⁻ cells, as well as well as their interactions, and PBL and TIL from HN16-18 are also unpublished.”

“2) Demonstrating PBL clustering by patients is a fairly low bar to show robustness of data. While I am pleased that the first author is the same person who processed samples for all studies, this point does not yet directly address the question. Robustness of the dataset should really be shown for the tumor cells (malignant and non-malignant) which are where greater variability is likely to be found compared to TILs.”

Response: We fully understand the concern about addressing technical questions. As we explain extensively above in the **general comments**, PBL are a suitable technical control for batch effect because they are primarily quiescent, and we believe that any difference in clustering most likely derives from technical artifacts (dissociation, staining, sorting). The “greater [biological] variability” of tumor cells is exactly what disqualifies them as a technical control, since they usually already cluster by patient, as shown by others (see above for references).

“3) Supplemental Figure 2C does not show the genes captured per cell.”

Response: We apologize. While the correct number is listed in the manuscript (**Suppl. Fig. 1B**, pp. 4-5) we mislabeled in the cover letter.

“4) The use of a uniform gene cutoff for PBL and tumor represents a continued flaw. The authors suggest a cutoff of 500 would be too stringent, but this missed the point that a more appropriate cutoff should have been used for tumor cells. The citations mentioned are not prominent single cell studies in the field, and it seems a bit surprising that the authors choose two references from low impact factor (<4) journals to justify their approach considering the 50+ major high impact studies using scRNA-seq.”

Response: As we explain extensively above in the answer to the **general comments** our QC metrics are very stringent and ensure a high-quality dataset. We compare our QC metrics to other high impact, published studies that used a uniform cutoff across immune and non-immune cells.

“5) The authors sorted tumors by CD45+ and CD45- thus they already have been able to “pull on the levers” of the cell type proportions, yet some tumors remain very low in epithelial fractions. I continue to have this concern and do not believe that magnetically sorted samples used for scRNA-seq intrinsically have this limitation, so remain a bit surprised and concerned about this.”

Response: We disagree with this statement for the following reasons:

1. Regarding epithelial cell fractions, there are many factors that could affect their numbers. The two most likely are: 1) interpatient heterogeneity and 2) intrinsic limitations of the 10X scRNAseq platform when it comes to non-immune cell analyses. While we share the concern about epithelial cell frequencies, this is something that we are unable to address unless we specifically enrich each non-immune population individually (which is beyond the scope of this study).
2. As we pointed out in our previous response letter, major high-impact papers using the 10x platform e.g., Lambrechts et al. 2018, *Nature Medicine*, have also seen high variability in epithelial fractions in their samples (in this study samples were not enriched by sorting but processed in a mix of immune and non-immune cells). Here, the by-eye estimation of cells in the “cancer” category ranges from 50-2500 cells across 5 patients (Lambrechts et al. 2018, Figure 1D)
3. Optimal sample dissociation and processing for scRNAseq, i.e., bulk vs. FACS-sorting vs. magnetic separation, is still an area of active research. The method used here is one possible option out of several, but comparing various methods for sample processing was not the aim of this study.

“6) The splitting of tumor into low, medium, and high addresses a portion of my comment in that regards, but now the medium inflamed tumors seems to have the lowest CD8+ T-cells. This raises concern again over the classification of tumors. I agree that there is not a good system to define such states in HNSCC, so why even use this metric. The authors own data does not support it. Note, there are ways to actually look at true cell type proportions such as bulk RNA-seq with deconvolution, but the authors did not do this.”

Response: As we explain extensively above in the **general comments**, the conclusion that “*medium inflamed tumors seems to have the lowest CD8+ T-cells*” is not supported by **Suppl. Fig. 2 B/C** most importantly due to the fact reviewer #3 previously pointed out: one should not draw “*quantitative conclusions about cell type proportions based on droplet-based approaches*”.

Concerning the deconvolution methods, while bulk RNAseq data could be a potential solution to profile the overall immune infiltration in tumor microenvironment (TME), we have exhausted our samples for scRNAseq and do not have the materials needed to perform bulk RNAseq for most of the patients in our cohort. Regarding deconvolution from bulk RNAseq, we would like to respectfully point out that *in silico* computational methods have limitations in assessing the true cell type proportions in TME (Gustaffson, *PLOS One*, 2020). In fact, those *in silico* methods are using scRNAseq data as the reference to develop and optimize their algorithms to deconvolute bulk RNAseq, not the other way around (Newmann et al., *Nature Biotechnology*, 2019; Wang, X. et al. *Nature Communications*, 2019). For these reasons, we have implemented a tissue-based lymphocyte scoring method to overcome the shortcomings of deconvolution algorithms to accurately quantify cell types from bulk RNAseq. Critically, tissue-based immunoscore methods allow us to obtain spatial information about the lymphocytes which is lost in bulk RNAseq.

“7) The authors indicate they removed the pseudotime plots in response to my comment that such analyzes are not informative across tumors (regardless of the cell type being analyzed), yet pseudotime analyses are still present in the manuscript (Fig 2B).”

Response: Pseudo-time analyses are present in Fig. 2B because as Reviewer #2 thought it was informative and suggested to move it there from **Suppl. Fig. 2D**.

We agreed with the reviewer’s previous comment – “(...) Pseudotime analyses across multiple tumors is fraught with error in my opinion and this analysis is not convincing or helpful (...)” – thinking that the reviewer meant tumor (= cancer cell) clusters, which usually group separately/ by patient of origin and for which pseudo-time analysis might be not suited. Therefore, we removed the Pseudotime analysis of cancer (malignant) cells.

For immune cells such as TIL, which cluster together across samples of origin, we find this analysis helpful. In this study, the CD8+ clusters align along a developmental trajectory from naïve to exhausted which is also biologically convincing. Pseudo-time analysis or “developmental trajectory inference” has been performed in many high impact scRNAseq studies, e.g., Guo et al, *Nature Medicine*, 2018, reference #42 and Savas et al, *Nature Medicine*, 2018, reference #45. Therefore, we kept the Pseudotime analysis of immune cells.

“8) Demonstrating consistency in PBLs is a fairly unconvincing method to suggest there are no batch effects due to longer processing or ischemia time. The tumors cells are what need to be demonstrated. I don’t agree that technical artifacts would affect all samples similarly, as ischemia time is well documented to affect samples at a transcriptomic and proteomic level.”

Response: As we explain extensively above in the **general comments** and in response to comment #2 of reviewer #3, unlike tumor cells that are affected by patient-specific somatic changes, PBL consist of normal cells that share more common inter-donor characteristics and, therefore, serve as a superior

reference for batch effect as any difference in PBL gene expression likely results in technical rather than biological variation.

“9) Fibroblasts with elastic differentiation in single cell analyses primarily came from HPV- tumors based on Fig 5, with only one HPV+ sample contributing to this cluster. Yet, the survival analysis is limited to HPV+ tumors only. I ask again: What does the same analysis show for HPV-? I described my interest in these analyses in my first critique and the authors have not addressed it. One would expect similar findings if this concept is biologically relevant and I find it curious that the TCGA analysis only utilized HPV+ tumors for which numbers are low (as reflected by the number at risk in 5D).”

Response: Please note that the critique was addressed previously by adding respective plots (**Suppl. Figure 8 A/B**) and briefly discussed on lines 263-264 of the updated manuscript. We will reiterate this here: For our updated analysis we showed the survival analysis for both CAF and elastic fibroblasts in the TCGA for both HPV+ and HPV- tumors (**Suppl. Fig. 8A/B**). Also, we show that the combination of low CAF/low elastic fibroblasts has the best overall survival in HPV+ patients (**Fig. 5D**).

“In addition the authors do not address my comments about not simply dividing their survival analysis cohort in high and low and a request for seeing tertiles/quartiles to confirm a step-wise effect.”

Response: The figure below shows the analysis of dividing the cohort in tertiles and quartiles both showing a trend towards significantly different survival in a stepwise approach. The log-rank test p-values are shown on the plot.

We also compared the p-values between the high and low groups using the survdiff function from the survival R package which uses the G-rho family test to compare two survival curves. The p-value between 1st and 3rd tertile is 0.01 when the samples are divided into three groups. On dividing the samples into quartiles, the p-value between the 1st and 4th quartile is 0.02.

“10) While deconvolution analyses are generally established for highly abundant cell types, it is not necessarily the case that unique populations (especially if defined by a high degree of DEGs) cannot be identified. In fact, this is one of the beautiful aspects of CIBERSORTx, which the authors used. CIBERSORTx should be able to independently pull out the fibroblasts with elastic differentiation, and its unclear why the authors did not use this with this methodology (...).”

Response: As we explain extensively above in the **general comments**, we see limitations in deconvolution tools as shown both in the literature as well as when we applied them to our data. Nonetheless, we now employ this method as a further application of our dataset.

“(…) I appreciate the clarification that high and low cohorts were those above and below the mean, respectively, but the authors do not comment on survival in any of the HPV- tumors which is strange.”

Response: In the previous response letter we showed the survival analysis for both CAF and elastic fibroblasts in the TCGA for both HPV⁺ and HPV⁻ tumors (**Suppl. Fig. 8A/B**). Also, we show that the combination of low CAF/low elastic fibroblasts has the best overall survival in HPV⁺ patients (**Fig. 5D**). We do not see any significance in HPV⁻ patients. We now briefly describe this observation on lines 263-264.

“11) The MFAP4 validation is shown for two tumors, but the authors mention six tumors had this pattern yet do not mention this in the text or show this in the figures, only in the response. Why not include this and show it if it is as strong as suggested?”

Response: To clarify, representative IHC images are shown for two of the six patients (**Suppl. Fig. 7B**) and the pathological scoring for all 6 patients were shown in the previous round of revisions in **Suppl. Table 3** and described on lines 252-255 of the updated manuscript. We chose to report a synthesized table of elastic fibroblast scoring by a trained pathologist since it adds more value and information than the raw IHC images. We provide representative IHC images of MFAP4 staining for the remaining 4 patients to the right.

12A) “The authors acknowledge that the high and low inflammation gene sets are not significantly different – this is concerning. While I appreciate the toned down description, the authors still state that “tumor inflammation status may associate with unique immune checkpoint signatures that could be used to tailor therapeutic strategies.” I don’t think showing higher LAG3 expression in macrophages is really enough to support this statement, and without that, there really is not much that is coming out of these analyses. I don’t see any statistical tests performed with correction for multiple testing.”

Response: Upregulation of PD-1 and LAG3 in “hot” and clinically favorable solid tumors is not a novel concept (Sobottka, B. et al. *Breast Cancer Res.* 2021; Ramy R. Saleh, et al. *Front. Oncol.* 2019), and we did observe this here also, though as a confirmatory and not statistically powered analysis. LAG3 is upregulated on lymphocytes as part of a feedback mechanism in response to IFN- γ and is commonly expressed with other checkpoints such as PD-L1. Furthermore, median LAG3 levels were elevated not only in macrophages, but also in DC (**Suppl. Fig. 9C**), NK and CD8 T cell populations from inflamed tissues (albeit these did not achieve statistical significance, $p > 0.05$; please see figures to the right).

“12B) Additionally, its not clear what the plots in Fig 7B and C contribute.”

Response: **Figs. 7B** and **7C**, cited on lines 310-318 on p.13 of our manuscript, represent simulated immune checkpoint receptor-ligand interactions between non-immune, myeloid and CD8+ T cells. These figures are important as they set stage for the idea that macrophages a major source of immune checkpoint ligands to immune cells (with specific focus on PD-L1 that is further evaluated in Fig. 8).

“13) The new MFI analyses are underwhelming – A) I don’t see much co-staining of macrophages with PDL1, B) the correct analysis for 8E should be looking at PD1+ macrophages in relationship to exhausted T-cells (not any T-cell), C) the comparison to panCK+ cells seems strange as this is the lowest possible bar one could set (showing that macrophages have PD1 > malignant cells is already established). D) cutoff of 35 um is very arbitrary.”

Response:

- A. Please revisit our representative staining image (**Fig. 8B**), as well as our analysis in **Fig. 8C**. It is clearly shown that average expression of PD-L1 on macrophages is statistically higher than on tumor cells [$p = 1.47 \times 10^{-05}$]. Also, these observations are further validated by flow cytometry data (**Fig. 8A** and **Suppl. Fig. 10**)
- B. To our knowledge, there is no single marker to identify exhausted cells, making IHC or multiplex IF not feasible currently. Rather, identification of exhausted cells requires a combination of multiple surface checkpoint receptors (e.g., PD-1, TIM-3, LAG3, etc.), functional assay (e.g., IFN- γ production) and gene expression (Tox, TCF-1, BLIMP). As we used a 6-color Vectra IF panel (using up 6 out of 7 possible channels) at present it is technically not possible or within the scope of this manuscript to specifically identify exhausted lymphocytes, which would go beyond our attempt to satisfy the previous concerns of the reviewer.
- C. To our knowledge this is the first study that has directly compared PD-L1 contributions (average expression and spatial arrangements) by macrophage and tumor cells side-by-side. Furthermore, we have explored PD-L1 expression on various myeloid and non-immune cell subsets by scRNAseq (Fig. 7) and flow cytometry (Fig. 8A). Based on aforementioned data, we have narrowed our comparison to macrophages and pan-CK+ tumor cells to further elucidate PD-L1 contributions within the TME.
- D. The cutoff was chosen based on size of cells and had been previously used in similar analysis (Ma, Z. et al. *Diagnostic Pathology*. 2017). An average macrophage size is 20-30 μm , while that of lymphocytes is 5-7 μm in diameter. Thus, 35 μm distance is supposed to represent 1-2 cell diameter distance between macrophages and lymphocytes, as a rational distance selected. To make this clearer to the reader, we added the following explanation to the legend of Suppl. Table 4:
“This distance represents 1-2 cell diameter distance between macrophages (20-30 μm in diameter) and lymphocytes (5-7 μm in diameter).”

Reviewer #3 input on the comments of Reviewer #2.

“1) The authors address a number of the original criticisms by simply removing the data on the R-L interactions. This not only undercuts the main title of the paper and findings, but also fails to undress the deeper underlying issue of the quality of the analysis and data. As noted inflamed tumors would be expected to have more interactions, yet the authors did not see this. Simply removing the data does not address this point. A similar example is the removal of HN05 (15 yo patient).”

Response: We disagree with this critique for the following reasons:

1. We removed the original figure because cellTalker, the package used, is an effective tool for general visualization of R-L interactions but not accurate quantitative and qualitative analysis. Consequently, we decided to remove these figures as they raised more questions than answers, which upon reflection we agreed with the prior reviewer’s point.
2. This does not undercut the main title of the page as we still explore R-L interactions using the CellPhoneDB, a well-established package for exploring R-L interactions.

3. We wish to clarify the data shown in the original Fig. 6A. We have not stated that inflamed tumors had less R-L interactions. In our first draft we stated that, “In contrast to inflamed tumors that had 63 predicted unique R-L interactions between CD8+ T and non-immune cell types, non-inflamed tumors had 42. The interactome between CD8+ T and APC had 653 shared interactions. 13 unique interactions were predicted in inflamed vs. 26 in non-inflamed tumors. This view of the HNSCC interactome descriptively demonstrates the complexity of in-flamed and non-inflamed tumors and stresses the unique roles that non-immune cell and APC-mediated interactions may play in the regulation of tumor inflammation status.” This statement does not mean that there were more overall interactions in non-inflamed tumors, but that there were more unique interactions (e.g., immunosuppressive ligands) that could be driving this particular tumor phenotype.
4. Patient HN05 was removed from our cohort at the recommendation of Reviewer 2, who pointed out that this case could be an outlier due to age-related issues and, because of that, should be removed.

“2) The additional analyses related to CD4 are clear and adequate,”

Response: We thank the reviewer.

“3) Although the authors have added the inflammation heat map in Fig 2D (bottom left) as requested, the density of cells here does not match the other plots (e.g. cluster 3). Something is not done correctly here.”

Response: Please note that the inflammation panel in Figure 2A (bottom left) was not added – it was present in the first manuscript draft (then Fig. 2D top right panel). Reviewer #2 requested the inflammation status to be added to the patient summary table (**Suppl. Tabl. 1**), which we did.

Concerning the different cell numbers between the panels of figure 2A (inflammation status vs. rest), this was extensively discussed in the answer to comment #12 of reviewer #2 in the previous response letter:

“Figure 2D panel inflammation why there are less cells?” **Response:** The number of cells is lower in the inflammation analysis because the peripheral blood cells were excluded from the analysis (lines #129-130 of the updated manuscript). Including them would be erroneous, because the inflammation score is a measurement of the TME that does not necessarily reflect the cells in the circulation. Also, patient HN03’s slides had no viable tumor cells (only necrosis and stroma) after its use for fresh digestion and scRNAseq, so it was excluded from analysis of inflammation status (lines #564-567 of the updated manuscript).”

Further, the lower number of cells in the inflammation UMAP is now clarified in the Materials and Methods (lines 582-583 of the revised manuscript) and figure legend (lines 961-962 of revised manuscript).

“4) Picking DEGs is certainly an art – but a volcano plot is often helpful to provide a visualization of relevant fold change and p-value to help identify the most relevant DEGs. Just because the analysis is subjective, does not mean it cannot be addressed rigorously.”

Response: We would like to elaborate on our choice of the gene cut-off used in the analysis. We picked the top 100 genes based on the z-score generated by the rank_genes_groups() function from the Scanpy package. By default, the genes are ranked by the z-score rather than log fold change or adjusted p-values. The z-score is more robust and hence we used that metric to pick the top 100 genes. We have added the following content to Methods to provide more clarification:

“A gene signature was created using the top 100 DEGs based on the z-score generated by the rank_genes_groups() function in Scanpy from fibroblasts with elastic differentiation as well as CAFs.”

To specifically address Reviewer’s question, we have generated a volcano plot showing all the genes in the comparison of elastic fibroblasts vs other types of fibroblasts is seen below. Red dots indicate the top 100 genes ranked by z-score. The cut-offs are drawn at 0.5 for the logFC and 0.05 for the adjusted p-value.

“5) The new analyses of HPV expression raise more questions than answers – is the difference in viral expression simply related to drop outs rather than meaningful? L1 and L2 are notoriously hard to detect so I think comments on the absence of signal in these transcripts is very risky. What HPV type were the tumors analyzed and did the authors use the corresponding annotations for their analysis for that type? The comments in the current form of the results on targeting viral transcripts is highly speculative and not particularly helpful.”

Response: We would like to note that the expression of HPV genes was present in the original manuscript (Fig. 3C and 3D of the original manuscript). During first round of revision, we had added the pathway analysis (Suppl. Fig. 5A of revised manuscript) to address comments from both Reviewer #1 and Reviewer #2.

The Reviewer’s technical question concerning the presence or absence of HPV genes and the HPV subtype are answered extensively above in the **general comments**.

Concerning the “comments (...) on targeting viral transcripts”, we were asked to “speculate on the virus state” by Reviewer #2 (comment #18 of first round of review) and to “elaborate on potential implications for therapeutic resistance in the discussion” by reviewer #1 (comment #5 of first round of review).

Reviewers' Comments:

Reviewer #3:

Remarks to the Author:

While the authors have toned down their claims and conclusions, I continue to remain concerned about the novelty of this study. The authors suggest the novelty is characterizing elastic fibroblasts and using this as a prognostic factor. It is unclear what the deep insights into elastic fibroblasts as described here really are and its prognostic value is in serious question (see below). In addition, the authors suggest that the study clarifies important observations about the relative contribution of PDL1 from TAMs, but this is already well described in many basic immunohistochemical studies. While the authors have done a better job of directly responding to my comments rather than argue them away in this round, I unfortunately have other concerns about aspects of the analysis/experiments:

1. Batch effects – the authors have now used a statistical pipeline to try to demonstrate there is no concern for batch effects. While I appreciate this approach, the cell mixing score is fundamentally limited by the ability to discern batch effects among malignant cells due to the overarching effects of inter-tumoral differences (which the authors and I both agree is expected). In this sense, the authors are unable to really demonstrate that batch effects related to processing across their different datasets are not contributing to their findings of inflammation, which might reflect ischemia time or other concerns. The simplistic analysis showing that cms did not change after a pipeline for batch correction is fairly underwhelming. I have previously outlined methods to demonstrate robustness of the data against batch effects and the authors have not performed these. In addition, the authors continue to focus on PBLs rather than tumor cells (TILs, non-malignant cells etc). For example, in addition to confirming their approaches in these other cell types, the authors could look at the distribution of Spearman correlation coefficients for genes and cells in comparison across different normalization methods. In addition, it would be important to see that the key clusters and conclusions such as identification of fibroblasts with elastic differentiation can be recapitulated across the distinct datasets that have been integrated here.

2. I am satisfied with the authors' responses to my comments about CIBERSORTx and also the gene cutoffs used.

3. The point about CD8 T-cells and correlation with inflammatory score continues to miss the obvious solution of bulk RNA-seq on the samples followed by deconvolution. The authors indicate that they do not have any additional tissue, but this could be done on FFPE which is likely not limiting. In addition, the authors do not address my comment that given the challenges with defining low, medium, and high inflammation, it is hard to justify even using this metric. I'm not asking for "clinical grade methods" just something that can be reliably used and calculated – the current approach is simply not reliable as executed/implemented. The authors note the limitations of using CD8 counts or content, yet the conclusions and emphasis on these observation in the manuscript remain dubitably unchanged.

4. I continue to find the results on HPV viral gene heterogeneity and expression tangential and speculative, with limited added value to the manuscript. At a minimum, I would simply remove them -- the conclusions on targeting these transcripts are not justified and it dilutes focus on the limited observations.

5. Concerns about CD45+/- cells: The authors are fair to suggest that inter-tumor differences could explain relative epithelial cell fractions; however, this fundamentally misses my concern that the epithelial fractions in some tumors are exceedingly low, well outside of what has been reported for other carcinomas and single cell sequencing studies that are well done. I do not believe this can be explained just by inter-patient heterogeneity as at the bulk level such dramatic differences are certainly not seen, thus raising concerns over the methods used and loss of specific cellular populations. Although the authors suggest I am asking them to compare methods of sample processing, I am not – I am merely suggesting that their method and its implementation is not adequate.

6. The authors try to quote my own words related to my comments about CD8 T-cells and the

inflammatory score. While this seems like a clever way to address my comment, it is misguided. The authors claim associations with inflammatory scores and the associated changes in the TME can be made (which I disagree with), but then when I point out flaws and inconsistencies (even if I take their conclusions at face value), they say we cannot make those conclusions according to the reviewer. Either the first observation is flawed (which is problematic) or the second point is true and their data demonstrate inconsistencies – regardless, it remains a concern.

7. Pseudotime even for immune cells is fraught with error when captured at a single time point. The prior studies are subject to these same flaws and I simply do not believe this is reliable without a temporal comparator of the same sample.

8. Thank you for clarifying the data in S8A and S8B related to elastic fibroblasts. While helpful this further reinforces my concern: Only one of their HPV+ samples contributed to the elastic fibroblast cluster with most of these cells arising from HPV- samples, yet survival data is from HPV+ tumors where these cells were not originally found. This is very strange because these elastic fibroblasts are not common in HPV+ tumors based on the authors' own data yet the survival analyses are in this cohort. It would be akin to making an observation in larynx tumors and then testing its prognostic significance in nasopharynx tumors (a completely different subsite with completely distinct biology) – it just does not make sense. I directly asked for the same survival analysis to be done in HPV- and unfortunately it shows that elastic fibroblasts do not affect survival in HPV- tumor (S8B, right), emphasizing my concerns about the robustness of the findings. Notably, Fig. 5 continues to use only HPV+ tumors so this portion of the response is irrelevant and ignores my original comment. Similarly, the reviewer figure with tertiles and quartiles provided is clearly not using HPV- tumors given the low sample size.

9. MFI analyses:

- a. PDL1 signal is not particularly associated with macrophages by eye – the images still don't really seem to show this (8B). There is plenty of PDL1 signal among the malignant cells and the merged image is underwhelming.
- b. I do not believe the authors can confidently parse out the PDL1 CD68+ cells from the PDL1 pan-CK+ cells when the CD68 cells are often infiltrating and among the tumor cells. The segmentation here would be challenging to yield results that are reliable.
- c. The comments on rationale for the bin size are helpful. Thank you.

10. While the authors suggest they performed a substantive list of additional new experiments, I would clearly note that most of the experiments listed were not done in response to my comments but rather other reviewers. Below are all comments from the first review the authors did not address and simply glossed over or explained away. Many of these comments have still not been addressed (directly quotes from first review):

- a. As noted in my original review, there is no MVA to suggest elastic fibroblasts are prognostic after adjusting for all other commonly used pathologic factors ("In addition, TCGA provides a rich source of clinical data and its possible that elastic fibroblasts merely reflect inflammation from smoking, for example, and therefore an MVA showing the presence of elastic fibroblasts has an independent effect on survival will be important.")
- b. "Could the inflammatory signature reflect post-processing or ischemia time? The phenomenon mentioned might be purely artefactual." I don't think the simple explanation that all were processed by the same individuals excludes this important confounder and have mentioned this twice now.
- c. "In Fig 4B, the same elastin markers do not come up in unbiased analyses raising further question."
- d. "Are there other clinical factors that vary between patients such as prior radiation or chemo?"

Response to reviewer #2 comments:

1. The authors state that they still make important observations about R-L interactions, but I am not sure what those are at this point. This has all been removed and what is left does not seem to do this, hence my concerns about the title and broader conclusions.
2. Thank you for clarifying the basis for the lower number of cells in 2D.
3. The volcano plot analysis raises concerns about the key genes for elastic fibroblasts – for example why isn't the marker MFAP4 that they use visualized.

Reviewer #4:

Remarks to the Author:

In this manuscript, Kurten et al. present an analysis of 136,947 cells from HNSCC tumors and peripheral blood. Although the authors claim to discover novel cell type-specific signatures associated with inflammation and HPV status and its prognostic value, the analysis results do not support the conclusion.

I agree with Reviewer 3's concern about the novelty of the paper and the evidence provided by the authors. Here are my comments that overlap with Reviewer 3's pending concerns:

3/ I agree with Reviewer 3's comment number 3 (also comment number 6 in round 2): Supplementary Figure 2B clearly shows that the medium inflamed tumors seem to have the lowest CD8+ T-cells, which is assumed to be correlated with the degree of inflammation. This indicates that the so-called "medium inflammation" should be classified as low inflammation.

If the authors think that "CD8+ T-cells" is not a "gold-standard" metric to indicate the degree of inflammation, then they should present other evidence demonstrating that their classification is accurate.

Also, I cannot see the difference between the heatmaps in Figure 7A (between low, medium and high). Unless the authors show some quantitative numbers, I would say there is no evidence that supports their conclusion. The authors should report the quantitative differences, as well as statistical significance of the difference (e.g., p-values of Wilcoxon or some other tests).

4/ I agree with Reviewer 3 that the results on HPV viral gene heterogeneity and expression seem to be speculative, and does not add significant value to the manuscript. Also, the conclusions on targeting these transcripts are not justified. Therefore, I would also suggest removing these results.

7/ I agree with Reviewer 3 that pseudo-time analysis does not bring much value in this particular situation, when the authors already have clearly defined cell stages in their data. If the authors want to keep the pseudo-time results, they need to provide a strong justification.

8/ This is also my main concern about the novelty/contribution of the article. I agree with Reviewer 3's comment number 8 (and also comment number 9 in round 2). The classification based on the signatures does not seem to be significantly correlated with the true survival, as shown in the Kaplan-Meier survival analysis.

First, in the comment 9 of round 2 (page 10 in the response), the authors plotted two figures in which they divide the patients into 3 and 4 groups. In these figures, the p-values are not significant in any case, which weaken their conclusions. I cannot find these figures in their updated manuscript, which is also concerning.

Second, Supplementary Figure 8 A and B clearly show that the classification only works on HPV+. This shows a rather weak prognostic value of these signatures.

Response to reviewer comments

Firstly, we would like to sincerely thank Reviewer #4 for stepping in as a new referee in the late stages of review process. We understand how difficult this may be, especially with all the communiqués and manuscript draft variations being evaluated. We will try to address all the issues raised to the best of our ability and to explain how critiques provided by previous reviewers guided the design of the current manuscript. We would also thank Reviewer #3 for once again providing us with feedback.

REVIEWER #3

“While the authors have toned down their claims and conclusions, I continue to remain concerned about the novelty of this study. The authors suggest the novelty is characterizing elastic fibroblasts and using this as a prognostic factor. It is unclear what the deep insights into elastic fibroblasts as described here really are and its prognostic value is in serious question (see below).”

Response: We appreciate Reviewer’s efforts and will try to address each of their questions to the best of our abilities. To re-state for clarity, there are several novel findings associated with our observations. First, fibroblasts with an elastic substate of differentiation have not been previously described in HNSCC. Second, we show a predictive potential of the elastic fibroblast gene signature, beyond that of cancer-associated fibroblasts (CAF), in the context of HPV+ HNSCC. Our aim was to validate our scRNAseq dataset as a valuable tool to explore the HPV⁻ and HPV⁺ HNSCC TMEs. Our aim was not perform an in-depth analysis of the function of fibroblasts with an elastic substate in the context of HNSCC. To make our claims clearer, we made changes to our Abstract. On lines 42 and 43 we now state that we, “describe the negative prognostic value of fibroblasts with elastic differentiation specifically in the HPV⁺ TME.” We also simplified our observations in text on p. 10, lines 239-243. We now state that:

“A negative prognostic impact was observed using the CAF signature scores in HPV⁺, but not in HPV⁻ samples (**Suppl. Fig. 8A**). Interestingly, the elastic fibroblast signature score also showed the same pattern in HPV⁺ samples (**Suppl. Fig. 8B**). We found that HPV⁺ patients with both low elastic fibroblast and CAF signature scores showed the best overall survival (p=0.0013, **Fig. 5D**)”

We address this point in greater detail in response to Reviewer #4’s review.

“In addition, the authors suggest that the study clarifies important observations about the relative contribution of PDL1 from TAMs, but this is already well described in many basic immunohistochemical studies.”

Response: Thank you for this comment. We would like to clarify our main point. We were not trying to prove that macrophages can express PD-L1. Instead, we were trying to determine, in an unbiased way, which cell type was the primary contributor of PD-L1 to CD8⁺ and CD4⁺ T cells in the head and neck squamous cell carcinoma (HNSCC) microenvironment. Using scRNAseq, flow cytometry and multispectral fluorescent microscopy (**Figs. 7-8; Suppl. Figs. 9-12**) we show that macrophages are the primary source of PD-L1 in this setting. This observation is of clinical significance as it gives support to the combined positive score (CPS) as an optimal biomarker for PD-1-based immunotherapy. Indeed, in HNSCC the standard of care predictive PD-L1 test is still undetermined, whether to test for tumor proportion score (TPS) of PD-L1 expression or combined positive score (CPS) which focuses also on macrophages and other inflammatory cells. Our data support the latter, using a novel methodology which can then be targeted more specifically, using this cellular source of PD-L1. To stress this point further, we made a small edit to our Abstract. On lines 45 we state that we, “show that tumor-associated macrophages are **dominant** contributors of PD-L1 and other immune checkpoint ligands in the TME.”

Finally, Reviewer #3 keeps stating how other studies have shown similar results to ours. We would greatly appreciate if the Reviewer could provide references supporting his claim.

“1. Batch effects – the authors have now used a statistical pipeline to try to demonstrate there is no concern for batch effects. While I appreciate this approach, the cell mixing score is fundamentally limited by the ability to discern batch effects among malignant cells due to the overarching effects of inter-tumoral differences (which the authors and I both agree is expected). In this sense, the authors are unable to really demonstrate that batch effects related to processing across their different datasets are not contributing to their findings of inflammation, which might reflect ischemia time or other concerns. The simplistic analysis showing that cms did not change after a pipeline for batch correction is fairly underwhelming. I have previously outlined methods to demonstrate robustness of the data against batch effects and the authors have not performed these. In addition, the authors continue to focus on PBLs rather than tumor cells (TILs, non-malignant cells etc). For example, in addition to confirming their approaches in these other cell types, the authors could look at the distribution of Spearman correlation coefficients for genes and cells in comparison across different normalization methods. In addition, it would be important to see that the key clusters and conclusions such as identification of fibroblasts with elastic differentiation can be recapitulated across the distinct datasets that have been integrated here.”

Response #1:

- A. Thank you for this comment. Epithelial cells are likely to be clustering away from one another due to patient-specific tumor differences, as numerous papers have shown for a heterogeneous disease like HNSCC. This in itself is an important finding of our single-cell dataset.
- B. Regarding outlining or suggesting the method(s) expected to demonstrate robustness of the data to satisfy this concern, we are open to any specific methods that we could undertake to demonstrate robustness of the data against batch effects. We have provided several analytic methods, all of which strongly support our conclusion against batch effects.
- C. We are puzzled by the claim that we have integrated “distinct datasets.” All of the data were generated by the same individual, in the same lab, using the same workflow pipeline. Our methodology and analysis has already been subjected to rigorous peer-review scrutiny and has been published (Cillo et al. *Immunity*. 2020; Ruffin et al. *Nature Communications*. 2021). Any specific analyses that we could perform in addition to these are welcomed, though we believe this has been shown in various methods so far and also in other publications.

“2. I am satisfied with the authors’ responses to my comments about CIBERSORTx and also the gene cutoffs used.”

Response #2: We are happy that we were able to clarify this point.

“3. The point about CD8 T-cells and correlation with inflammatory score continues to miss the obvious solution of bulk RNA-seq on the samples followed by deconvolution. The authors indicate that they do not have any additional tissue, but this could be done on FFPE which is likely not limiting. In addition, the authors do not address my comment that given the challenges with defining low, medium, and high inflammation, it is hard to justify even using this metric. I’m not asking for “clinical grade methods” just something that can be reliably used and calculated – the current approach is simply not reliable as executed/implemented. The authors note the limitations of using CD8 counts or content, yet the conclusions and emphasis on these observation in the manuscript remain dubitably unchanged.”

Response #3: Unfortunately, we have essentially exhausted FFPE blocks from these patients, as we had to address some of the critiques about HPV genes. In order to perform qPCR to quantitatively profile the relative abundance of HPV RNA for L1, L2, E6 and E7 transcripts, we had to use 8-10 sections at 10 µm thickness from the HPV+ patients. Furthermore, we had to utilize 5 µm serial sections for fibroblast staining, as well as to optimize and perform our Vectra multispectral microscopy staining. Consequently, using the remaining tumor sections from 15 patients we have performed inflammation scoring based on CD8 staining. These data are shown and discussed in greater detail under **Response #1** to Reviewer #4 below. We note that CD8 scoring by itself is not a perfect system either as CD8 can be expressed by other cell types, including NK cells. Our methods are as widely used and valid as any other, as we reference.

“4. I continue to find the results on HPV viral gene heterogeneity and expression tangential and speculative, with limited added value to the manuscript. At a minimum, I would simply remove them -- the conclusions on targeting these transcripts are not justified and it dilutes focus on the limited observations.”

Response #4: This is discussed under **Response #3** to Reviewer #4's critique.

“5. Concerns about CD45+/- cells: The authors are fair to suggest that inter-tumor differences could explain relative epithelial cell fractions; however, this fundamentally misses my concern that the epithelial fractions in some tumors are exceedingly low, well outside of what has been reported for other carcinomas and single cell sequencing studies that are well done. I do not believe this can be explained just by inter-patient heterogeneity as at the bulk level such dramatic differences are certainly not seen, thus raising concerns over the methods used and loss of specific cellular populations.”

Response #5: Thank you. The concern for losing cellular populations should be mitigated, since we show all of the expected cell populations, as well as sufficient epithelial cells for these studies, buttressed additionally by IHC from tissue sections. To recapitulate, it is not just interpatient heterogeneity that is affecting epithelial cell fractions but also intrinsic limitations of the 10X scRNAseq platform when it comes to non-immune cell analyses. This is something that we are unable to address unless we specifically enrich each non-immune population individually (which is beyond the scope of this study). As we pointed out in our previous response letter, major high-impact papers using the 10x platform (e.g., Lambrechts et al. 2018, *Nature Medicine*) have also seen high variability in epithelial fractions in their samples (in this study samples were not enriched by sorting but processed in a mix of immune and non-immune cells).

“6. The authors try to quote my own words related to my comments about CD8 T-cells and the inflammatory score. While this seems like a clever way to address my comment, it is misguided. The authors claim associations with inflammatory scores and the associated changes in the TME can be made (which I disagree with), but then when I point out flaws and inconsistencies (even if I take their conclusions at face value), they say we cannot make those conclusions according to the reviewer. Either the first observation is flawed (which is problematic) or the second point is true and their data demonstrate inconsistencies – regardless, it remains a concern.”

Response #6: We apologize for any misunderstanding or confusion. Regarding the critique that the CD8+ T cells from scRNAseq do not linearly correlate with the low, medium and high infiltration, we did not use these descriptive aspects of our scRNAseq dataset (i.e. to claim inflammation score based on the abundance of CD8+ T cells in these lesions) to make our main scientific conclusions. Rather, we are making our conclusions based on standard and well established, tissue-based cellular infiltration score of inflammatory cells, as well as cell type-specific gene signatures that are associated with specific inflammation levels. This is valid and well published, in an area without a single, accepted

standard. We believe that the reader will interpret our detailed inflammatory/infiltration data and single cell signatures, as appropriate for an evolving field.

“7. Pseudotime even for immune cells is fraught with error when captured at a single time point. The prior studies are subject to these same flaws and I simply do not believe this is reliable without a temporal comparator of the same sample.”

Response #7: We have removed this figure from the manuscript.

“8. Thank you for clarifying the data in S8A and S8B related to elastic fibroblasts. While helpful this further reinforces my concern: Only one of their HPV+ samples contributed to the elastic fibroblast cluster with most of these cells arising from HPV- samples, yet survival data is from HPV+ tumors where these cells were not originally found. This is very strange because these elastic fibroblasts are not common in HPV+ tumors based on the authors’ own data yet the survival analyses are in this cohort. It would be akin to making an observation in larynx tumors and then testing its prognostic significance in nasopharynx tumors (a completely different subsite with completely distinct biology) – it just does not make sense. I directly asked for the same survival analysis to be done in HPV- and unfortunately it shows that elastic fibroblasts do not affect survival in HPV- tumor (S8B, right), emphasizing my concerns about the robustness of the findings. Notably, Fig. 5 continues to use only HPV+ tumors so this portion of the response is irrelevant and ignores my original comment. Similarly, the reviewer figure with tertiles and quartiles provided is clearly not using HPV- tumors given the low sample size.”

Response #8: In order to clarify this point, one can see that HPV+ HN14, HN16 and HN18 patients are the primary contributors to the elastic fibroblast pool of cells. This is 50% of all the HPV+ samples evaluated. Only one HPV- patient (HN08) is heavily contributing to this fibroblast population. Consequently, we do not see how our data do not make sense or are not relevant when applied to the HPV+ TCGA cohort. If anything, our scRNAseq data indicate that elastic differentiation of fibroblasts may play a unique role in HPV+ lesion outgrowth, potentially opening the way for further study to develop specific therapeutic strategies targeting fibroblasts in this subset of patients. Please see our **Response #5** to Reviewer #4 for more details.

“9. MFI analyses:

a. PDL1 signal is not particularly associated with macrophages by eye – the images still don’t really seem to show this (8B). There is plenty of PDL1 signal among the malignant cells and the merged image is underwhelming.

b. I do not believe the authors can confidently parse out the PDL1 CD68+ cells from the PDL1 pan-CK+ cells when the CD68 cells are often infiltrating and among the tumor cells. The segmentation here would be challenging to yield results that are reliable.

c. The comments on rationale for the bin size are helpful. Thank you.”

Response #9: We appreciate this comment. In our image example, we wanted to display an image that shows macrophage presence in the stroma (including the outer layer of tumor border) and tumor, as well as expression of PD-L1 on macrophages and tumor cells. Second, if one pays close attention to representative single-color images for CD68 and PD-L1, the most intense PD-L1 staining was observed in macrophage-rich regions. Third, each technology comes with a unique set of challenges, as does multispectral IF. While we have performed rigid segmentation and identification of cell types for our IF analytical workflow, it is critical to validate microscopy observations using additional methods. Consequently, when we combine our scRNAseq (**Fig. 7A**), FACS (**Fig. 8A**) and multispectral IF data (**Fig. 8B-E**), we can conclude that macrophages are the primary source of PD-L1 in the majority of tumors evaluated.

“10. While the authors suggest they performed a substantive list of additional new experiments, I would clearly note that most of the experiments listed were not done in response to my comments but rather other reviewers. Below are all comments from the first review the authors did not address and simply glossed over or explained away. Many of these comments have still not been addressed (directly quotes from first review):

a. As noted in my original review, there is no MVA to suggest elastic fibroblasts are prognostic after adjusting for all other commonly used pathologic factors (“In addition, TCGA provides a rich source of clinical data and its possible that elastic fibroblasts merely reflect inflammation from smoking, for example, and therefore an MVA showing the presence of elastic fibroblasts has an independent effect on survival will be important.”

b. “Could the inflammatory signature reflect post-processing or ischemia time? The phenomenon mentioned might be purely artefactual.” I don’t think the simple explanation that all were processed by the same individuals excludes this important confounder and have mentioned this twice now.

c. “In Fig 4B, the same elastin markers do not come up in unbiased analyses raising further question.”

d. “Are there other clinical factors that vary between patients such as prior radiation or chemo?”

Response #10: First, we appreciate the enthusiasm, thoughtfulness and effort put into this review. To clarify, due to the Editor’s request that Reviewer #3 interpret our responses to previous Reviewer #2, we were in fact addressing Reviewer 2/3 jointly. We found no other substantial prognostic factors in univariate analysis to drive an MVA. Indeed, neither smoking nor extranodal extension as demonstrated by O’Sullivan (*Lancet Oncology*, 2016) has prognostic impact on HPV+ HNSCC, thus obviating such an analysis. Further, as a poorly understood disease HPV+ HNSCC biomarkers are urgently needed to segregate those with worse prognosis than the expected outstanding clinical outcome of HPV+ disease, given recent treatment deintensification strategies which should avoid these patients.

We have carefully measured ischemia time which we recorded and there were no substantial differences between specimen processing.

We note that ELN is the 33rd highest ranked gene (**Suppl. Table 2**). Also, MFAP4 and MFAP5, genes that are associated with EFs, are among the top 20 ranked genes (**Fig. 4B** and **Suppl. Table 2**).

These were treatment-naïve patients, as noted in the manuscript.

“Response to reviewer #2 comments:

1. The authors state that they still make important observations about R-L interactions, but I am not sure what those are at this point. This has all been removed and what is left does not seem to do this, hence my concerns about the title and broader conclusions.

2. Thank you for clarifying the basis for the lower number of cells in 2D.

3. The volcano plot analysis raises concerns about the key genes for elastic fibroblasts – for example why isn’t the marker MFAP4 that they use visualized.”

Response #11:

Comment 1. We respectfully disagree, since **Figs. 7** and **8** are exploring cell-cell interactions.

Comment 3. We respectfully disagree. **Fig. 4B** and **Suppl. Table 2** show that MFAP4 is the 19th ranked gene.

REVIEWER #4

“In this manuscript, Kurten et al. present an analysis of 136,947 cells from HNSCC tumors and peripheral blood. Although the authors claim to discover novel cell type-specific signatures associated

with inflammation and HPV status and its prognostic value, the analysis results do not support the conclusion. I agree with Reviewer 3's concern about the novelty of the paper and the evidence provided by the authors. Here are my comments that overlap with Reviewer 3's pending concerns."

Response: Before addressing each of the concerns individually, we would like to highlight the novelty of our manuscript, which has also been acknowledged by Reviewers 1 and 2 and by the editorial team, in hopes of convincing the editorial leadership and reviewer #4 that our submission is ready for publication

- This is the most comprehensive single-cell RNA sequencing (scRNAseq) dataset of the head and neck tumor microenvironment (134,606 cells).
- This is the first scRNAseq dataset that contains matching blood and tumor (cancer, stroma, and immune cells) samples from both HPV⁺ and HPV⁻ diseases, providing a rich dataset beyond our novel analyses here
- For the first time, HPV-16 encoded genes are added to the human genome reference file and their expression is quantified and visualized in epithelial cell transcriptomes from our scRNAseq cohort. This analytic strategy is important as it opens the way to other scientists to analyze virus-driven malignancies, which comprise nearly 20% of the world's cancer cases.
- We identify fibroblasts with an elastic substate of differentiation in HNSCC lesions. This population of fibroblasts has not been previously described in HNSCC. Furthermore, we show a prognostic potential of the elastic fibroblast gene signature, beyond that of cancer-associated fibroblasts (CAF), in the context of HPV⁺ HNSCC.
- We show that tumor-associated macrophages, and not tumor cells, are the key contributors of PD-L1 in the HNSCC TME, using relevant bioinformatic methods for predicting receptor:ligand interactions. This discovery can have a significant clinical impact when it comes to PD-L1 scoring and inhibitory treatment.

Comment #1 *"3/ I agree with Reviewer 3's comment number 3 (also comment number 6 in round 2): Supplementary Figure 2B clearly shows that the medium inflamed tumors seem to have the lowest CD8⁺ T-cells, which is assumed to be correlated with the degree of inflammation. This indicates that the so-called "medium inflammation" should be classified as low inflammation. If the authors think that "CD8⁺ T-cells" is not a "gold-standard" metric to indicate the degree of inflammation, then they should present other evidence demonstrating that their classification is accurate."*

Response #1: We thank the reviewer for this comment and would like to make three points:

1. For our initial submission, patients were segregated into two cohorts (low and high) based on medium infiltration score cut-off. We changed that analytic strategy as Reviewers 1 and 3 felt that patients should be segregated into at least three cohorts (low, medium and high inflammation scores) to better represent the continuum of immune cell tumor infiltration.
2. With the last remaining tumor sections available from 15 of our 18 scRNAseq patients we performed CD8 staining and scoring of inflammation in hopes of addressing Reviewer 3 and 4's concerns. As you can see in the figure on the right, the CD8⁺ T cell RNA count from scRNAseq does not linearly correlate with infiltration counts provided by either CD8 IHC or H&E staining. In contrast, we do see a statistically significant association between CD8 and H&E cell counts. Histological examination of tissue provides a more direct representation of the real TME composition than methods that use single cell suspensions as the latter is affected by tissue distribution (i.e. which tumor section was processed for scRNAseq and which was used to make

tumor blocks), processing (physical and enzymatic dissociation), flow cytometry sorting and library creation. So, Reviewer 4's conclusion that the medium inflammation group has the lowest CD8 cells cannot be drawn from the data provided. One could only say, "that from tumors that were classified as medium inflammation by H&E staining, CD8 cells had the lowest relative proportion in the scRNAseq dataset," which was digested, and cell proportions equalized with CD45-cells. This is a key point which necessitates the tissue-based *in situ* scoring which we employ here using all inflammatory cell scoring, or CD8 IHC-based scoring, both of which tightly correlate with each other ($p < 0.002$) as shown above in the right-most panel.

3. In all our comparisons we have compared the two "extremes", i.e. low vs. high inflammation. So even if scRNAseq was a direct representation of the TME, the low counts of CD8 in the medium group would not affect this comparison.

To resolve this ongoing issue, we have removed the Supplemental Figures 2A and 2B. They were supposed to facilitate an understanding of the relative contribution each cell type made to the low, medium, and high group inflammation group in the scRNAseq dataset, but we can see how this visualization may have been more confusing than helpful.

We have deleted the following description from the paper (page 5., line 115-127):

"When comparing inflammation-associated variability in relative cell frequencies, as expected, we observed an increase in CD8+ T cells in highly inflamed lesions (median: low-31.9%, medium-17.2% and high-50.9%; Suppl. Fig. 2B). This was offset by decreased macrophage (median: low-13.1%, medium-18.5% and high-4.03%) and CD4+ T cell (median: low-20.1%, medium-29.1% and high-11.2%) frequencies. While analyzing non-immune cells (Suppl. Fig. 2B), we noted an increase in relative epithelial cell numbers in highly inflamed vs. low and medium-inflamed TMEs (median: low-15.3%, medium-54.2% and high-77.9%) and a corresponding decrease in relative endothelial cell (median: low-49.8%, medium-28.7% and high-7.8%) and fibroblast numbers (median: low-17.2%, medium-14.9% and high-3.5%). Within PBL, a relative increase in circulating CD8+ T cells (median: low-8.5%, medium-12.3% and high-20.7%) and a relative decrease in CD4+ T cells (median: low-27.9%, medium-33.9% and high-20.7%) and monocytes (median: low-40.6%, medium-42.1% and high-31.9%) was observed (Suppl. Fig. 2C)."

Comment #2: "Also, I cannot see the difference between the heatmaps in Figure 7A (between low, medium and high). Unless the authors show some quantitative numbers, I would say there is no evidence that supports their conclusion. The authors should report the quantitative differences, as well as statistical significance of the difference (e.g., p-values of Wilcox or some other tests)."

Response #2: We agree with Reviewer 4 that differences in gene expression cannot easily be visualized by heatmaps, and did show quantitative differences, as requested by the Reviewer. Of all the genes evaluated in **Fig. 7A**, LAG3 expression was most dramatically affected by the inflammation status (**Fig. 7A**; **Suppl. Fig. 9A** and **Suppl. Fig. 9C**). To quantify this observation, we presented LAG3 data for macrophages and dendritic cells (**Suppl. Fig. 9C**), with statistical significance being confirmed by Kruskal-Wallis and Dunn's multiple comparisons tests.

Comment #3: *"4/ I agree with Reviewer 3 that the results on HPV viral gene heterogeneity and expression seem to be speculative and does not add significant value to the manuscript. Also, the conclusions on targeting these transcripts are not justified. Therefore, I would also suggest removing these results."*

Response #3: The reason we performed these studies was because we were urged to "speculate on the virus state" by Reviewer #2 (comment #18 from the first round of review) and to "elaborate on potential implications for therapeutic resistance in the discussion" by Reviewer #1 (comment #5 from the first round of review). In response to Reviewer 3's critiques, we validated scRNAseq findings and ensured that our observations were not driven by dropouts by studying HPV gene expression by *in situ* hybridization (**Suppl. Fig 5B**) and qPCR (shown in the previous response letter).

We note that this is, to our knowledge, the first demonstration of directly mapping viral transcripts in a virus-induced cancer using a single cell RNA sequencing cohort, opening the way conceptually and technically for further, detailed pathway analyses and validation.

To address Reviewer 4's critique, we removed the exploratory pathway analysis (**Suppl. Fig. 5A**) and the associated text on p. 9, line 201-203: "Cells expressing specific viral genes (e.g. E1 vs. all other) showed divergent gene pathways, though many of these fell into the categories of DNA transcription and translation, as well as protein synthesis (**Suppl. Fig. 5A**)."

Also, we rephrased and shortened our outlook in the discussion (p. 17, lines 412- 415):

From: "Further, the analysis of viral gene expression patterns highlights the heterogeneity of viral transcripts in HPV+ cancers at a cellular and an inter-patient level, *which has potential implications for immunotherapies and vaccines targeting viral neoantigens, especially in the context of therapeutic resistance.*" ...

To: "Further, the analysis of viral gene expression patterns highlights the heterogeneity of viral transcripts in HPV+ cancers at a cellular and an inter-patient level, *which may have potential therapeutic implications when targeting these transcripts*" ...

Comment #4: *"7/ I agree with Reviewer 3 that pseudo-time analysis does not bring much value in this particular situation, when the authors already have clearly defined cell stages in their data. If the authors want to keep the pseudo-time results, they need to provide a strong justification."*

Response #4: The pseudotime analysis was initially moved up from the supplemental figures as per Reviewer #2's suggestion (previously **Suppl. Fig. 2D**) and is a standard analysis technique for single cell data. In the context of this study it was valuable, as it recapitulated the developmental trajectory from naïve to exhausted CD8 T cells.

As both Reviewers #3 and #4 find this figure problematic, we removed **Fig. 2B** and the associated text (p. 6, lines 144-146):

“Pseudotime analysis showed the cells to be aligned on a transition from negative to high ICR-expression status, progressing from naïve to terminally dysfunctional cells on the pseudotime trajectory (**Fig. 2B**)”.

Comment #5: “8/ This is also my main concern about the novelty/contribution of the article. I agree with Reviewer 3’s comment number 8 (and also comment number 9 in round 2). The classification based on the signatures does not seem to be significantly correlated with the true survival, as shown in the Kaplan-Meier survival analysis. First, in the comment 9 of round 2 (page 10 in the response), the authors plotted two figures in which they divide the patients into 3 and 4 groups. In these figures, the p-values are not significant in any case, which weaken their conclusions. I cannot find these figures in their updated manuscript, which is also concerning. Second, Supplementary Figure 8 A and B clearly show that the classification only works on HPV+. This shows a rather weak prognostic value of these signatures.”

Response #5: We would like to address this critique by explaining how the review process shaped the current state of the manuscript. First, we would like to point out that the elastic fibroblast aspect of the manuscript was only one Figure and one descriptive sentence and that, from the beginning, we stated that the association with survival was present specifically in HPV+ patients.

From the original submission: *“Looking separately at the two diverging etiologies, we found our elastic fibroblast signature to be significantly associated with worse overall survival of HPV+ (p=0.006) (Fig. 4D) but not HPV–or all HNSCC patients (data not shown)”*

In response to the first round of reviewer comments, we added **Suppl. Fig. 8** to show individual survival curves for elastic fibroblasts and CAFs and added a combined graph to the main figures (**Fig. 5D**). We thereby, refined our analysis and showed that elastic fibroblast signature adds a *statistically significant*, prognostic benefit (p =0.001) to the CAF signature by creating a combined model (CAF + elastic fibroblast signature). Further, we provided MFAP4 staining by IHC from tissue as a confirmatory analysis. Since our original figure description caused confusion, we simplified our figure description. On p.10 of the revised submission, we state:

*“Given the impact of CAFs on patient survival, we explored the potential prognostic value of fibroblasts with elastic differentiation in HNSCC using bulk RNA sequencing data from The Cancer Genome Atlas (TCGA) database. A negative prognostic impact was observed using the CAF signature scores in HPV+, but not in HPV- samples (**Suppl. Fig. 8A**). Interestingly, the elastic fibroblast signature score also showed the same pattern in HPV+ samples (**Suppl. Fig. 8B**). We found that HPV+ patients with both low elastic fibroblast and CAF signature scores showed the best overall survival (p=0.0013, **Fig. 5D**).”*

We would like to point out that we also added the Cibersortx analysis in the last round of review as further confirmatory study. Concerning the figure with tertiles/quartiles, we provided this figure following the last round of reviewer comments because it was specifically requested. Indeed, when comparing the p-value between 1st and 3rd tertile the p-value is 0.01 and on dividing the samples into quartiles, and comparing 1st and 4th quartile the p-value is 0.02. The relatively low total number of HPV+ patients in the TCGA cohort (n= 89/497) and generally much better prognosis contribute to the borderline significance when the cohort is divided into smaller (3 or 4) subgroups.

Further, we do not agree that an association with survival that is only present in HPV+ patients limits the impact of our findings. Indeed, we believe that the opposite is the case. HPV+ and HPV- HNSCC are increasingly seen to be two clinically, prognostically and biologically separate diseases, so one

would expect them to have divergent prognostic biomarkers. Furthermore, since HPV infection is an increasing but a relatively new etiology of HNSCC, new prognostic biomarkers are needed for a tailored treatment approach for this subgroup of patients. Given the well-accepted, much better prognosis of HPV+ HNSCC patients, this is especially true in the era of de-escalation clinical trials in HPV+ patients (recently reported NRG HN-002 and ECOG 3311 trials published in *JCO*) for which prognostic markers would be especially valuable to avoid reducing treatment intensity in this higher-risk subset of patients.

We would conclude by saying that these extensive bioinformatic analyses give an intriguing assessment of the potential prognostic impact of elastic fibroblasts in the context of HPV+ HNSCC, which is appropriate for a novel, resource-type single cell sequencing paper. Future studies could follow up on these findings and provide more investigation, but are out of scope for the present manuscript.

Reviewers' Comments:

Reviewer #4:

Remarks to the Author:

In the new response and revision, the authors have clearly explained their novelty and scientific contribution. I am also satisfied that the authors have rephrased their claims to better reflect the findings of their work. They also removed the text that may cause potential confusion. Overall, the authors have satisfactorily addressed my concerns.